



# Forest-atmosphere exchange of reactive nitrogen in a low polluted area – temporal dynamics and annual budgets

Pascal Wintjen[1], Frederik Schrader[1], Martijn Schaap[2,3], Burkhard Beudert[4], and Christian Brümmer[1]

[1]Thünen Institute of Climate-Smart Agriculture, Bundesallee 68, 38116, Braunschweig, Germany
[2]TNO, Climate Air and Sustainability, Utrecht, 3584 CB, the Netherlands
[3]Institute of Meteorology, Freie Universität Berlin, 12165 Berlin, Germany
[4]Bavarian Forest National Park, 94481, Grafenau, Germany

**Correspondence:** Pascal Wintjen (pascal.wintjen@thuenen.de)

**Abstract.** Accurate modeling of nitrogen deposition is essential for identifying exceedances of critical loads and designing effective mitigation strategies. However, there are still uncertainties in modern deposition routines due to a limited availability of long-term flux measurements of reactive nitrogen compounds for model development and validation. In this study, we investigate the performance of dry deposition inferential models with regard to annual budgets and the exchange patterns of total

reactive nitrogen ($\Sigma N_r$) at a low-polluted mixed forest located in the Bavarian Forest National Park (NPBW), Germany. Flux measurements of $\Sigma N_r$ were carried out with a Total Reactive Atmospheric Nitrogen Converter (TRANC) coupled to a chemiluminescence dectector (CLD) for 2.5 years. Average $\Sigma N_r$ concentration was approximately 5.2 ppb. Denuder measurements with DELTA samplers and chemiluminescence measurements of nitrogen oxides ($NO_x$) have shown that $NO_x$ has the highest contribution to $\Sigma N_r$ ($\sim 52\%$), followed by ammonia ($NH_3$) ($\sim 22\%$), ammonium ($NH_4^+$) ($\sim 14\%$), nitrate $NO_3^-$ ($\sim 7\%$), and

nitric acid ($HNO_3$) ($\sim 6\%$). We observed mostly deposition fluxes at the measurement site with median fluxes ranging from -15 ng N m$^{-2}$s$^{-1}$ to -5 ng N m$^{-2}$s$^{-1}$ (negative fluxes indicate deposition). In general, highest deposition was recorded from May to September. $\Sigma N_r$ deposition was enhanced by higher temperatures, lower relative humidity, high $\Sigma N_r$ concentration, and dry leaf surfaces. Our results suggest that dry conditions seem to favour nitrogen dry deposition at natural ecosystems. For determining annual dry deposition budgets we used the bidirectional inferential scheme DEPAC (DEPosition of Acidifying

Compounds) with locally measured input parameters, called DEPAC-1D, as gap-filling strategy for TRANC measurements. In a second approach, the mean-diurnal-variation method (MDV) was applied to gaps of up to five days whereas DEPAC-1D was used for remaining gaps. We compared them to results from the chemical transport model LOTOS-EUROS (LOng Term Ozone Simulation – EURopean Operational Smog) v2.0 and from the canopy budget technique conducted at the measurement site. After 2.5 years, dry deposition based on TRANC measurements resulted in $(11.1 \pm 3.4)$ kg N ha$^{-1}$ with DEPAC-1D as

gap-filling method and $(10.9 \pm 3.8)$ kg N ha$^{-1}$ with MDV and DEPAC-1D as gap-filling methods. Both values are close to dry deposition by DEPAC-1D (13.6 kg N ha$^{-1}$) considering the uncertainties of measured fluxes and possible uncertainty sources of DEPAC-1D. The difference of DEPAC-1D to TRANC can be related to parameterizations of reactive gases or the missing exchange path with soil. 16.8 kg N ha$^{-1}$ deposition were calculated by LOTOS-EUROS for considering land-use class weighting. We further showed that predicted $NH_3$ concentrations, an input parameter of LOTOS-EUROS, were the main reason for

the discrepancy in dry deposition budgets between the different methods. On average, annual TRANC dry deposition was





4.5 kg N ha$^{-1}$ a$^{-1}$ for both gap-filling approaches, DEPAC-1D showed 5.3 kg N ha$^{-1}$ a$^{-1}$, and LOTOS-EUROS modeled 5.2 kg N ha$^{-1}$ a$^{-1}$ to 6.9 kg N ha$^{-1}$ a$^{-1}$ depending on the weighting of land-use classes within the site's grid cell. 7.5 kg N ha$^{-1}$ a$^{-1}$ was estimated with the canopy budget technique for the period from 2016 to 2018 as upper estimate and 4.6 kg N ha$^{-1}$ a$^{-1}$ as lower estimate. Our findings provide a better understanding of exchange dynamics occurring at low-polluted,
natural ecosystems and show opportunities for further development of deposition models.

## 1 Introduction

Reactive nitrogen (N$_r$) compounds are essential nutrients for plants. However, an intensive supply of nitrogen by fertilisation or atmospheric deposition is harmful for natural ecosystems and leads to a loss of biodiversity through soil acidification and eutrophication and may also threaten human health (Krupa, 2003; Galloway et al., 2003; Erisman et al., 2013). Atmospheric nitrogen load increased significantly during the last century due to intensive crop production and livestock farming (Sutton et al., 2011; Flechard et al., 2011, 2013; Sutton et al., 2013) (mainly through ammonia) and fossil fuel combustion by traffic and industry (mainly through nitrogen dioxide and nitrogen oxide). The additional amount of N$_r$ enhances biosphere-atmosphere exchange of N$_r$ (Flechard et al., 2011), affects plant health (Sutton et al., 2011) and influences the carbon sequestration of ecosystems such as forests (Magnani et al., 2007; Högberg, 2007; Sutton et al., 2008; Flechard et al., 2020), although the impact of increasing nitrogen deposition on forests carbon sequestration is still under investigation.

For estimating the biosphere-atmosphere exchange of N$_r$ compounds such as nitrogen monoxide (NO), nitrogen dioxide (NO$_2$), ammonia (NH$_3$), nitrous acid (HONO), nitric acid (HNO$_3$) and particulate ammonium nitrate (NH$_4$NO$_3$), the eddy-covariance (EC) approach has proven its applicability on various ecosystems. The sum of these compounds is called total reactive nitrogen ($\Sigma$N$_r$) throughout this manuscript. For evaluating fluxes of NO and NO$_2$ the EC technique has been tested in earlier studies (Delany et al., 1986; Eugster and Hesterberg, 1996; Civerolo and Dickerson, 1998; Li et al., 1997; Rummel et al., 2002; Horii et al., 2004; Stella et al., 2013; Min et al., 2014). In recent years, progress has been made in EC measurements of NH$_3$ (Famulari et al., 2004; Whitehead et al., 2008; Ferrara et al., 2012; Zöll et al., 2016; Moravek et al., 2019). First attempts in applying EC had been made on HNO$_3$, organic nitrogen molecules, nitrate (NO$_3^-$), and ammonium aerosols (NH$_4^+$) (Farmer et al., 2006; Nemitz et al., 2008; Farmer and Cohen, 2008; Farmer et al., 2011). Due to typically low concentrations, high reactivity, and water solubility, measuring fluxes of N$_r$ compounds is still challenging since instruments need a low detection limit and a response time of $< 1$ s (Ammann et al., 2012). Thus, fast-response instruments for measuring N$_r$ compounds like HNO$_3$ or NH$_3$ are equipped with a special inlet and short heated tubes to prevent interaction with tube walls (see Farmer et al., 2006; Zöll et al., 2016). However, these instruments need regular maintenance, have a high power consumption, and need a climate controlled environment for a stable performance. Considering the high technical requirements of these instruments, measuring fluxes of HNO$_3$ or NH$_3$ with these instrument is still challenging.

The Total Reactive Atmospheric Nitrogen Converter (TRANC) (Marx et al., 2012) converts all above mentioned N$_r$ compounds to NO. In combination with a fast-response chemiluminescence detector (CLD) the system allows measurements of $\Sigma$N$_r$ with a high sampling frequency. Due to a low detection limit and a response time of about 0.3 s the TRANC-CLD system





can be used for flux calculation based on the eddy-covariance (EC) technique. The TRANC-CLD system has been shown to be
suitable for EC measurements above a number of different ecosystems (see Ammann et al., 2012; Brümmer et al., 2013; Zöll et al., 2019; Wintjen et al., 2020).

Most of the mentioned EC studies about $\Sigma N_r$ or its compounds were carried out above managed field sites or close to agricultural or industrial emission hotspots, in order to focus on measuring the impact of environmental pollution or fertilization on (crop) plants. Only a few studies were conducted at remote locations, but were mainly focusing only on single $N_r$ compounds
(e.g., Wyers and Erisman, 1998; Horii et al., 2004, 2006; Wolff et al., 2010; Min et al., 2014; Geddes and Murphy, 2014; Hansen et al., 2015). At remote sites, concentrations of reactive $N_r$ compounds are typically low and close to the detection limit of the deployed instruments. Zöll et al. (2019) demonstrated that the TRANC-CLD system is able to detect concentrations and fluctuations of $\Sigma N_r$ accurately even at low concentrations of air pollutants. It was the first study presenting short-term flux measurements of $\Sigma N_r$ at that site with a focus on establishing a link between the drivers of both $\Sigma N_r$ and $CO_2$. For
a reliable prediction of $\Sigma N_r$ fluxes and annual budgets through the use of dry deposition (inferential) models, long-term flux measurements are needed to verify the background nitrogen load and examine natural exchange characteristics at low concentrations of $N_r$ compounds. Therefore, flux measurements at remote locations are required to improve deposition models and increase knowledge about the exchange behaviour of $\Sigma N_r$ under various environmental conditions.

During a measurement campaign instrumental performance issues and/or periods of insufficient turbulence arise, which
require a quality flagging of processed fluxes. Afterwards, the resulting gaps in the measured time-series need to be filled in order to properly estimate long-term deposition budgets. Known gap-filling strategies include the Mean-Diurnal-Variation (MDV) method (Falge et al., 2001), look-up tables (LUT) (Falge et al., 2001), non-linear regression (NLR) (Falge et al., 2001), marginal distribution sampling (MDS) (Reichstein et al., 2005), and artificial neural networks (Moffat et al., 2007). However, most of these methods have in common that they were originally designed for carbon dioxide ($CO_2$) or other inert gases.
Applying the MDS method to $\Sigma N_r$ is not recommended, since exchange characteristics during night-time, the light-response curve, and controlling factors of $\Sigma N_r$ differ from those of $CO_2$ (Zöll et al., 2019). It is, on the other hand, possible to use statistical methods like MDV or linear interpolation to fill short gaps in flux time series. This was done by Brümmer et al. (2013), but filling long gaps with this technique is not recommended. Since exchange pattern of $\Sigma N_r$ can substantially vary each day, it is questionable if statistical methods are suitable for $\Sigma N_r$ considering the high reactivity and chemical properties
of its compounds. Up to now, no common gap-filling procedure exists for $N_r$ compounds.

For nitrogen deposition assessments over large regions modeling approaches are needed due to low number of measurements. Chemical transport models (CTM) like LOTOS-EUROS (LOng Term Ozone Simulation (LOTOS) – EURopean Operational Smog (EUROS)) (Schaap et al., 2008; Wichink Kruit et al., 2012; Hendriks et al., 2016; Wichink Kruit et al., 2017; Manders et al., 2017; van der Graaf et al., 2020) and the Operational Priority Substance (OPS) model (van Jaarsveld, 2004) are the
method of choice. LOTOS-EUROS predicts the dry deposition of various $N_r$ compounds in a grid cell by utilizing meteorological data from the European Centre for Medium-Range Weather Forecasts (ECMWF) and information about the land-use class of the grid cell. Both CTMs use the deposition module DEPAC (DEPosition of Acidifying Components) (Erisman et al., 1994) for calculating deposition velocities. DEPAC is a dry deposition inferential scheme featuring bidirectional $NH_3$ exchange





(van Zanten et al., 2010). However, calculated budgets from CTM are affected by uncertainties in the emission of several $N_r$
compounds, transport range, (atmospheric) chemistry, and deposition processes. For improving models in these aspects, a validation to flux measurements is required. Such comparisons with novel measurement techniques are sparse and only available from few field campaigns.

It is also possible to use DEPAC as a stand-alone model for estimating dry deposition of $N_r$ compounds. For site-based modeling with DEPAC, decoupled from a CTM and henceforth called DEPAC-1D, only measurements of common micrometeorological variables and concentrations of the individual $N_r$ compounds are needed. Since all of these requirements were
measured at the study site, DEPAC-1D results can be used as a further gap-filling option. Hence, an estimation of the $\Sigma N_r$ dry deposition from flux measurements can be performed and a comparison of complete flux time series against DEPAC-1D and LOTOS-EUROS can be carried out for the measurement site.

Additionally, deposition measurements using the so-called "canopy budget method" of the forested and open land portion of
the site were conducted close to the flux tower. These measurements were taken after the International Co-operative Programme on Assessment and Monitoring of Air Pollution Effects on Forests (ICP Forests) (Clarke et al., 2010) established by the United Nations Economic Commission for Europe (UNECE). Measurements of the canopy outflow allow the calculation of the nitrogen deposition after canopy budgets technique (CBT) (Draaijers and Erisman, 1995; de Vries et al., 2003). Thus, we had the opportunity to compare four independent techniques for estimating the nitrogen dry deposition.

The study presented here is the first one showing long-term flux measurements of $\Sigma N_r$ above a remote forest and conducting comparison to different methods used for estimating nitrogen dry deposition. We discuss the observed flux pattern of $\Sigma N_r$ (1), investigate the influence of micrometeorology on the estimated fluxes (2), and compare the nitrogen dry deposition of LOTOS-EUROS with DEPAC-1D, flux measurements, and nitrogen outflow measurements based on CBT.

## 2  Materials and Methods

### 2.1  Site and meteorological conditions

Measurements were carried in the Bavarian Forest National Park (NPBW) (48°56'N 13°25'E, 807 m a.s.l) in southeast Germany. The unmanaged site located in the Forellenbach catchment ($\sim 0.69\,km^2$ (Beudert and Breit, 2010)) and is surrounded by a natural, mixed forest and is about 3 km away from the Czech border. Due to the absence of emission sources of $N_r$ in the surroundings of the measurement site, annual concentrations of $NO_2$ (1.9-4.4 ppb), NO (0.4-1.5 ppb) and $NH_3$ (1.3 ppb)
are low (Beudert and Breit, 2010). The site is characterized by low annual temperatures (6.1°C) and high annual precipitation (1327 mm) measured at 945 m a.s.l (Beudert pers. Comm.). Annual temperature in 2016, 2017, 2018 was 6.8°C, 6.9°C, and 8.0°C and precipitation was 1208 mm, 1345 mm, and 1114 mm, respectively. There are no industries or power plants nearby, only small villages with moderate animal housing and farming (Beudert et al., 2018). Due to these site characteristics, measurements of the $\Sigma N_r$ background deposition are possible. For monitoring air quality and micrometeorology a 50 m tower was
installed in the 1980s. Measurements of ozone, sulphur dioxide, and $NO_x$, the sum of NO and $NO_2$, have been conducted since 1990 (Beudert and Breit, 2010). The Forellenbach site is part of the International Cooperative Program on Integrated



Monitoring of Air pollution Effects on Ecosystems (ICP IM) within the framework of the Geneva Convention on Long-Range Transboundary Air Pollution (UNECE, 2020) and belongs to the Long Term Ecological Research (LTER) network (LTER, 2020). The Federal Environment Agency (UBA) and NPBW Administration have been carrying out this monitoring program

in the Forellenbach catchment, which is remote from significant sources of emission. The flux footprint consists of Norway spruce (*Picea abies*) and European beech (*Fagus sylvatica*) covering approximately 80% and 20% of the footprint, respectively (Zöll et al., 2019). During the study period, maximum stand height was less than 20 m since dominating Norway spruce are recovering from a complete dieback by bark beetle in the mid-1990s and 2000s (Beudert and Breit, 2014).

## 2.2 Experimental setup

Flux measurements of $\Sigma N_r$ were carried out from January 2016 until end of June 2018 at a height of 30 m above ground. A custom-built $\Sigma N_r$ converter (total reactive atmospheric nitrogen converter, TRANC) after Marx et al. (2012) and a 3-D ultrasonic anemometer (GILL-R3, Gill Instruments, Lymington, UK) were attached on different booms close to each other at 30 m height. The TRANC was connected via a 45 m opaque PTFE tube to a fast-response chemiluminescence detector (CLD 780 TR, ECO PHYSICS AG, Dürnten, Switzerland), which was set in an air-conditioned box at the bottom of the tower. The

CLD was coupled to a dry vacuum scroll pump (BOC Edwards XDS10, Sussex, UK), which was placed at ground level, too. The inlet of the TRANC is designed after Marx et al. (2012) and Ammann et al. (2012). The conversion of $\Sigma N_r$ to NO is split in two steps. Firstly, a thermal conversion occurs in an iron-nickel-chrome tube at 870°C resulting in an oxidization of reduced $N_r$ compounds. The thermal conversion of $NH_4NO_3$ leads to gaseous $NH_3$ and $HNO_3$. The latter is split up into to $NO_2$, $H_2O$, and $O_2$. $NH_3$ oxidized by $O_2$ at a platinum gauze to NO. HONO is split up to NO and a hydroxyl radical (OH).

Afterwards, a catalytic conversion takes place in a passively heated gold tube at 300°C while remaining oxidized $N_r$ species are further reduced to NO. In this process, carbon monoxide (CO) is acting as reducing agent. More details about the chemical conversion steps can be found in Marx et al. (2012). A critical orifice was mounted at the TRANC's outlet and restricted the flow to 2.1 L min$^{-1}$ assuring low pressure along the tube. The conversion efficiency of the TRANC had been investigated by Marx et al. (2012). They found 99% for $NO_2$, 95% for $NH_3$, and 97% for a gas mixture of $NO_2$ and $NH_3$. For determining

local turbulence - wind speed, wind direction, friction velocity ($u_*$) - measurements of the wind components ($u$, $v$, and $w$) were conducted using the sonic anemometer. Close to the sonic, an open-path LI-7500 infrared gas analyzer (IRGA) for measuring $CO_2$ and $H_2O$ concentrations was installed.

For investigating the local meteorology, air temperature and relative humidity sensors (HC2S3, Campbell Scientific, Logan, Utah, USA) were mounted at four different heights (10, 20, 40, and 50 m above ground). At the same levels, wind propeller

anemometers (R.M. Young, Wind Monitor Model 05103VM-45, Traverse City, Michigan, USA) were mounted on booms. Three leaf wetness sensors (Decagon, LWS, Pullman, Washington, USA) were attached to branches of a spruce and a beech tree near the tower. The branches of the beech tree were at heights of approximately 2.1 m, 5.6 m, and 6.1 m, the branches of the spruce tree at 2.1 m, 4.6 m, and 6.9 m. These measurements started in April 2016. For calculating the leaf wetness value, the following calculation scheme was conducted. If the leave wetness value, an arbitrary unit, was lower than 10, the leaf was

considered as dry. Otherwise, the leaf area surface was considered as wet. To take differences between the sensors into account,





all sensors were used to derive a wetness Boolean. Therefore, the number of dry sensors were counted for each half-hour: If at least three sensors were considered as dry, the corresponding half-hour was considered as mostly dry. A cleaning of sensors was not conducted because contamination effects could be corrected by implemented algorithms. Measurements of $NH_3$ were carried out by passive samplers at 10, 20, 40, and 50 m. DELTA measurements (DEnuder for Long-Term Atmospheric sampling

(e.g., Sutton et al., 2001; Tang et al., 2009)) of $NH_3$, $HNO_3$, $SO_2$, $NO_3^-$, and $NH_4^+$ were taken at the 30-m platform. Fast-response measurements of $NH_3$ were performed with an $NH_3$ Quantum Cascade Laser (QCL) (model mini QC-TILDAS-76 from Aerodyne Research, Inc. (ARI, Billerica, MA, USA)) at 30 m height, too. These measurements were used for inferential modeling of reactive nitrogen dry deposition. Further details about the location and specifications of the installed instruments can be found in Zöll et al. (2019) and Wintjen et al. (2020).

At the top of the tower (50-m platform), measurements of NO and $NO_2$ were conducted by the NPBW using a chemi-luminescence detector (APNA - 360, HORIBA, Tokyo, Japan). Measurements of global radiation and atmospheric pressure were also conducted at 50 m. Precipitation was measured at a location in 1 km southwest distance from the tower according to WMO (World Meteorological Organization) guidelines (Jarraud, 2008), and data were quality-checked by the NPBW (Beudert and Breit, 2008, 2010). Deposition was collected as bulk sample in weekly intervals in close vicinity to the tower using three

samplers at open site (bulk deposition) and 15 and 10 samplers beneath the canopy of a mature European beech and Norway Spruce stand (throughfall), respectively. This procedure is in common with the guidelines proposed by Clarke et al. (2010).

The canopy budget technique (CBT) is the most common method for estimating total and dry nitrogen deposition in ecological field research based on inorganic nitrogen fluxes ($NO_3^-$, $NH_4^+$) only (see Staelens et al., 2008, Table 1). Total deposition of dissolved inorganic nitrogen ($DIN_t$) was estimated on yearly basis after the CBT approach of Draaijers and Erisman (1995)

and de Vries et al. (2003) whose results differed only marginally and were therefore averaged. The biological conversion of deposited inorganic nitrogen into dissolved organic nitrogen (DON) in the canopy which is not addressed in CBT was estimated by the difference of DON fluxes between throughfall and bulk deposition ($\Delta$DON). Adding $\Delta$DON to throughfall DIN or to $DIN_t$ reveals a frame of minimum and maximum estimates of total nitrogen deposition $N_t$ and, by subtracting DIN deposition at open site from these $N_t$, of minimum and maximum estimates of dry deposition (Beudert and Breit, 2014).

## 2.3 Flux calculation and post processing

The software package EddyMeas, included in EddySoft (Kolle and Rebmann, 2007), was used to record the data with a time resolution of 10 Hz. Analog signals from CLD, LI-7500, and the sonic anemometer were collected at the interface of the anemometer and joined to a common data stream. Flux determination covered the period from 1 January 2016 to 30 June 2018. Half-hourly fluxes were calculated by the software EddyPro 7.0.4 (LI-COR Biosciences, 2019). For flux calculation a 2-D

coordinate rotation of the wind vector was selected (Wilczak et al., 2001), spikes were detected and removed from time series after Vickers and Mahrt (1997), and block averaging was applied. Due to the distance the from inlet of TRANC to the CLD, a time lag between concentration and sonic data was inevitable. The covariance maximization method allows to estimate the time lag via shifting the time series of vertical wind and concentration against each other until the covariance is maximized (Aubinet et al., 2012; Burba, 2013). The time lag was found to be about 20 s (see Fig. A1). We instructed EddyPro to compute





the time lag after covariance maximization with default setting while using 20 s as default value and set the range from 15 s to 25 s (for details see Wintjen et al., 2020). For correcting flux losses in the high-frequency range we used an empirical method suggested by Wintjen et al. (2020), which uses measured cospectra of sensible heat ($\mathrm{Co}(w, T)$) and $\Sigma \mathrm{N_r}$ flux ($\mathrm{Co}(w, \Sigma \mathrm{N_r})$) and an empirical transfer function. We followed their findings and used bimonthly medians of the damping factors for correcting calculated fluxes. On average, the damping factor was 0.78, which corresponds to flux loss of 22% (Wintjen et al., 2020).

The low-frequency flux loss correction was done with the method of Moncrieff et al. (2004), and the random flux error was calculated after Finkelstein and Sims (2001).

Previous measurements with the same CLD model by Ammann et al. (2012) and Brümmer et al. (2013) revealed that the device is affected by ambient water vapour due to quantum mechanical quenching. Excited $\mathrm{NO_2}$ molecules can reach ground state without emitting a photon by colliding with a $\mathrm{H_2O}$ molecule, thereby no photon is detected by the photo cell. It results

in a sensitivity reduction of 0.19% per $1 \, \mathrm{mmol \, mol^{-1}}$ water vapour increase. Thus, calculated fluxes were corrected after the approach by Ammann et al. (2012) and Brümmer et al. (2013) using the following equation:

$$F_{\mathrm{NO,int}} = -0.0019 \cdot c_{\Sigma \mathrm{N_r}} \cdot F_{\mathrm{H_2O}} \tag{1}$$

The NO interference flux $F_{\mathrm{NO,int}}$ has to be added to every estimated flux value. $c_{\Sigma \mathrm{N_r}}$ is the measured concentration of the CLD and $F_{\mathrm{H_2O}}$ the estimated $\mathrm{H_2O}$ flux from the LI-7500 eddy-covariance system.

After flux calculation, we applied different criteria to identify low-quality fluxes. We removed fluxes, which were outside the range of -420 $\mathrm{ng \, N \, m^{-2} s^{-1}}$ to 220 $\mathrm{ng \, N \, m^{-2} s^{-1}}$, discarded periods with insufficient turbulence ($u_* < 0.1 \, \mathrm{m s^{-1}}$) (see Zöll et al., 2019), fluxes with a quality flag of "2" (Mauder and Foken, 2006), and variances of $T$, $w$, and $\Sigma \mathrm{N_r}$ exceeding a threshold of two times $1.96\sigma$. These criteria ensure the quality of the fluxes, but lead to systematic data gaps in flux time series. Instrumental performance problems led to further gaps in the time series. Most of them were related to maintaining and repairing of

the TRANC and/or CLD, for example, heating and pump issues, broken tubes, empty $\mathrm{O_2}$ gas tanks ($\mathrm{O_2}$ is required for CLD operation), power failure, or a reduced sensitivity of the CLD. Considering the time period of ongoing measurements from the beginning of January 2016 till June 2018, the quality flagging resulted in 52.2% missing data. The loss in flux data is higher than values reported by Brümmer et al. (2013). However, they applied only a $u_*$ filter, which caused a loss 24%. In this study, the same $u_*$ threshold caused a flux loss of approximately 14.8%. 21% data loss from January 2016 to June 2018 is caused by

instrumental performance problems showing that TRANC-CLD system was operating moderately stable. For gap-filling we used DEPAC with locally measured input variables, here called DEPAC-1D. This procedure is described in Sect. 2.4.3.

## 2.4 Modeling fluxes as gap-filling strategy

### 2.4.1 Bidirectional resistance model DEPAC

DEPAC (Erisman et al., 1994) is a bidirectional resistance model, which models the canopy resistance $R_{\mathrm{c}}$ and determines

the effective compensation point for $\mathrm{NH_3}$. In addition to $R_{\mathrm{c}}$, the aerodynamic resistance $R_{\mathrm{a}}$ and the quasi-laminar boundary resistance $R_{\mathrm{b}}$ are also needed for the calculation of the deposition velocity and therewith the flux. $R_{\mathrm{c}}$ is the sum of parallel





connected resistances, which model the exchange behaviour of atmosphere with vegetation and soil: 1) stomatal resistance $R_{\text{stom}}$, 2) cuticular resistance $R_{\text{w}}$, and the soil resistance $R_{\text{soil}}$, which is connected in series to an in-canopy resistance $R_{\text{inc}}$. These resistances are treated differently for each $N_r$ compound. Further details about the implementation of the resistances for
each gas can be found in Sutton and Fowler (1993); Erisman et al. (1994); Van Pul and Jacobs (1994); Emberson et al. (2000); van Zanten et al. (2010); Wichink Kruit et al. (2010); Massad et al. (2010); Wichink Kruit et al. (2017).

### 2.4.2   Modeling of $\Sigma N_r$ deposition (LOTOS-EUROS)

DEPAC is integrated in the 3D chemical transport model LOTOS-EUROS. The land-use specific and total dry deposition is calculated by LOTOS-EUROS on hourly basis for each $N_r$ compound within a grid cell of $7 \times 7 \, \text{km}^2$. For this reason,
modeled concentrations, weather data from the European Centre for Medium-range Weather Forecast (ECMWF), and a land-use classification for each grid cell are needed. The land-use classification of the grid cell, in which the tower is located, was divided into 46.0% semi-natural vegetation, 37.2% coniferous forest, 15.9% deciduous forest, 0.7% water bodies, and 0.2% grassland. The land-use class weighting is based on the Corine Land Cover 2012 classification. However, the actual structure of the forest stand shows 81.1% coniferous forest and 18.9% deciduous forest within the footprint of the tower.
Due to the differences in the the distribution of vegetation types in the footprint of the tower, concentrations and depositions were recalculated with a corrected weighting of the land-use classes. The low contribution of coniferous forest and deciduous forest within the grid cell may be related to the evaluation of older aerial photographs showing larger areas of deadwood. Finally, the dry deposition of $\Sigma N_r$ is calculated as the sum of NO, $NO_2$, $HNO_3$, $NH_3$, and particulate $NH_4NO_3$ fluxes. The version of DEPAC used in this study differs from the one documented in van Zanten et al. (2010) in two main aspects: Firstly,
the implementation of a function considering codeposition of $SO_2$ and $NH_3$ (Wichink Kruit et al., 2017) in the non-stomatal pathway and secondly, the usage of a monthly moving average of $NH_3$ concentration for determining the stomatal compensation point (Wichink Kruit et al., 2012).

### 2.4.3   Site-based modeling of $\Sigma N_r$ deposition (DEPAC-1D)

As mentioned before, DEPAC-1D was used for filling the gaps in flux data. For running DEPAC as stand-alone, it was ex-
tended with a FORTRAN90 (Adams et al., 1992) program that allows the use of arbitrary input data sources. DEPAC-1D uses measured parameters of micrometeorology and concentration for the determination of $R_c$ and the compensation point of $NH_3$. The atmospheric resistances $R_a$ and $R_b$ and the fluxes of $NH_3$, NO, $NO_2$, and $HNO_3$ were calculated with a Python script. Parameterizations were done for $R_a$ after Garland (1977) and for $R_b$ after Jensen and Hummelshøj (1995, 1997) followed by stability corrections after Webb (1970) and Paulson (1970). $R_{\text{stom}}$ was calculated after Emberson et al. (2000). Further details
can be found in van Zanten et al. (2010). For estimating fluxes with DEPAC-1D, concentration measurements on monthly and half-hourly basis are used. $NH_3$ fluxes were based mostly on $NH_3$ half-hourly concentration measurements of the $NH_3$-QCL. Gaps in $NH_3$ concentration time series were filled with DELTA measurements or – if these were missing, too – with passive sampler data. $HNO_3$ was taken from DELTA measurements, and $NO_x$ was provided by the NPBW with half-hourly time resolution. The difference in measurement height was considered in the calculation of $R_a$. Temperature and relative humidity data





corresponded to the average of measurements from 20 m and 40 m. Since profile measurements of temperature and relative humidity started in April 2016, measurements by the NPBW were used until end of March 2016. Pressure and global radiation were provided by the NPBW. Indicators of stability and turbulence such as Obukhov-Length $L$ and $u_*$ were taken from momentum flux measurements of the sonic anemometer. All micrometeorological and turbulent flux data were aggregated half-hourly. For determining compensation points and additional deposition corrections, $SO_2$ and $NH_3$ concentrations collected by DELTA

samplers were used. Passive sampler measurements were used to replace missing or low-quality $NH_3$ measurements in DELTA time series, and gaps in the $SO_2$ data were filled by the long-term average. Leaf area index (LAI) was modeled as described by van Zanten et al. (2010). For modeling $R_a$ the solar zenith angle, which is calculated by using celestial-mechanic equations, the roughness length $z_0$ and displacement height $d$ are needed. By using the same height as proposed by LOTOS-EUROS for $z_0$ (2.0 m), fluxes were slightly underestimated. However, influence on the dry deposition budget was negligible. Thus, we set

$z_0$ to 2.0 m and $d$ to 12.933 m for coniferous forest and to 11.60 m for deciduous forest. Shifting $z_0$ or $d$ by $\pm50\%$ caused a change of +5.0%/-3.2% and +5.6%/-9.1%, respectively, in the dry deposition after 2.5 years. An incorrect assessment of the LAI by $\pm50\%$ has significant influence on the dry deposition. It leads to a change of +18.9%/-27.7%. The calculation of the dry deposition was done for $NH_3$, NO, $NO_2$, and $HNO_3$ with the mentioned parameters on half-hourly basis. Fluxes of DEPAC-1D were weighted after the actual land-use classes (81.1% coniferous forest and 18.9% deciduous forest). The LAI,

which is based on the LOTOS-EUROS land-use weighting, ranges between 1.9 and 2.8 while considering only deciduous and coniferous forest land-use classes in the flux footprint. The LAI based on the actual land-use weighting ranges between 4.1 and 4.8. Including grassland in the determination of LAI is less useful since characteristics, for example an increase in LAI from the beginning of year, is not representative for the vegetation within the flux footprint. Thus, modeled nitrogen budgets of LOTOS-EUROS should be seen as lower and upper estimates.

After post-processing of TRANC data, we applied two gap-filling strategies. In the first one, DEPAC-1D was used for replacing all missing values in flux data. The second one used MDV for filling gaps up to five days and DEPAC-1D for longer gaps. For comparing the methods with each other we developed a validation strategy: After filling the gaps in the TRANC time series with DEPAC-1D, we used LOTOS-EUROS with the corrected weighting of land-use classes for closing remaining gaps in DEPAC-1D results as well as in TRANC data ensuring a comparison for every time step. Gaps in DEPAC-1D are mostly

related to power outages causing gaps micrometeorological data. Since DEPAC-1D did not include deposition of particles and the actual land-use class in the grid cell did not agree with the land-use class used in LOTOS-EUROS, recalculations of LOTOS-EUROS with a corrected land-use class and/or without considering particulate deposition were performed. Averaged flux time series of LOTOS-EUROS, DEPAC-1D, and TRANC were compared to look for seasonal deviations throughout the observation period. Finally, the annual dry deposition sums of LOTOS-EUROS, DEPAC-1D, TRANC, and CBT were

evaluated.



## 3 Results

### 3.1 Concentrations and fluxes during the measurement campaign

Figure 1 shows ambient concentrations of $\Sigma N_r$ (black), $NH_3$ (red) and $NO_x$ (blue) as half-hourly averages for the entire measurement campaign. Data gaps are mostly related to instrumental performance problems. No $\Sigma N_r$ measurements were
possible until end of May 2016 due to heating problems of the TRANC.

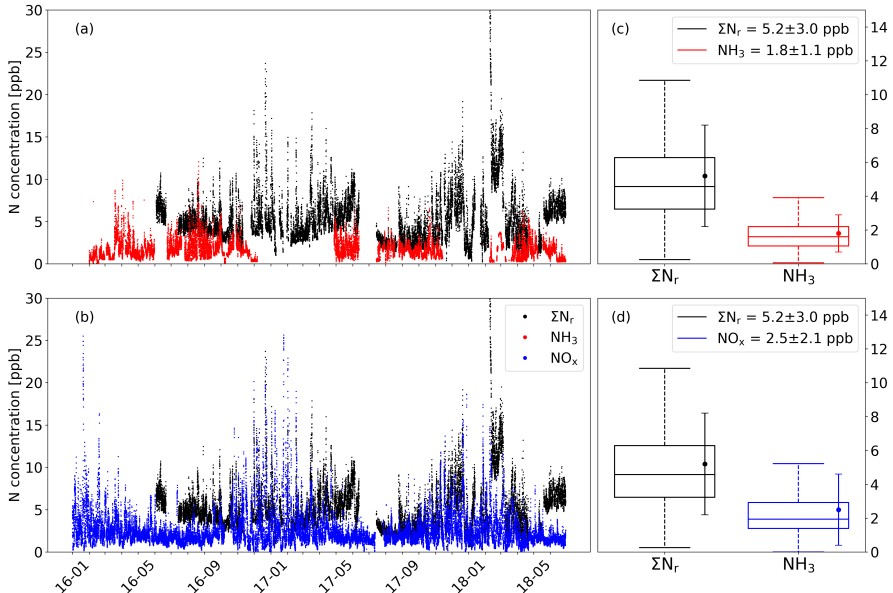

**Figure 1.** Half-hourly averaged concentrations of $\Sigma N_r$ (black), $NH_3$ (red) and $NO_x$ (blue) in ppb from 1 January 2016 to 30 June 2018 displayed in (a) and (b). Box plots (box frame = 25 % to 75 % interquartile range (IQR), bold line = median, whisker = 1.5· IQR) with average values (dots) shown in (c) and (d). Error bars represent one standard deviation.

$\Sigma N_r$ concentrations exhibit highest values during winter months. For example, values were higher than 20 ppb during January 2017 and February 2018. $NO_x$ shows a relatively high concentration level during winter, too. During spring and summer $NO_x$ values are mostly lower than 5 ppb and hence, their contribution to $\Sigma N_r$ decreases. However, $\Sigma N_r$ values remain around 5 ppb and reach values up to 10 ppb, which is related to higher $NH_3$ concentrations during these periods. $\Sigma N_r$ concentration is
5.2 ppb on average, $NH_3$ is approximately 1.8 ppb, and $NO_x$ is 2.5 ppb on average. Values are in agreement with concentrations reported by Beudert and Breit (2010). The elevated $NO_x$ concentration level also affects its contribution to $\Sigma N_r$ measured by the TRANC. Figure B1 shows the contribution of $N_r$ species, which are converted inside the TRANC, to $\Sigma N_r$ as pie charts. Contributions from $NO_3$, $NH_3$, $NH_4$, and $HNO_3$ are determined from monthly DELTA measurements. $NO_x$ concentrations are averaged to the exposition periods of the DELTA samplers. The $\Sigma N_r$ concentration measurements are dominated by $NO_x$. On
average, $NO_x$ contributes with 51.6% to $\Sigma N_r$. At lowest and highest $\Sigma N_r$ concentrations, its influence on $\Sigma N_r$ differs only



slightly. $NH_3$ exhibits a contribution of 21.6% on average, which is lower than the sum of $HNO_3$, $NH_4$, and $NO_3$ ($\sim 26.8\%$). Compared to $NO_x$, $NH_3$ varies significantly from lowest to highest $\Sigma N_r$ concentrations. At the lowest average $\Sigma N_r$ concentration, the contribution of $NH_3$ is significantly high whereas the contribution of $NH_3$ gets negligible compared to the contribution of particulate and acidic $N_r$ compounds ($\sim 35.5\%$) at the highest average $\Sigma N_r$ concentration.

Figure 2 shows non-gapfilled $\Sigma N_r$ fluxes depicted as box plots on monthly time scale. The convention is as follows: Negative fluxes represent deposition, positive fluxes emission. Quality screening and post-processing was done after the criteria mentioned in Sec 2.3.

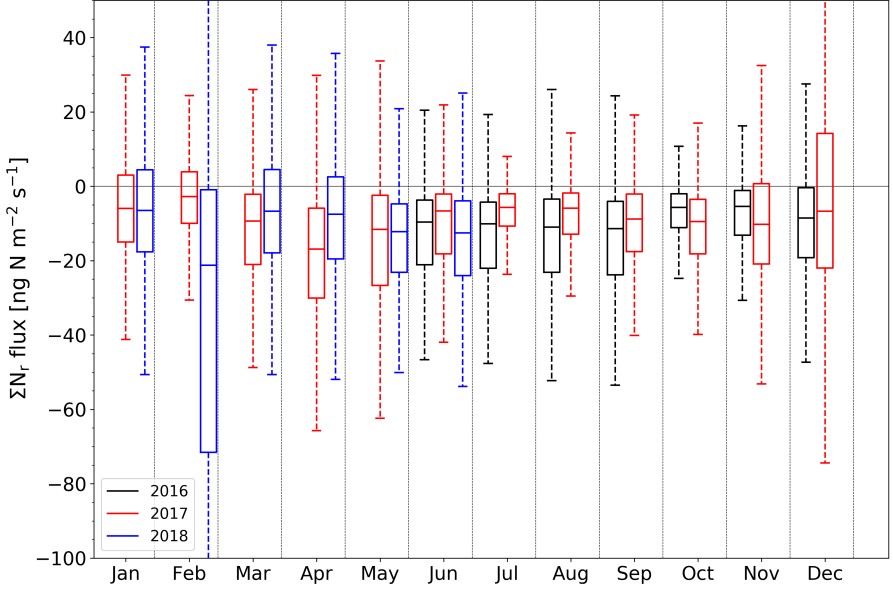

**Figure 2.** Time series of measured high-quality (flags "0" and "1") $\Sigma N_r$ fluxes depicted as box plots on monthly basis (box frame = 25% to 75% interquartile ranges (IQR), bold line = median, whisker = $1.5\cdot$ IQR) in ng N m$^{-2}$s$^{-1}$. Colors indicate different years. The displayed range was restricted from -100 to 50 ng N m$^{-2}$s$^{-1}$.

Almost all $\Sigma N_r$ flux medians are between -15 and -5 ng N m$^{-2}$s$^{-1}$ indicating that mainly deposition of $\Sigma N_r$ occurred at our measurement site. Quality assured half-hourly fluxes showed 85% deposition and 15% emission fluxes. On half-hourly basis,

fluxes are in the range from -409 to 216 ng N m$^{-2}$s$^{-1}$. The mean flux error of non-gapfilled, half-hourly fluxes is 5.7 ng N m$^{-2}$s$^{-1}$ after Finkelstein and Sims (2001). The flux detection limit is calculated by multiplying 1.96 with the flux error (95% confidence limit) (see Langford et al., 2015). The latter is 11.3 ng N m$^{-2}$s$^{-1}$. Both values refer to the entire measurement campaign. Similar values were found by Zöll et al. (2019).

In general, median deposition is almost on the same level for the entire campaign with slight seasonal differences. For

instance, median deposition is slightly higher during spring and summer than during winter for 2016. However, median deposition during winter 2017 is similar to median deposition in summer 2017. Median deposition was significantly stronger from June 2016 till September 2016 than for the same period in 2017. IQR and whisker cover a wider range, too. The pattern





changes for the time period from October to December. In December 2017, the IQR expands in the positive range indicating emission events for a significant time period. The largest median deposition with 25 ng N m$^{-2}$s$^{-1}$ and the widest range in IQR

reaching approximately -70 ng N m$^{-2}$s$^{-1}$ were registered in February 2018 indicating strong deposition phases during that month with sporadic emission events. Such phenomenons were not observed in the years before. In the following month, the deposition is slightly higher from March to April 2017 than for the same period in 2018. Fig. 3 shows averaged daily cycles for every month.

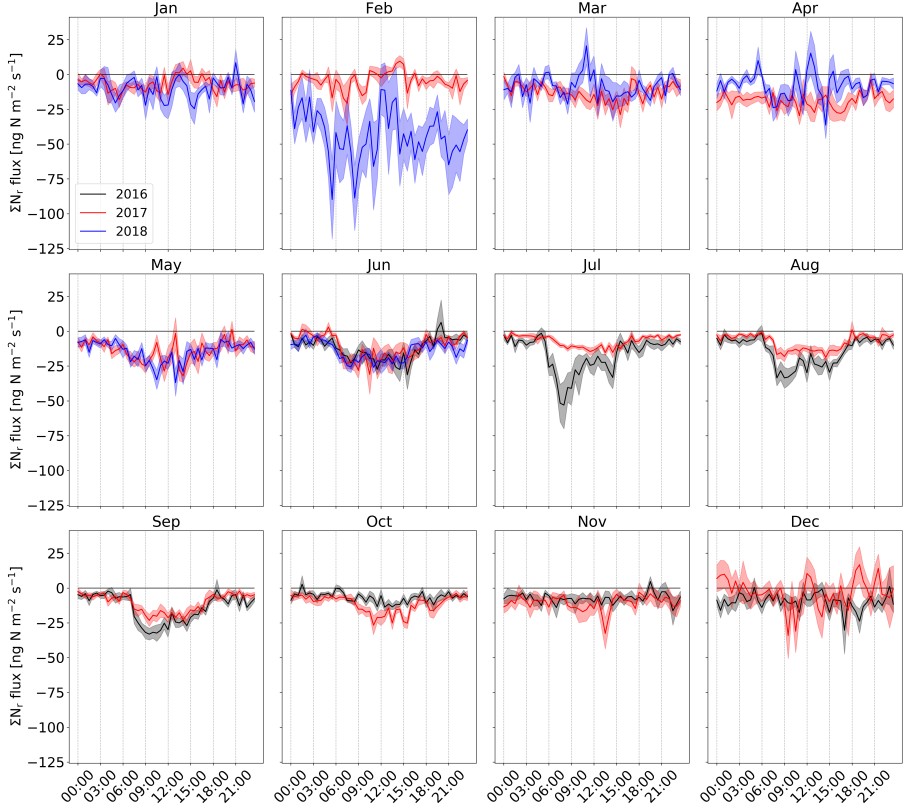

**Figure 3.** Mean daily cycle for every month of $\Sigma N_r$ fluxes from June 2016 to June 2018 on half-hourly basis. The shaded area represents the standard error of the mean. Colors indicate different years.

In general, the $\Sigma N_r$ daily cycle exhibits low deposition or neutral exchange during nighttime/evening and increasing deposi-

tion during daytime. Deposition rates are similar during the night for the entire campaign except for February 2018. Maximum deposition is reached between 9:00 and 15:00 CET. Deposition is enhanced from May until September showing fluxes between -40 and -20 ng N m$^{-2}$s$^{-1}$. During autumn (October-November) and winter (December-February), the daily cycle weakens with almost neutral or slightly negative fluxes, mostly lower than -10 ng N m$^{-2}$s$^{-1}$. The daily cycles of the respective same months are mainly similar. However, during certain months, which differ in their micrometeorology and/or in the composition of $\Sigma N_r$,

differences can be significant. For example, the daily cycle of March and April 2017 is clearly different to daily cycle of





March and April 2018. During spring 2017, slight deposition fluxes are found whereas the $\Sigma N_r$ exchange is close to neutral a year later. The median deposition is also slightly larger in March and April 2017 than in the year after (Fig. 2). In December 2017, the daily cycle is close to the zero line and positive fluxes were observed, although standard errors are relatively large ($\pm$ 10.5 ng N m$^{-2}$s$^{-1}$ on average). In December 2016, slight deposition fluxes are observed for the entire daily cycle. The
daily cycle of February 2018 shows high deposition values during the entire day, the highest values during the measurement campaign. Again, average standard error is relatively large ($\pm$ 17.9 ng N m$^{-2}$s$^{-1}$) for February 2018 compared to February 2017.

## 3.2 Controlling factors of measured $\Sigma N_r$ fluxes

Fig. 3 reveals that the pattern of $\Sigma N_r$ daily cycle is characterized by lower deposition during the night and highest values
around noon. The deposition is enhanced from May until September compared to the rest of the year. Micrometeorological parameters such as temperature (Wolff et al., 2010), humidity (Wyers and Erisman, 1998; Milford et al., 2001), concentrations (Brümmer et al., 2013; Zöll et al., 2016), and dry/wet leaf surfaces (Wyers and Erisman, 1998; Wentworth et al., 2016) were reported to control the deposition of $N_r$ compounds. Therefore, we investigate the dependency of $\Sigma N_r$ fluxes on temperature, humidity, dry/wet leaf surface, and $\Sigma N_r$ concentration. We separate half-hourly fluxes into classes of low and high temperature,
humidity, and concentration. The threshold values, which are calculated from May to September, based on the median of the mentioned parameters. Leaf wetness value is calculated after the scheme described in Sec. 2.2 for same time period. No significant influence of the different installation height on leaf surface wetness was found.

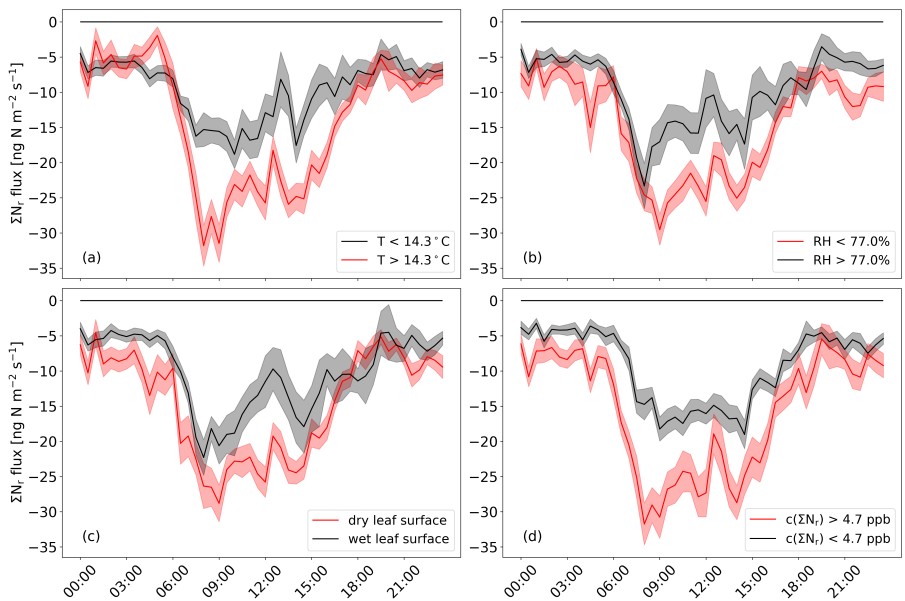

**Figure 4.** Mean daily cycle from May to September of $\Sigma N_r$ fluxes for low and high temperature, humidity, and concentration. Median values of temperature, humidity, and concentration, which are derived for the same time period, are used as threshold values for separating fluxes. For separating dry and wet leaf surfaces, the scheme proposed in Sec. 2.2 is applied. The shaded area represents the standard error of the mean.

In general, higher temperatures, less humidity, higher concentrations, and dry leaf surfaces favour deposition of $\Sigma N_r$. Temperature seem to affect $\Sigma N_r$ fluxes from 6:00 to 18:00 CET stronger leading to differences of more than -10 ng N m$^{-2}$s$^{-1}$,

for instance around 9:00 and 15:00 CET. During dawn/nighttime fluxes show no significant temperature dependence. Concentration has the strongest impact on the deposition. The effect is increased from 6:00 to 15:00 CET exhibiting a difference -5.5 ng N m$^{-2}$s$^{-1}$ on average, but also nighttime deposition fluxes are enhanced at higher concentrations. The impact of less humidity and dry leaves is slightly lower than concentration and temperature, but they affect nighttime deposition stronger than temperature. Finally, it should be mentioned that the shapes of the daily cycles for each parameter shown in Fig. 4 are similar

for both threshold values and differ only in amplitude. It indicates that other drivers may influence the pattern of $\Sigma N_r$ fluxes stronger than the shown parameters here.

### 3.3 Cumulative N exchange and method comparison

For determining the $\Sigma N_r$ dry deposition, gaps were filled in flux time series with DEPAC-1D and MDV (see Sec. 2.4.3). Fluxes estimated through the EC technique covered 47.8% of the measurement period after quality filtering. The low amount of valid

flux measurements was expected, for example, related to insufficient turbulence during nighttime, performance issues of the instruments, etc. Applying MDV allows to increase the coverage to 65.0%. With DEPAC-1D alone nearly all gaps were closed. Remaining gaps in DEPAC-1D were about 4% due to power failures and were filled with LOTOS-EUROS results. Afterwards,





fluxes were added up to get a cumulative sum. In the following, the results of the method comparison described in Sec. 2.4 are presented. Figure 5 shows the cumulative $\Sigma N_r$ dry deposition of the different methods for the duration of the campaign.

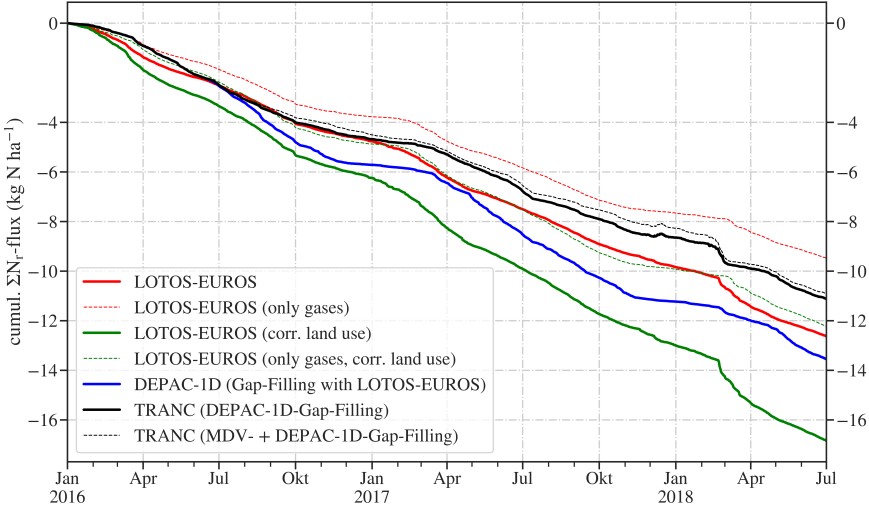

**Figure 5.** Comparison of measured and modeled cumulative $\Sigma N_r$ dry deposition after gap-filling for the entire measurement campaign. Colors indicate different methods: TRANC+DEPAC-1D (black, solid), TRANC+MDV+DEPAC-1D (black, dashed), DEPAC-1D+LOTOS-EUROS (blue), LOTOS-EUROS with corrected land use (green, solid), LOTOS-EUROS with corrected land use, but only gases (green, dashed), LOTOS-EUROS (red, thick), and LOTOS-EUROS with corrected land use, but only gases (red, dashed)

The $\Sigma N_r$ dry deposition values estimated by each method for 2.5 years are listed in Table 1.





**Table 1.** $\Sigma N_r$ dry deposition of TRANC, DEPAC-1D, LOTOS-EUROS, and CBT for the entire measurement campaign, i.e. January 2016 to June 2018. Annual dry deposition of 2018 is extrapolated for TRANC, DEPAC-1D, and LOTOS-EUROS. CBT lower and upper estimates were weighted according to the measured land use. For a visualisation of annual dry deposition see Fig. 6.

| Data set | Gap-filling strategy | $\Sigma N_r$ dry deposition [kg N ha$^{-1}$] | | | |
|---|---|---|---|---|---|
| | | after 2.5 years | 2016 | 2017 | 2018 |
| TRANC | MDV+DEPAC-1D | 10.9 | 4.68 | 3.97 | 5.0 |
| TRANC | DEPAC-1D | 11.1 | 4.50 | 3.78 | 5.34 |
| DEPAC-1D | LOTOS-EUROS | 13.6 | 5.71 | 5.51 | 4.69 |
| LOTOS-EUROS *only gases* | - | 9.5 | - | - | - |
| LOTOS-EUROS | - | 12.6 | 4.76 | 5.07 | 5.63 |
| LOTOS-EUROS *with corrected land use and only gases* | - | 12.2 | - | - | - |
| LOTOS-EUROS *with corrected land use* | - | 16.8 | 6.24 | 6.75 | 7.76 |
| CBT lower estimate | - | 13.7 | 3.30 | 4.35 | 6.09 |
| CBT upper estimate | - | 22.6 | 6.44 | 6.98 | 9.14 |

Overall, DEPAC-1D and and LOTOS-EUROS seem to overestimate $\Sigma N_r$ dry deposition compared to our measurements, in particular LOTOS-EUROS with the corrected land use. The dry deposited $\Sigma N_r$ modeled by DEPAC-1D consists of 76% $NH_3$, 13% $HNO_3$, 11% $NO_2$, and less than 1% NO. It shows that modeled deposition of DEPAC-1D is mostly driven by $NH_3$. $HNO_3$ and $NH_3$ deposition velocities are nearly equal (1.81 cm s$^{-1}$ and 1.86 cm s$^{-1}$). Also, emission phases are modeled for
$NH_3$ due to the low compensation point indicated by the negative whisker of the box plot (Fig.C1.). However, their influence on total deposition is negligible since only short emission phases of $NH_3$ were modeled. Deposition velocity for $NO_2$ and NO are relatively low. 0.08 cm s$^{-1}$ is determined for $NO_2$ and 0.0 cm s$^{-1}$ for NO.

  $\Sigma N_r$ exchange of DEPAC-1D is rather neutral during the entire winter, and thus the difference to measured deposition is close to zero. During summer a systematic overestimation of DEPAC-1D to measured fluxes is observed. Modeled deposition
by LOTOS-EUROS is slightly lower than DEPAC-1D during summer and consequentially closer to measured fluxes. However, during autumn and spring predicted deposition by LOTOS-EUROS is significantly higher than deposition determined by DEPAC-1D and TRANC. The agreement of the measured, non gap-filled $\Sigma N_r$ fluxes (results not shown) with LOTOS-EUROS for the same half-hours without particulate input is conspicuous after 2.5 years. TRANC measurements show a cumulative, non gap-filled dry deposition of 4.7 kg N ha$^{-1}$, LOTOS-EUROS exhibits 4.5 kg N ha$^{-1}$. This agreement has to be regarded





with caution since the TRANC also converts particulate $\Sigma N_r$ compounds and the land-use class weighting of LOTOS-EUROS is not valid for the measurement site. Correcting the land-use class based on actual vegetation of the flux footprint, exhibit a significant overestimation of the dry deposition. We determined $8.2\,\mathrm{kg\ N\ ha^{-1}}$ with LOTOS-EUROS for measured, non gap-filled half-hours including particulate deposition and the actual land-use class weighting, and $16.8\,\mathrm{kg\ N\ ha^{-1}}$ is calculated for the entire measurement campaign. The applied gap-filling strategies result in similar dry deposition after 2.5 years (Table 1).

The difference between both curves is enhanced from July 2017 to mid February 2018. Due to the strong deposition occurring in late February 2018, the difference between the curves is significantly reduced. Obviously, DEPAC-1D could not model the deposition event accurately.

Since all cumulative curves exhibit generally the same shape, we conclude that the variability in fluxes is reproduced by DEPAC-1D and LOTOS-EUROS well, although the amplitude and duration of certain deposition events is different. This

observation is valid for the strong deposition event in late February 2018 observed by the TRANC, but it is treated differently by DEPAC-1D and LOTOS-EUROS. As stated before, it is not accurately modeled by DEPAC-1D and also not by LOTOS-EUROS without considering particle deposition. Including particle deposition in LOTOS-EUROS leads to better agreement with TRANC measurements for a few weeks. It seems that the deposition during late February 2018 is most likely driven by particulate $N_r$ compounds. Such compounds are not implemented in DEPAC-1D. After the deposition event, measured $\Sigma N_r$

exchange is almost neutral whereas modeled deposition of LOTOS-EUROS increases resulting in significant disagreement in $\Sigma N_r$ deposition. However, the emission event, which is calculated from TRANC measurements for December 2017, is not captured by LOTOS-EUROS and DEPAC-1D.

In the following, a comparison of the $\Sigma N_r$ dry deposition separated by method and measurement years is given in Fig. 6. The dry deposition values for 2018 are extrapolated. The extrapolation is kept simple. We extrapolated the deposition of the

first half of 2018 until the end of the year.





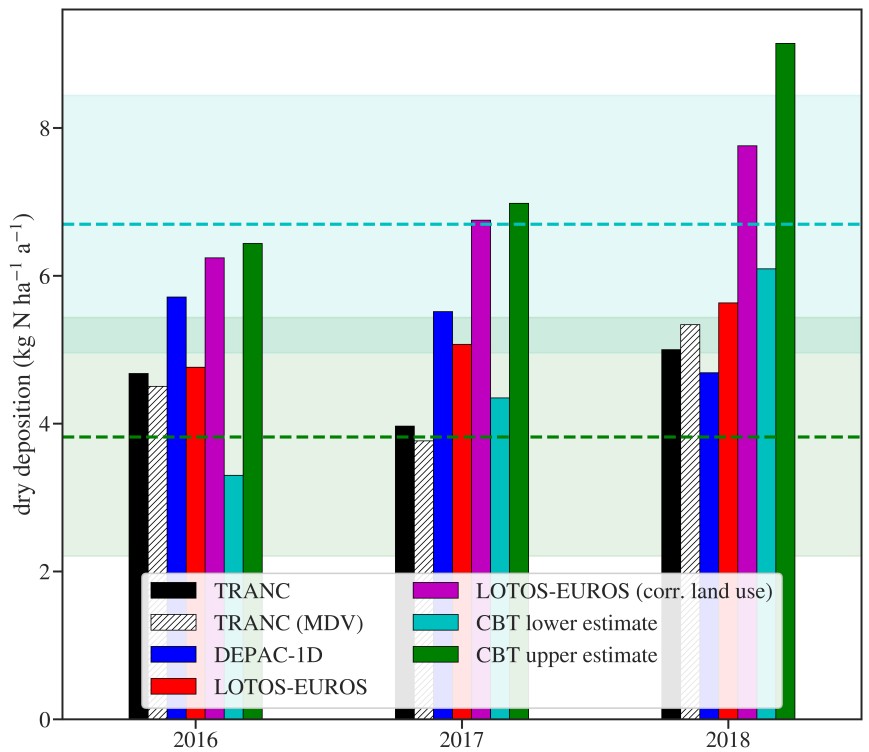

**Figure 6.** $\Sigma N_r$ dry deposition for the years 2016, 2017, and 2018 displayed as bar chart. Colors indicate different methods: TRANC+DEPAC-1D (black), TRANC+MDV (shaded),DEPAC-1D (blue), LOTOS-EUROS (red), LOTOS-EUROS with corrected land use (purple), and canopy budget technique (turquoise and green). Data from TRANC, DEPAC-1D, and LOTOS-EUROS are extrapolated for 2018. CBT lower and upper estimates were weighted according to the measured land use. The colored dashed lines indicate the averaged dry deposition of the lower and upper estimates from 2010 to 2018, the shaded areas represent their standard deviation.

Annual dry deposition of the TRANC ranges from 3.8 kg N ha$^{-1}$ a$^{-1}$ to 5.3 kg N ha$^{-1}$ a$^{-1}$. 4.7 kg N ha$^{-1}$ a$^{-1}$ to 5.7 kg N ha$^{-1}$ a$^{-1}$ is modeled by DEPAC-1D, 4.8 kg N ha$^{-1}$ a$^{-1}$ to 5.6 kg N ha$^{-1}$ a$^{-1}$ is predicted by LOTOS-EUROS with uncorrected land use, and 6.2 to 7.8 kg N ha$^{-1}$ a$^{-1}$ by LOTOS-EUROS with corrected land use. Annual dry deposition estimated by CBT are similar for 2016 and 2017. Values are close to the long-term average estimated by CBT for 2010 until 2018 ($\sim$ 3.8 kg N ha$^{-1}$ a$^{-1}$ as lower estimate and $\sim$ 6.7 kg N ha$^{-1}$ a$^{-1}$ as upper estimate). For 2018 the application of CBT results in a significantly higher lower and upper estimates (6.1 and 9.1 kg N ha$^{-1}$ a$^{-1}$). Therewith, CBT estimates for 2018 are outside the range of one standard deviation of the long-term average.

Averaged annual $\Sigma N_r$ dry deposition is 4.5 kg N ha$^{-1}$ a$^{-1}$ for both gap-filling approaches, DEPAC-1D shows 5.3 kg N ha$^{-1}$ a$^{-1}$, and LOTOS-EUROS predicts 5.2 kg N ha$^{-1}$ a$^{-1}$ to 6.9 kg N ha$^{-1}$ a$^{-1}$ depending on the weighting of land-use classes. 7.5 kg N ha$^{-1}$ a$^{-1}$ is estimated with CBT for the period from 2016 to 2018 as upper estimate. 4.6 kg N ha$^{-1}$ a$^{-1}$ are determined as lower estimate. It shows that dry depostion estimated by TRANC, DEPAC-1D, and LOTOS-EUROS is within





the frame of minimum and maximum deposition estimated by CBT but generally closer to the lower estimate of CBT except for LOTOS-EUROS with the corrected land use weighting.

Annual dry deposition of LOTOS-EUROS and CBT is higher in 2017 than in 2016, whereas TRANC and DEPAC-1D exhibit less deposition in 2017. Values of TRANC with and without MDV are almost similar for 2016 and 2017. Using only DEPAC-1D as gap-filling strategy results in slightly higher dry deposition for 2016 and 2017. For 2018 using MDV leads to higher deposition since DEPAC-1D predicts the lowest deposition compared to years before. The difference for 2018 is caused by the deposition event in February 2018, which has an influence on the MDV method leading to significantly larger deposition fluxes. The high deposition values of 2018 modeled by LOTOS-EUROS are probably related to the generally higher modeled concentrations in the first half of 2018.

### 3.4 Sensitivity of measured vs. modeled input parameters to deposition estimates

As stated before, LOTOS-EUROS exhibits relatively high deposition values. Running LOTOS-EUROS with the corrected land-use class, leads to the highest dry deposition values for all years, without considering canopy budgets technique. For a closer investigation of this issue we conduct a comparison of model input parameters such as temperature, relative humidity, $NH_3$ concentration, global radiation, and friction velocity to measured data and evaluate their impact on $NH_3$ fluxes modeled by DEPAC within LOTOS-EUROS. These parameters hold an important role in the modeling of the $NH_3$ exchange (e.g., Nemitz et al., 2001). Air temperature controls the influence of the emission potential, the apoplastic concentration ratio, at surfaces on the $NH_3$ compensation point (Sutton et al., 1994; Nemitz et al., 2000). Relative humidity is used as approximation for the canopy humidity and controls the cuticular deposition (Sutton et al., 1994). $NH_3$ concentration is proportional to the $NH_3$ flux (van Zanten et al., 2010), global radiation enhances the opening width of the stomata (Wesely, 1989), and friction velocity is a measure of the turbulence and has an influence on the aerodynamic and quasi-laminar resistance (Webb, 1970; Paulson, 1970; Garland, 1977; Jensen and Hummelshøj, 1995, 1997). $NH_3$ was chosen since it is the most abundant compound in modeled $\Sigma N_r$ (see Fig. E1), and resistance models are most developed for $NH_3$. Fig. 7 illustrates the results of the sensitivity study.





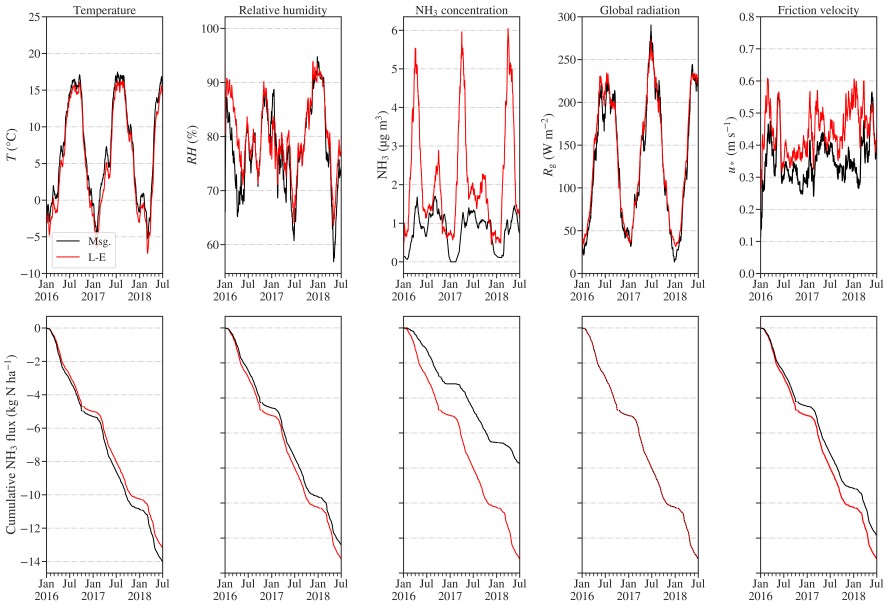

**Figure 7.** Comparison of modeled (red) and measured (black) input data and their impact on cumulative $NH_3$ deposition predicted by DEPAC-1D for land-use class spruce forest. The comparison is carried out for air temperature ($T$), relative humidity ($RH$), $NH_3$ concentration, global radiation ($R_g$), and friction velocity ($u_*$). A 30 day running average is applied to the input data for better visibility. Modeled input data are the same as used for the LOTOS-EUROS calculations.

Overall, the agreement of measured and modeled input data is excellent for temperature and global radiation. Values of $r^2$

are 0.78 for global radiation and 0.97 for temperature. A slight difference is visible for relative humidity in the first half of 2016 with $r^2$ being 0.67. In case of relative humidity, using locally measured values leads to a reduction in deposition by 6%. The deposition increases by approximately 6% if measured temperature values are used. The impact on deposition using measured global radiation is negligible. $u_*$ of LOTOS-EUROS is systemically higher, and the seasonal pattern is different to values determined from the sonic anemometer. Thus, $r^2$ is only 0.43 but using measured values for $u_*$ leads only to 10% less deposition.

The difference between measured and modeled $NH_3$ is most pronounced. Modeled concentrations are approximately 2 to 3 times larger in spring and autumn. Furthermore, the seasonal pattern of the measured $NH_3$ disagrees with the modeled values. Using measured $NH_3$ concentration reduces the deposition by approximately 42% compared to the modeled deposition. Consequentially, $NH_3$ concentration is most responsible for the discrepancy of modeled and measured $\Sigma N_r$ fluxes. The generally high $NH_3$ concentration also influences its contribution to $\Sigma N_r$ concentration modeled by LOTOS-EUROS. Figure E1 shows

the contribution of the $N_r$ species to modeled $\Sigma N_r$ as pie charts. LOTOS-EUROS states out $NH_3$ as the main contributor. $NO_x$, which is identified as main contributor to $\Sigma N_r$ from measurements, takes only 22.2% of the modeled $\Sigma N_r$. At highest $\Sigma N_r$ concentration, $NH_3$ corresponds to almost half of the $\Sigma N_r$. Particulate and acidic $N_r$ compounds have a higher contribution than $NO_x$ on average ($\sim 41.7\%$). Their contribution is also higher than values extracted from DELTA measurements, but decreases



from lowest to highest $\Sigma N_r$ concentration. $HNO_3$ gets even negligible for the highest $\Sigma N_r$ concentration. On average, $NH_3$
and $NO_x$ account for 58.2% of $\Sigma N_r$ concentrations whereas DELTA measurements show 73.2% for both gases on average.

## 4 Discussion

### 4.1 Interpretation of measured concentrations and fluxes

Measured half-hourly $\Sigma N_r$ concentrations are very low in comparison to other sites. On average, we measured 5.2 ppb $\Sigma N_r$,
1.8 ppb $NH_3$, and 2.5 $NO_x$. Wintjen et al. (2020) determined an average $\Sigma N_r$ concentration level of 21 ppb for a seminatural
peatland, Brümmer et al. (2013) measured between 7 and 23 ppb as monthly average above a cropland site, and Ammann
et al. (2012) measured half-hourly $\Sigma N_r$ concentrations ranging from less than 1 ppb to 350 ppb for grassland site. Only for
certain time periods, $\Sigma N_r$ concentrations reached significantly higher values. During winter $NO_x$ increased due to emission
from heating with fossil fuels and from combustion processes, for example through traffic and power plants. A generally
lower mixing height, which is often observed during winter, also leads to a higher concentration of air pollutants. In spring and
autumn higher $\Sigma N_r$ concentrations can be attributed to $NH_3$ emission from the application of fertilizer and livestock farming in
the surrounding environment (Beudert and Breit, 2010). $NH_3$ emissions from livestock farming in rural districts, which belong
to the NPBW, are approximately half of the emissions compared to rural districts located in the Donau-Inn valley (Beudert and
Breit, 2010), who measured concentrations of $NO_2$ (1.9-4.4 ppb), NO (0.4-1.5 ppb) and $NH_3$ (1.3 ppb) at the same site. The low
concentration level and seasonal variability of $\Sigma N_r$ compounds, in particular $NH_3$ and $NO_2$, are in agreement with Beudert
and Breit (2010). Values are expectable for a site, which is some kilometers away from anthropogenic emission sources.
Studies like Wyers and Erisman (1998); Horii et al. (2006); Wolff et al. (2010); Geddes and Murphy (2014) dealing with
different $\Sigma N_r$ compounds, which were conducted for different time periods of the year, confirm the seasonal pattern of $\Sigma N_r$.
Obviously, measured concentration levels were significantly higher since the observed ecosystems were subject of agricultural
management or in close proximity to industrial or agricultural emissions. In general, a comparison of $\Sigma N_r$ concentrations and
fluxes to other studies is difficult due to the measurement of the total nitrogen. Most studies, which have been published so far,
focused only on a single or a few compounds of $\Sigma N_r$ and are limited to selected sites and time periods of a few days or months.
Only a few studies had been focusing on $\Sigma N_r$.

   Brümmer et al. (2013) measured $\Sigma N_r$ exchange above agricultural land. During unmanaged phases fluxes were between
-20 ng N m$^{-2}$ s$^{-1}$ and 20 ng N m$^{-2}$ s$^{-1}$. Apart from managing events, fluxes of the arable field site were closer to neutral
conditions compared to our unmanaged forest site, which is mainly characterized by deposition fluxes and is therefore a larger
sink for reactive nitrogen. Ammann et al. (2012) measured $\Sigma N_r$ fluxes above managed grassland. In the growing season mostly
deposition fluxes up -40 ng N m$^{-2}$ s$^{-1}$ were measured. The authors reported slightly increased deposition due to weak NO
emission during that phase. Similar to Brümmer et al. (2013), their flux pattern is influenced by fertilizer application and
thus, varying contributions of $N_r$ compounds, for instance by bidirectionally exchanged $NH_3$ leading to both net emission
and deposition phases of $\Sigma N_r$. A few studies measured $N_r$ compounds above (mixed) forests. Hansen et al. (2015) measured
$NH_3$ fluxes between -60 and 120 ng N m$^{-2}$ s$^{-1}$ above a deciduous forest. Due to the selected measuring time of the year,





emission of $NH_3$ through fallen leaves had an influence on measured fluxes leading to probably less deposition during late summer and autumn. High emission fluxes were also measured at our measurement site in December, which could be induced by the decomposition of fallen leaves. Pictures from the 50 m platform showed a substantial snow cover for the whole footprint.

The snow layer acts as an insulator for the soil, prevents soil from frost penetration effectively, and thus protects plants and microorganisms (Bleak, 1970; Vogt et al., 1983; Moore, 1983; Inouye, 2000). Thus, processes, which lead a decomposition of leaves, needles or lichens by microorganisms, can happen under the snow layers with substantial losses, especially for lichens (Taylor and Jones, 1990). The authors further discovered an increase in nitrogen concentration in the investigated samples. Since we observed a slower varying air temperature with temperatures below zero for 2 to 3 days followed short periods of less

than one day with temperatures close to zero degrees and even higher, the accumulation of nitrogen under the snow layer and a immediate release due to freeze-thaw cycles probably happened. The determined order of magnitude by Hansen et al. (2015) is comparable to our flux measurements.

Since we also measured other $N_r$ compounds such as $HNO_3$ and $NO_2$, which exhibit mostly deposition (Horii et al., 2004, 2006), deposition fluxes predominated at our measurement site compared to Hansen et al. (2015). NO is mainly observed as

emission from soil if it is produced through (de)nitrification processes (Butterbach-Bahl et al., 1997; Rosenkranz et al., 2006). The contribution of NO to $\Sigma N_r$ is probably negligible because NO is rapidly converted to $NO_2$ in the presence of $O_3$ within the forest canopy, especially close to the ground (Rummel et al., 2002; Geddes and Murphy, 2014). Therefore, a comparison with chamber measurements, which could had been conducted at the ground for measuring $N_r$ compounds was considered as less useful due to the large footprint of the flux measurements, fast conversion processes within the forest canopy, and uptake

possibilities like leaf surfaces for $N_r$ compounds (e.g., Wyers and Erisman, 1998; Rummel et al., 2002; Sparks et al., 2001; Geddes and Murphy, 2014; Min et al., 2014).

The findings of DELTA measurements revealed that $NO_x$, in particular $NO_2$, is the most abundant compound in $\Sigma N_r$ followed by $NH_3$. Both gases account for 73.2% of $\Sigma N_r$. The values of $NO_x$ and $NH_3$ differ significantly from values proposed by Zöll et al. (2019), in particular $NO_x$. This is related to the different periods, which were considered for averaging. Zöll et al.

(2019) reported values for summertime. In this study, values are influenced by seasonal impacts. It has to be considered that the contribution of $NH_3$ differs with increasing $\Sigma N_r$ concentration whereas the contribution $NO_x$ remain almost similar. At the highest average $\Sigma N_r$ concentration, we determined a substantial contribution of particulate and acidic $N_r$ species, which is higher than the influence of $NH_3$ on $\Sigma N_r$ at that concentration level. Findings of Tang et al. (2020) had shown that $HNO_3$ concentrations measured by DELTA system using carbonate coated denuders may be significantly overestimated (45% on average)

since HONO sticks also at those prepared surfaces. Thus, the $HNO_3$ contributions should be seen as an upper estimate.

Consequentially, other compounds such as $NO_2$ and $HNO_3$ are also important for the interpretation of the $\Sigma N_r$ flux pattern. $NO_2$ deposition and emission fluxes, which depend on the concentration level, were observed during the day by Horii et al. (2004) above a mixed forest, and mostly deposition of $NO_2$ during the night. $NO_2$ exhibits also a bidirectional exchange pattern in natural ecosystems (Horii et al., 2004; Geddes and Murphy, 2014; Min et al., 2014). The diurnal cycle of NO is reversed

to $NO_2$ during the day and is almost neutral with a tendency of slight emission during the night (Horii et al., 2004; Geddes and Murphy, 2014). It has to be taken into account that $NO_2$ is removed from the atmosphere by the reaction with $O_3$. During





the day and night $NO_3$ reacts with $NO_2$ to $N_2O_5$. The latter can react with $H_2O$ to $HNO_3$. $HNO_3$ is an effective removal for $NO_2$ and has a significant impact on the measured deposition flux (Munger et al., 1996). However, Min et al. (2014) stated that peroxy nitrates and akyl nitrates are also responsible for the removal of $NO_x$, apparently more important than $HNO_3$. Horii et al. (2006) measured all oxidized nitrogen species ($NO_y$), which is the sum of NO, $NO_2$, $NO_3$, dinitrogen pentoxide ($N_2O_5$), $HNO_3$ + peroxyacetyl nitrate (PAN), other organic nitrates, and aerosol nitrate such as $NH_4NO_3$, at the same site as Horii et al. (2004). They measured only deposition fluxes for $NO_y$. Fluxes were mostly below -40 ng N m$^{-2}$ s$^{-1}$, but could achieve up to -80 ng N m$^{-2}$ s$^{-1}$. $HNO_3$ fluxes were almost as high as the $NO_y$ fluxes. Munger et al. (1996) did also $NO_y$ flux measurements above the same forest some years earlier and took measurements at less polluted spruce forest. At the latter location, only slight deposition of $NO_y$ occurred. At the former location, results are similar to Horii et al. (2006). It shows that $HNO_3$ seems to have a significant influence on the deposition of $\Sigma N_r$ even at sites exhibiting a low concentration level of $\Sigma N_r$ compounds like $NO_2$.

The observed daily cycle, which exhibits low negative or neutral fluxes during the night, increasing deposition in the morning, and decreasing deposition in the evening, is in agreement with other studies dealing with $\Sigma N_r$ compounds above different forest ecosystems. For example, Wyers and Erisman (1998) measured similar daily cycles of $NH_3$ above a coniferous forest, Munger et al. (1996), Horii et al. (2006), and Geddes and Murphy (2014) reported daily patterns of $NO_y$ above mixed forests, Horii et al. (2006) did similar observations for $HNO_3$, and Wolff et al. (2010) observed higher deposition of total ammonium ($NH_4^+$) and total nitrate $NO_3^-$ fluxes, the aqueous phase of $NH_4NO_3$, above a spruce forest during the day.

Apparently, fluxes measured at our location have high $NO_x$, or, more precisely, a high $NO_2$ fraction, a generally low $NH_3$ fraction, which is higher for low $\Sigma N_r$ fluxes, and considerable fraction of particulate and acidic $N_r$ species, especially for high $\Sigma N_r$ fluxes. In principle, the order of magnitude of the $\Sigma N_r$ flux is similar to values reported in the above-mentioned publications. Even if other $NO_y$ compounds are not the main flux contributors, they change the composition of the $\Sigma N_r$ flux. $NO_y$ compounds have an influence on the $NO$-$NO_2$-$O_3$ cycle and on the reaction pathways of $NH_3$ and $HNO_3$. These are not limited to gas phase reactions (Meixner, 1994), but also gas-particle interactions (Wolff et al., 2010) can occur. Thus, individual measurement devices are needed to measure single $N_r$ species for a precise quantification of the $\Sigma N_r$ flux. Implementing such a setup will be challenging due to high technical requirements of the instruments in case of technical complexity, dimensions, and power consumption. Running such a setup for at least a year should also be considered for a representative data set.

## 4.2 Influence of micrometeorology on deposition and emission

Overall, the shape and maximum deposition of the daily cycles shown in Fig. 3 is mostly driven by global radiation, which acts as primary driver for the $\Sigma N_r$ exchange, recently verified by an artificial neural network approach conducted by Zöll et al. (2019). The authors identified $\Sigma N_r$ concentration as secondary driver for $\Sigma N_r$ deposition. The influence of concentration on $\Sigma N_r$ fluxes and its compounds had been reported in several studies (e.g., Brümmer et al., 2013; Zöll et al., 2016). Also, micrometeorological parameters such as relative humidity and temperature favor the exchange of $\Sigma N_r$ compounds (Wyers and Erisman, 1998; Milford et al., 2001; Wolff et al., 2010; Wentworth et al., 2016). Global radiation was not identified as primary controlling factor for $NH_3$ by Milford et al. (2001). They found that $NH_3$ exchange was mostly driven by canopy



temperature, canopy wetness, and ambient concentrations. Thus, global radiation favoring the exchange through the stomatal pathway appears to be an important controlling factor under low $NH_3$ concentrations.

The higher deposition in March and April 2017 (Fig. 2) compared to spring 2018 could be related to enhanced photosynthetic activity in spring 2017. Average temperature was approximately 5°C in March 2017, and during March 2018 average

temperature was only 0.3°C. Mid of April 2018, an immediate increase of temperature was observed leading to temperatures comparable to April 2017. Consequentially, the wider opening of the stomata was most likely shifted to mid or end of April 2018, which is confirmed by the similar shape of the daily cycle for May 2017 and 2018. Also, $\Sigma N_r$ concentration was approximately 3.3 ppb on average for April 2018 and approximately 6.3 ppb a year before. Higher concentration level probably induced by agricultural management in the surrounding region likely favoured N deposition, too. Deviations in deposition dur-

ing summer 2016 and 2017, especially from July to September, were probably related to different $\Sigma N_r$ concentration levels. $\Sigma N_r$ concentration was 4.7 ppb for summer 2016 and only 2.8 ppb on average for summer 2017. Standard deviations were almost similar at $1.96\sigma$ level demonstrating comparable variability in concentrations. Almost the same average and pattern were investigated for humidity and temperature in July and August. It seems that an enhanced concentration level of $\Sigma N_r$ compounds were most responsible for discrepancies in the observed fluxes confirming results of Zöll et al. (2019), who identified

$\Sigma N_r$ concentration as an important driver for $\Sigma N_r$ exchange at the same site.

Therefore, an in-depth investigation of relative humidity, temperature, leaf surface wetness, and concentration was conducted. The analysis of Fig.4 has shown that dry conditions, induced by higher temperatures and low relative humidity, favour $\Sigma N_r$ deposition. Higher concentrations values lead to higher deposition values through the entire daily cycle. The impact of increasing concentration on nitrogen (deposition) fluxes is well documented, for example, by Ammann et al. (2012) and Brüm-

mer et al. (2013) for $\Sigma N_r$, by Horii et al. (2006) for $NO_y$, Horii et al. (2004) for $NO_x$, and by Zöll et al. (2016) for $NH_3$. The effect of temperature on the $\Sigma N_r$ fluxes is most pronounced during daytime. Higher temperatures increase the opening size of the stomata leading to increased photosynthetic activity. Wolff et al. (2010) observed higher deposition for total ammonium and total nitrate under dry conditions, which correspond to temperatures higher than 15°C and relative humidity below 70%. During foggy or rainy conditions, deposition was close to neutral or even emission occurred. Their ranges and corresponding

limits for temperature and humidity are comparable to the values examined at our site. However, Wyers and Erisman (1998) reveal that $NH_3$ deposition is maximized if canopy exhibits a high canopy water storage level ($> 2\,\mathrm{mm}$). They found that leaf surfaces could act as a sink and as a source of $NH_3$. An elevated relative humidity level increase the thickness of the water layer covering the leaf surface, and thus wet leaves act as an effective removal of atmospheric $NH_3$ until a certain equilibrium in concentration is reached. Thus, we examined the influence of precipitation on measured fluxes. A separation of fluxes into

different precipitation classes is shown in Fig. F1. In general, median deposition gets lower with increasing precipitation, and emission fluxes can be found in classes with significant rainfall ($>0.5\,\mathrm{mm\,h^{-1}}$). Strongest dry deposition occurs mainly during dry conditions, which is in contrast to the observations of Wyers and Erisman (1998). It has to be considered that the catchment, in which the flux tower is located, has a size of approximately $0.69\,\mathrm{km^2}$ (Beudert and Breit, 2010) and is larger than the catchment of Wyers and Erisman (1998). Also, the surrounding forested area is much larger and the entire area is mountainous.

The forest stand is relatively young since it is recovering from a bark beetle outbreak in the 1990s and 2000s (Beudert and





Breit, 2014). Wyers and Erisman (1998) determined an average $NH_3$ concentration of $5.2\,\mu gm^{-3}$ and median concentration of $3.5\,\mu gm^{-3}$. Their values are at least two times higher than measured $NH_3$ concentrations at our site. Presumably, if $NH_3$ concentrations are low, $\Sigma N_r$ dry deposition seems to be favored by dry conditions. Also, Wolff et al. (2010) measured low $NH_3$ concentrations at their forest site. Figure F1 also demonstrates that concentrations of $\Sigma N_r$ are elevated if leave surfaces are dry.

It shows that wet deposition is important for the uptake of $\Sigma N_r$ compounds at our measurement site. As mentioned in Sec. 2.1, we measured substantial rainfall during 2.5 years at our measurement site. Due to the remoteness of the measurement site, air mass transport starting at potential nitrogen emission sources has to overcome long distances before reaching the site. Thus, a significant amount of $\Sigma N_r$ is probably deposited outside the footprint of the flux tower during rainy periods.

### 4.3  Comparison of different methods for calculating N budgets

#### 4.3.1  Uncertainties of flux measurements and gap-filling approaches

The different gap-filling approaches led to almost the same deposition after 2.5 years. The advantage of inferential modeling is that long gaps in flux time series can be filled. This is not possible with MDV or other recently published gap-filling methods (e.g., Falge et al., 2001; Reichstein et al., 2005; Moffat et al., 2007; Wutzler et al., 2018; Foltýnová et al., 2020; Kim et al., 2020) because the latter are optimized for inert gases. Statistical methods like MDV assume a periodic variability of fluxes.

This assumption is mostly valid for inert gases, which have a distinctive daily cycle. Reactive gases mostly do not exhibit a predictable flux variability. Their flux variability depends on micrometeorological conditions and their chemical and physical properties sometimes leading to instationarities in the data time series. Therefore, the application of statistical methods is rather questionable. However, also DEPAC-1D has some issues, which are not solved or implemented yet. For example, particle deposition is not considered, the implementation of $N_r$ species like $HNO_3$ is relatively straightforward compared to $NH_3$,

an exchange path with soil is not implemented yet, and the cuticular compensation point of $NH_3$ is underestimated under high concentrations and temperatures (Schrader et al., 2016). DEPAC-1D fluxes during winter were close to neutral whereas TRANC measurements show slight deposition and even emission under special circumstances. Further comparison to flux measurements at different sites can help to solve these issues. Gap-filling techniques based on artificial neural networks may be a further valuable option - if available. Uncertainties of the $\Sigma N_r$ fluxes were estimated with the method by Finkelstein and

Sims (2001). The uncertainties of gap-filled fluxes through MDV were calculated by the error of the average. Gap-filled fluxes through DEPAC-1D were not assigned with an uncertainty by the model. As an approximation, we assigned DEPAC-1D fluxes with a relative error of 20%. This relative error is a guess based on uncertainties in the implementation of DEPAC-1D and of the input data. In the following, possible uncertainties sources are mentioned. Considering the input data needed for site based modeling, uncertainties in concentration of $N_r$ compounds and turbulence measurements seem to have the largest impact on

the modeled fluxes. Besides some power outages of a few days, instruments for recording meteorological data were operating continuously. The agreement with modeled data from the ECMWF for the investigated grid cell was excellent (Fig. 7). Thus, uncertainties in meteorological data have a negligible impact on the modeled fluxes. Due to the low time resolution of DELTA and passive samplers, short-term variability is missing in $NH_3$ and $HNO_3$ concentration time series, especially for $HNO_3$.





NH$_3$ measurements were conducted by the NH$_3$ QCL, which allows to measure NH$_3$ with a high time resolution. The low
deposition fluxes modeled by DEPAC-1D during winter are caused by measurement outages of QCL, which led to a missing
variability in concentrations of NH$_3$. Thus, missing values had to be replaced by monthly averages measured by passive and
DELTA samplers. Lower temperatures, which are (at mid-latitude sites) directly related to high stomatal resistances, also lead
to low deposition values during winter. Since NH$_3$ concentration level is generally low during winter and assigned with a
low variability as found by measurements, this procedure is reasonable for a limited time period. Differences in half-hourly
fluxes during these times are difficult to interpret due to the low time resolution of the input data. No fast recording of HNO$_3$
was available at the measurement site. Since HNO$_3$ has also a significant contribution to the $\Sigma$N$_r$ flux, using fast-response
measurements of HNO$_3$ (Farmer et al., 2006; Farmer and Cohen, 2008) in DEPAC-1D or other site-based inferential deposition
models would be a much needed approach for further campaigns. At the moment, the implementation of HNO$_3$ in DEPAC
is relatively simple (see Sec. 4.3.2). At agricultural sites, such an instrumentation for HNO$_3$ is not needed since exchange
processes of $\Sigma$N$_r$ are most likely driven by a high NH$_3$ background concentration.

Uncertainties also arise from the measurement setup: Insufficient pump performance, issues in temperature stability of the
TRANC and CLD, sensitivity loss of the CLD, and problems in the O$_2$ and CO supply. Therefore, regular maintenance and
continuous observation of instrument performance parameters such as TRANC temperature and flow rate were done. With
manual screening of measured half-hours and the recording of these parameters, compromised half-hours could be effectively
excluded from analysis. Since certain sonic anemometers give an incorrect sonic temperature signal, which can be biased or
exhibit a non-linear relationship (Aubinet et al., 2012), sonic temperature was adjusted with the averaged temperature deter-
mined from measurements at 20 m and 40 m. Incorrect high-frequency temperature measurements affect the high-frequency
damping, and therefore the determination of damping factors for $\Sigma$N$_r$. Periods of insufficient turbulence were ruled out with
a threshold for $u_*$ lower than 0.1 ms$^{-1}$ (for details see Zöll et al., 2019, Sec. 2.4) and with the criteria of Mauder and Foken
(2006). A basic assumption for the eddy covariance method is that the terrain needs to be flat, and the canopy height and
density should be uniform (Burba, 2013). These site criteria are not perfectly fulfilled at our measurement site. The site is
located in a low mountain range and tree density is rather sparse south of the flux tower. Such diverse terrain characteristics
could lead to unwanted turbulent fluctuations (non-stationarity of time series), which introduce noise in flux cross-covariance
function. Consequentially, time lag estimation is compromised, and in particular fluxes close to the detection limit may not be
determined correctly. However, situations of insufficient turbulence are mostly likely identified by the applied quality selection
criteria.

Adding the random flux errors determined with Finkelstein and Sims (2001) to the assumed relative errors that correspond
to 20% of DEPAC-1D fluxes results in approximately $\pm$3.4 kg N ha$^{-1}$ for TRANC+DEPAC-1D and $\pm$3.8 kg N ha$^{-1}$ if MDV
is used before applying DEPAC-1D. An uncertainty of $\pm$2.6 kg N ha$^{-1}$ is determined for DEPAC-1D. The dry deposition
budget errors of the different approaches are similar. It shows that the discrepancy to DEPAC-1D lies in the upper range of
the estimated flux uncertainties. Yearly uncertainties of $\Sigma$N$_r$ fluxes were between $\pm$1.0 kg N ha$^{-1}$ a$^{-1}$ and $\pm$1.3 kg N ha$^{-1}$
a$^{-1}$ for 2016 and between $\pm$1.2 kg N ha$^{-1}$ a$^{-1}$ and $\pm$1.7 kg N ha$^{-1}$ a$^{-1}$ for 2017 resulting in an agreement with annual dry





deposition modeled by DEPAC-1D within the flux error range. Higher flux errors correspond to the gap-filling approach that applies MDV to short gaps.

### 4.3.2 Uncertainties of site-based modeling of fluxes

Generally, dry deposition of $\Sigma N_r$ was overestimated by DEPAC-1D. The high contribution of $NH_3$ to $\Sigma N_r$, followed by $HNO_3$, $NO_2$, and NO predicted by DEPAC-1D seems reasonable since $NH_3$ is the most abundant $\Sigma N_r$ compound in certain ecosystems. However, most of the studies were conducted above ecosystems, which are close to $N_r$ sources and agriculturally managed sites. Sites with low variability in pollutants show a different contribution of $N_r$ compounds as shown by our DELTA measurements, especially in $NH_3$. Particulate and acidic $N_r$ compounds hold also an important fraction of the $\Sigma N_r$ flux. On average, their contribution was higher than $NH_3$ showing that the current implementation of $N_r$ compounds such as $HNO_3$ or $NO_2$ should be reevaluated, and the inclusion of exchange mechanisms for $NO_3$ and $NH_4$ should be considered in-situ modeling approaches.

Since a direct comparison to $NH_3$ flux measurements, the main compound in the deposition models, was not possible, only assumptions about the difference to measured fluxes can be given. The parameterization of the $NH_3$ exchange inside DEPAC could be responsible for the discrepancy to TRANC fluxes. Schrader et al. (2016) discovered problems in the calculation of the cuticular $NH_3$ compensation point, especially under high ambient $NH_3$ concentrations and high temperatures, for instance during summer. Thus, cuticular deposition is overestimated. This issue is not solved yet and could not be verified for our measurement site due to generally low $NH_3$ concentrations and to the implementation of monthly averaged $NH_3$ concentration instead of half-hourly values. Since flux measurements on $\Sigma N_r$ were conducted, we cannot extract the reason for the overestimation from measurements. Due to the low $NH_3$ concentrations cuticular compensation point exhibits no bell-shaped trend, which pronounced at high temperatures and high $NH_3$ concentrations (see Fig. 2(b) of Schrader et al., 2016). Thus, this issue is not the main reason for the difference to flux measurements at our site. It should be kept in mind that the determination of the compensation point may be critical, and a precise determination may not be possible under low concentrations of $\Sigma N_r$ compounds. The measurement site is located in a low polluted mountain range. As stated in Sec. 4.2, mechanisms for favoring the dry deposition of $\Sigma N_r$ are different to sites located in high polluted surroundings. Currently, a compensation point for the exchange path with soil is not implemented in DEPAC. Including such an exchange path in DEPAC, can lead to a reduction in deposition at sites with generally low $\Sigma N_r$ deposition.

As mentioned in Sec. 4.1, $HNO_3$ has a significant influence on the $\Sigma N_r$ fluxes. The median deposition velocity of $HNO_3$ modeled by DEPAC-1D is almost similar to $NH_3$. Thus, $HNO_3$ holds an important role in the $\Sigma N_r$ exchange at our site. The implementation of $HNO_3$ inside DEPAC by a constant, low canopy resistance is rather simple. Compensation points are only calculated for $NH_3$. Thus, other $N_r$ compounds can only be deposited in the model. It is expressed in the positive deposition velocities. Overall, median, modeled deposition velocities are close to the values propagated by VDI (2006) (Fig.C1.). $NH_3$ deposition velocity is in agreement with Schrader and Brümmer (2014) for different forest types. The negative whisker indicates few phases of emission, but they had hardly any influence on the nitrogen budget. In the case of $HNO_3$, the assumption of an ideal uptake seems to be questionable (Tarnay et al., 2002). Flux measurements of $HNO_3$ were conducted by Farmer and Cohen





(2008) above spruce forest. They detected significant emission of $HNO_3$ during summer. $HNO_3$ emission during summer can be caused by evaporation of $NH_4NO_3$, which favored at temperatures above 20°C (Wyers and Duyzer, 1997; Van Oss et al., 1998). The mechanism explaining the $HNO_3$ emission is still under investigation (Nemitz et al., 2004).

DEPAC-1D models a low, positive deposition velocity for $NO_2$ since no bidirectional pathway is implemented for $NO_2$ in DEPAC. Low deposition velocities of $NO_2$ for different tree types are also reported by Wang et al. (in review, 2020), but the investigated tree types are not representative for our site. However, the order of magnitude is comparable to the modeled deposition velocity of $0.08 \, cms^{-1}$ for $NO_2$. Since they detected no NO uptake for all tree types, a modeled deposition velocity of $0.0 \, cms^{-1}$ with a negligible extension of the box for NO seems to be reasonable. Delaria et al. (2018) also observed low
deposition velocities for $NO_x$. They found a stomatal deposition velocity of $0.007 \, cms^{-1}$ and a cuticular deposition velocity of $0.005 \, cms^{-1}$ for NO. This indicates a marginal NO uptake, which was about one magnitude smaller than the $NO_2$ uptake (Delaria et al., 2018). In general, canopy resistance mostly driven by water solubility. Thus, gases with a low water solubility like NO and $NO_2$ exhibit similar deposition velocities for different tree types. A compensation point for $NO_2$ was not found by Delaria et al. (2018) showing forest as an effective removal of $NO_2$ (Rosenkranz et al., 2006; Geddes and Murphy, 2014).
Taking no compensation point for $NO_2$ by DEPAC seems to be reasonable. For verifying these assumptions further comparisons of flux measurements with exchange models are recommended because they can lead to significant improvements of the implemented parameterizations for various $\Sigma N_r$ compounds. Focusing on $NH_3$, the most abundant species in rural areas, is also recommendable.

### 4.3.3   Uncertainties in the implementation of LOTOS-EUROS

The high nitrogen deposition values modeled by LOTOS-EUROS at the measurement site is mostly related to a general over-estimation of ammonia concentrations especially occurring above Baden-Württemberg and Bavaria (Schaap et al., 2017). The disagreement to CBT deposition estimates was observed for elevated locations, which are exposed to a high amount of occult deposition (Schaap et al., 2017). Ge et al. (2020) compared LOTOS-EUROS $NH_3$ emission for two emission scenarios to satellite and surface observations for Germany and Benelux. The first emission scenario is the emission inventory from
MACC-III (Modeling Atmospheric Compostion and Climate), which is originally used by LOTOS-EUROS, the second one is an updated version with increased detail level in nitrogen emission sources. Calculated annual total columns from the first scenario underestimated $NH_3$ from the satellite IASI (Infrared Atmospheric Sounding Interferometer), annual total columns from the second scenario under and overestimated $NH_3$ satellite-derived total columns. In the latter case, the overestimation was located to Southern Germany. A comparison to surface observations showed that LOTOS-EUROS overestimates $NH_3$
concentrations from January to March for both scenarios. At the measurement site, we also found a disagreement to $NH_3$ measurements conducted with QCL, DELTA, and passive samplers during winter (Fig. 7). Until mid of February, measured values were lower than 0.5 ppb whereas modeled concentrations ranged from 0.5 to 1.5 ppb. The difference to LOTOS-EUROS $NH_3$ concentrations was highest during periods with significant amount of $NH_3$ in the atmosphere like in spring and autumn, which is caused by emissions from fertilizer leading to a high load of modeled concentrations. Hence, modeled dry deposition
is clearly overestimated.





The influence of emissions caused by management processes at adjacent sites on measured $\Sigma N_r$ fluxes could not be verified. The largest amount of $N_r$ released from those processes into the atmosphere will be deposited close to their sources. A small amount will be transported up to distances of 100 km (Asman et al., 1998; Ferm, 1998; Loubet et al., 2009). The released $NH_3$ going into long-range transport is highly variable (Loubet et al., 2009), and the distance depends on several parameter like

atmospheric stability, atmospheric chemistry, topology, etc. In case of stable stratification, inversion layers often occurring in mountain ranges can prohibit air mass exchange. Probably, the measurement site is mostly outside the transport range. Thus, nitrogen enriched air-masses are deposited before reaching the height of the flux tower. A reduction in grid cell size could lead to a more precise localisation of potential nitrogen emission sources. Since all exchange processes contribute to single concentration within a grid cell, an improvement in horizontal resolution will lead to a refinement in predicted concentrations.

The aerodynamical reference height, which is used by LOTOS-EUROS for flux calculation, is also lower than the measurement height of the flux tower. Thus, slight differences in micrometeorological data can be expected, for example the difference in relative humidity in the first half of 2016. Differences for that time period are related to the usage of meteorological data provided by the NPBW, with their instrumentation being installed at the 50 m platform. The deviations in $u_*$ are most likely related to the complex terrain within the foot print of the flux tower. The surface roughness length and the tree composition

is not uniform for the entire footprint. It is not possible to model such a diverse canopy structure within $7 \times 7$ km$^2$ grid cell accurately. As stated earlier, the weighting of the land-use classes within the grid cell was not representative for the foot print. The class "semi-natural grassland" has the highest contribution. However, Norway spruce and European Beech were found to be the most dominated tree type within the flux foot print. This issue could be partly solved by increasing the spatial resolution. The reduction in grid cell size could affect the fractions of $N_r$ compounds to modeled $\Sigma N_r$ concentrations (Fig. E1). The

influence of $NH_3$ on $\Sigma N_r$ could change, and thus the predicted $\Sigma N_r$ dry deposition can be lowered since reduction in $NH_3$ has the strongest influence on the deposition (Fig. 7).

As stated in Sec. 2.4.3, an incorrect setting of the LAI and $z_0$ can have a significant influence on $\Sigma N_r$ deposition. The results of our sensitivity analysis for LAI and $z_0$ are comparable to values presented recently by van der Graaf et al. (2020), who used satellite-derived LAI and $z_0$ data from Moderate Resolution Imaging Spectroradiometer (MODIS) to calculate $\Sigma N_r$

deposition with LOTOS-EUROS for a grid cell size of $7 \times 7$ km$^2$. Overall, they observed changes in $\Sigma N_r$ dry deposition of up to 30%. However, there is almost no change in $\Sigma N_r$ dry deposition and in $NH_3$ concentration observable for the Bavarian Forest measurement site if LAI and $z_0$ from MODIS are used. However, the attempts of van der Graaf et al. (2020) and Ge et al. (2020) did not provide a solution for the general overestimation of $NH_3$ deposition above southern Germany. It seems that the larger scale and temporal discrepancies in input $NH_3$ concentrations in LOTOS-EUROS are mainly responsible for the

disagreement to flux measurements, and overestimation is only partly related to other issues, for example, the grid cell size of $7 \times 7$ km$^2$.

Finally, two special $\Sigma N_r$ exchange events need to be discussed, the $\Sigma N_r$ emission fluxes in December 2017 and the deposition fluxes in February 2018. The emission phase in December 2017 may be related to the decomposition of fallen leaves (Hansen et al., 2015). Since the compensation point of the soil is set to zero for all land-use classes, the decomposition of fallen

leaves is not considered in the models, and thus emissions from the soil could not be modeled. The deposition event in February





2018 seen by the TRANC seems to be driven by particulate $N_r$. Comparing the different runs of LOTOS-EUROS shows that the contribution of particulate deposition to total deposition is much larger than gaseous deposition during that time. However, the amount of deposited $\Sigma N_r$ of this event is underestimated by DEPAC-1D and LOTOS-EUROS. A second deposition event, which occurred directly after the mentioned one, was predicted by the models, but not confirmed by the measured fluxes.

Considering the yearly uncertainties of TRANC measurements, upper CBT estimates of nitrogen deposition values are outside the error range of flux measurements. TRANC values are closer to the lower estimate of CBT. CBT values for 2016 and 2017 are almost similar whereas high dry deposition was determined for 2018. The difference to the previous years may be related to the higher particle input in February 2018 as shown by LOTOS-EUROS and TRANC measurements. However, the order of magnitude is the same and measured dry deposition is within one standard deviation of the averaged lower CBT
estimates from 2010 to 2018 under consideration of the flux error range. LOTOS-EUROS and DEPAC-1D yearly estimates are within the error range of the CBT estimates, in particular close to the overlap area of the error ranges. By applying the correct land-use class weighting, LOTOS-EUROS values are close to the upper estimate of CBT. It shows that dry deposition of the different methods are in the range of statistical uncertainty. Deviations from TRANC measurements are most likely related to differences in the vegetation of the footprint and the selected tree types. Inside the footprint, the forest stand consists of dead
wood in south direction and young and matured trees in easterly direction. The investigated trees for CBT were selected from a matured tree stand. Thus, the leaf area surfaces can be significantly different. Their susceptibility to precipitation may differ, too. Different leaf sizes and different tree ages are probably the main reasons for the disagreement to TRANC fluxes.

## 5   Conclusions

Our study is the first one presenting 2.5 years flux measurements of $\Sigma N_r$ measured with a custom-built converter (TRANC)
coupled to fast-response CLD above a protected mixed forest. We investigated temporal dynamics of $\Sigma N_r$ exchange, discussed conditions favouring natural exchange characteristics of $\Sigma N_r$ under low atmospheric concentrations, and compare annual budgets of flux measurements to an in-situ deposition model, DEPAC-1D, and a long-range chemical transport model, LOTOS-EUROS.

    Measured concentrations of $\Sigma N_r$ were 5.2 ppb on average. Reactive compounds such as $NH_3$ and $NO_2$ had a concentration
level of 1.8 ppb and 2.5 ppb, respectively. The latter exhibits highest concentrations during winter, the former during spring. Elevated concentration level is possibly related to anthropogenic emission during those periods. DELTA measurements showed that $NH_3$ and $NO_2$ are the main contributors to $\Sigma N_r$. On average, these gases contribute with 73.2% to $\Sigma N_r$. These reactive gases are most responsible for observed exchange pattern of $\Sigma N_r$ at the measurement site. However, also particulate and acidic $N_r$ compounds are important for the dynamics of $\Sigma N_r$ exchange, especially at high $\Sigma N_r$ concentrations. We observed
mostly deposition during 2.5 years of flux measurements. Median deposition ranges from -15 to -5 ng N m$^{-2}$s$^{-1}$. Highest deposition was observed during mid spring and summer, lowest deposition occurred during late autumn and winter. From May to September deposition was favored under high ambient concentration ($> 4.7$ ppb), low humidity level ($< 77\,\%$), and high temperatures ($> 14.3\,^\circ$C). Additionally, dry leaf surfaces seem to enhance deposition. We conclude that dry conditions seem to





favour $\Sigma N_r$ dry deposition at natural ecosystems supposedly related to a low contribution of $NH_3$ to the $\Sigma N_r$ fluxes. We found

that concentrations of $\Sigma N_r$ were elevated in presence of dry leaf surfaces. Thus, wet deposition seems to be important for $\Sigma N_r$ deposition at our measurement site during rainy periods. After 2.5 years, nitrogen dry deposition of TRANC measurements resulted in $(11.1 \pm 3.4)\,\text{kg N ha}^{-1}$ with DEPAC-1D as gap-filling method, and $(10.9 \pm 3.8)\,\text{kg N ha}^{-1}$ was determined with MDV and DEPAC-1D as gap-filling methods. Both values are rather close to modeled fluxes of DEPAC-1D $(13.6\,\text{kg N ha}^{-1})$ considering the uncertainties of measured fluxes and possible uncertainty sources of DEPAC-1D. Difference of DEPAC-1D to

TRANC could be related to the parameterizations of reactive gases or the missing exchange path with soil. Further comparisons of in-situ models to flux measurements are needed to address these issues. Both gap-filling approaches result in similar nitrogen dry deposition values. The advantage of DEPAC-1D is based on the gap-filling of long time series of missing data. However, there are still issues in the bidirectional resistance model DEPAC, which need to be solved. Up to now, there is no further option in replacing long-term gaps because most gap-filling methods are designed for inert gases. Gap-filling methods, which

based on artificial neural networks, could also be useful for reactive gases.

LOTOS-EUROS exhibited the highest discrepancy to flux measurements, in particular for the actual land use of the grid cell $(16.8\,\text{kg N ha}^{-1})$. We showed that modeled $NH_3$ concentrations used as input parameter by LOTOS-EUROS were significantly higher than measured concentrations, and they disagreed in their seasonal pattern. Thus, modeled $NH_3$ concentrations were the main reason for the discrepancy in annual budgets. Also, the vegetation of the grid cell does not correspond to the vegetation

of the flux footprint. Increasing the horizontal resolution could be a solution to that issue. Supposedly, a large-scale issue is related to the overestimation of $NH_3$ concentration by LOTOS-EUROS.

Averaged annual $\Sigma N_r$ dry deposition was $4.5\,\text{kg N ha}^{-1}\,\text{a}^{-1}$ for both gap-filling approaches applied to TRANC measurements, DEPAC-1D showed $5.3\,\text{kg N ha}^{-1}\,\text{a}^{-1}$, and LOTOS-EUROS modeled $5.2\,\text{kg N ha}^{-1}\,\text{a}^{-1}$ to $6.9\,\text{kg N ha}^{-1}\,\text{a}^{-1}$ depending on the weighting of land-use classes. The application of CBT resulted in $7.5\,\text{kg N ha}^{-1}\,\text{a}^{-1}$ as upper estimate and

$4.6\,\text{kg N ha}^{-1}\,\text{a}^{-1}$ as lower estimate. Dry deposition estimated by TRANC, DEPAC-1D, and LOTOS-EUROS is within the frame of minimum and maximum deposition estimated by CBT. The difference of flux measurements to CBT could be induced by the discrepancy in tree age of the selected trees for CBT compared to the forest stand within the footprint, and leaf area surfaces may also be different.

For a further improvement of deposition models and the investigation exchange characteristics of $\Sigma N_r$, long-term flux mea-

surements are needed for different ecosystems differing in their nitrogen stress. However, installing a setup presented in this study at several locations is quite challenging due to power consumption, costs of the instruments, and their high technical requirements. A continuous monitoring of $N_r$ species by low-cost samplers complemented by high-frequency measurements of $\Sigma N_r$ and selected compounds like $NH_3$ for a limited time, for example during fertilization periods, can result in a better understanding of exchange processes and thus in a improvement of deposition models (Schrader et al., 2018). Recently, Schrader

et al. (2020) showed that stomatal conductances, essential for controlling the $NH_3$ exchange between vegetation and atmosphere, can be determined from $CO_2$ flux measurements. Using $CO_2$-derived stomatal conductances will lead to a significant improvement of biosphere–atmosphere exchange models making them sensitive to climate change effects.





## Appendix A: Time lag determination of the TRANC-CLD system

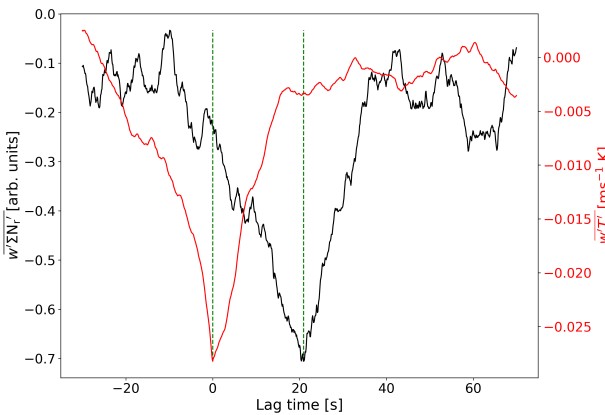

**Figure A1.** Covariance function of vertical wind and temperature (red) and covariance function of vertical wind and $\Sigma N_r$ concentration
(black). Green, dashed lines indicate the maximum covariance, which is around 20 s for the TRANC-CLD. Data were recorded at the 22
April 2017 from 05:00 to 05:30 CET




## Appendix B: Contribution of different of $N_r$ gases and particles to $\Sigma N_r$ based on DELTA measurements

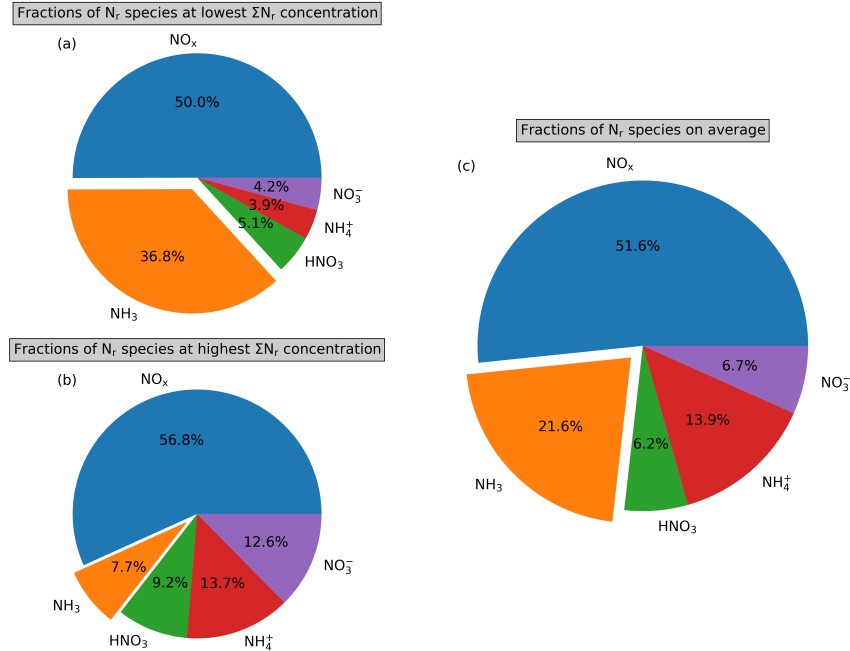

**Figure B1.** Pie charts showing the contribution of $NO_x$, $NH_3$, $NO_3$, $NH_4$, and $HNO_3$ to $\Sigma N_r$ based on measurements of DELTA samplers and $NO_x$ measurements. $NO_x$ measurements are averaged to exposition periods of the DELTA samplers. (a) and (b) show the contributions to the highest and lowest average $\Sigma N_r$ concentration found for the measurement campaign. (c) shows the average contribution to $\Sigma N_r$ for the entire measurement period.





## Appendix C: Deposition velocities determined by DEPAC-1D

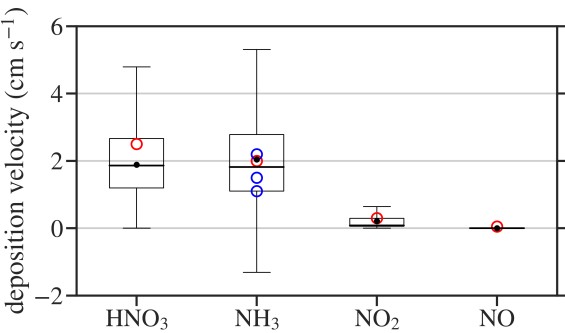

**Figure C1.** Box plots of deposition velocities for $NH_3$, $HNO_3$, $NO_2$, and NO modeled by DEPAC-1D without outliers (the box frame is the 25% to 75% interquartile range (IQR); the length of whiskers is 1.5 times the IQR; the bold line is the median). Blue circles are $NH_3$ deposition velocities by Schrader and Brümmer (2014) for deciduous forest, mixed forest, and spruce forest (from low to high), red circles show deposition velocities after VDI (2006). Negative deposition velocities of $NH_3$ are related to modeled emission phases.

## Appendix D: Difference between measured and modeled $\Sigma N_r$ fluxes for the entire campaign

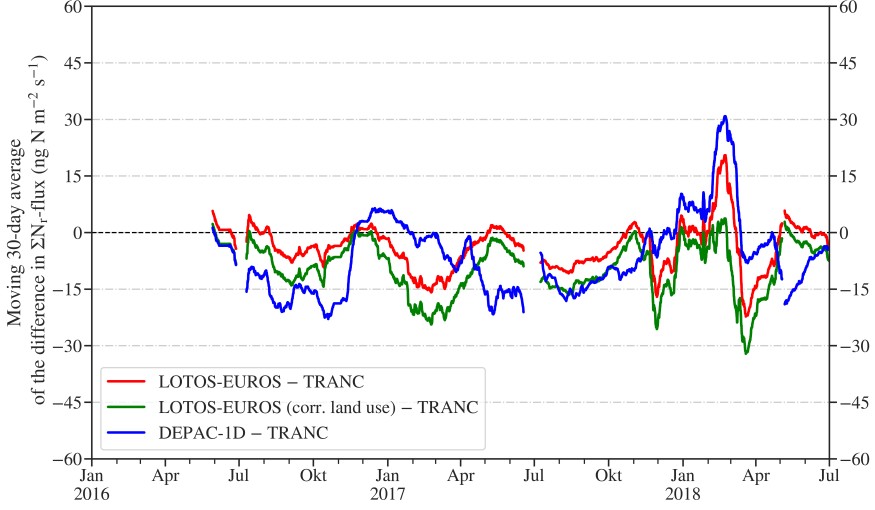

**Figure D1.** Moving 30-days average of the difference between half-hourly measured and modeled $\Sigma N_r$ fluxes. Negative values indicate an overestimation of the deposition by the DEPAC-1D and LOTOS-EUROS, positive values refer to an underestimation.





**Appendix E: Contribution of different of $N_r$ gases and particles to $\Sigma N_r$ based on LOTOS-EUROS**

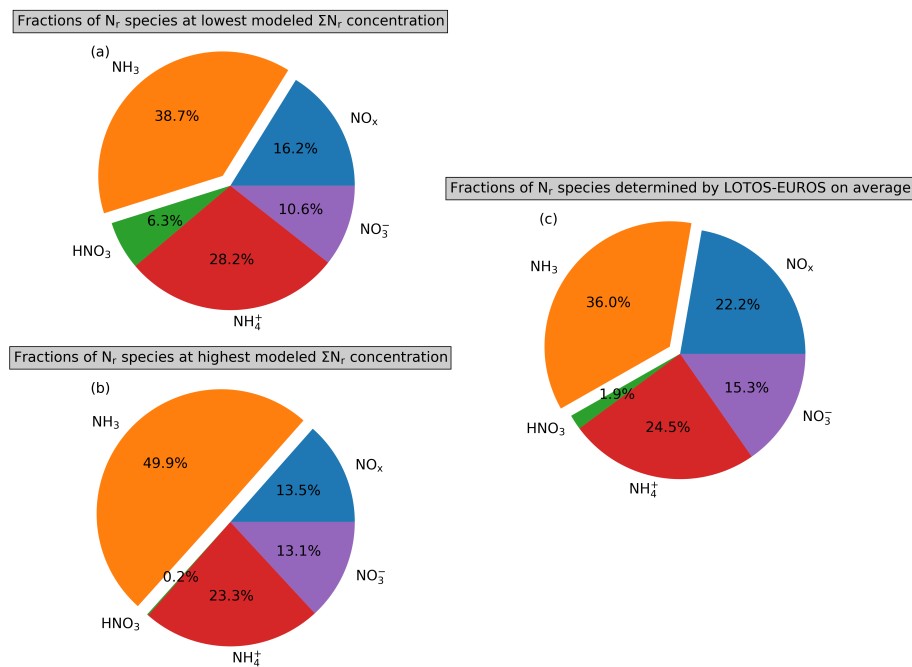

**Figure E1.** Pie charts showing the contribution of $NO_x$, $NH_3$, $NO_3$, $NH_4$, and $HNO_3$ to $\Sigma N_r$ based on modeled concentrations of LOTOS-EUROS. Modeled concentrations are averaged to exposition periods of the DELTA samplers. (a) and (b) show the contributions to the highest and lowest average $\Sigma N_r$ concentration found for the measurement campaign. (c) shows the average contribution to $\Sigma N_r$ for the entire measurement period.





**Appendix F: Box plots of $\Sigma N_r$ concentrations for wet and dry leaves and fluxes separated into precipitation classes.**

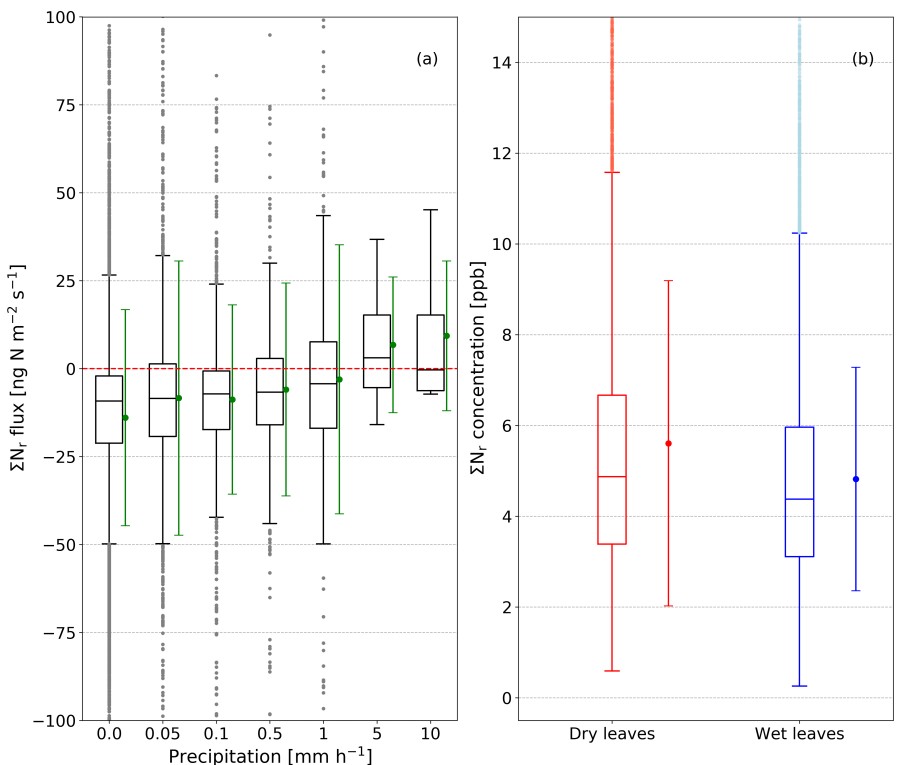

**Figure F1.** Box plots of $\Sigma N_r$ concentrations for wet (blue) and dry (red) leaves (b) and fluxes separated into precipitation classes (a) (the box frame is the 25% to 75% interquartile range (IQR); the length of whiskers is 1.5 times the IQR; the bold line is the median). Averaged values of the corresponding classes (green) are plotted to the right of the box. Uncertainty of the averaged values are indicated by error bars, whose lengths correspond to one standard deviation.

*Author contributions.* PW, FS, and CB conceived the study. PW wrote the manuscript, carried out the measurements at the forest site, and conducted flux data analysis and interpretation. FS implemented the stand-alone version of DEPAC, analyzed the model output, and evaluated meteorological measurements. MS provide the LOTOS-EUROS data, and BB performed the canopy budgets analysis. CB installed the flux
tower equipment and gave scientific advise to the overall data analysis and interpretation. All authors discussed the results and FS, MS, BB, and CB reviewed the manuscript.

*Competing interests.* The authors declare that they have no conflict of interest.





*Acknowledgements.* Funding by the German Environment Agency (UBA) (project FORESTFLUX, support code FKZ 3715512110) and by the German Federal Ministry of Education and Research (BMBF) within the framework of the Junior Research Group NITROSPHERE

(support code FKZ 01LN1308A) is greatly acknowledged. We thank Undine Zöll for scientific and logistical help, Jeremy Rüffer and Jean-Pierre Delorme for excellent technical support, Ute Tambor, Andrea Niemeyer, and Dr. Daniel Ziehe for conducting laboratory analyses of denuder and filter samples, and the Bavarian Forest Nationalpark (NPBW) Administration, namely Wilhelm Breit and Ludwig Höcker for technical and logistical support at the measurement site.





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
