# Peer review of "Forest-atmosphere exchange of reactive nitrogen in a low polluted area – Part I: Measuring temporal dynamics"

_Biogeosciences, 2020_

## Referee Comment (RC1) · Anonymous Referee #1 · 9 Dec 2020

This manuscript presents 2.5 years of measurements of total reactive nitrogen (Nr) fluxes above a mixed forest in Germany. The measurements are used to assess annual dry deposition budgets and are then compared to deposition estimates derived from a field scale model and a gridded chemical transport model. This study directly addresses the need for new Nr flux measurements to improve Nr deposition budgets, assess exceedances of critical loads of Nr, and improve models of reactive N deposition. The dataset developed is novel and should prove useful to the ecological and atmospheric chemistry communities interested in N deposition. Furthermore, the interpretation of the measurements in relation to micrometeorology and atmospheric chemistry sheds new light on the processes influencing air-surface exchange of Nr and the

relative importance of Nr species to the dry Nr deposition budget. However, there are a number of technical details of the analysis and discussion, along with some organizational issues, that should be addressed before the paper is suitable for publication.

In general, the paper would benefit from a more thorough quantitative analysis of the flux patterns and their relation to micrometeorology and atmospheric chemistry. Section 4.2 touches on these relationships but could be extended along the lines of several suggestions outlined below. As also suggested below, the current content of Section 4.2 could be reorganized and shortened by eliminating some redundancies, making it possible to expand the analysis without significantly lengthening the Section overall. Sections 4.3.2 and 4.3.3, which describe uncertainties in the modeling approaches, as well as the Conclusions section, could be significantly reduced in length. More specific comments are detailed below.

Specific comments: Line 83: Change "pattern" to "patterns".

Line 90: Should the first word be "methods"?

Line 94: CTM should be plural

Line 117: "site located" should be "site is located"

Line 145: For clarity, consider rewording this sentence to something like: "In a 2nd step, a gold tube passively heated to 300C catalytically converts the remaining oxidized Nr species to NO."

Line 148: Is 2.1 L/min the flow rate through the converters (atmospheric pressure) or through the reduced pressure portion of the tubing downstream of the orifice? If the latter, please indicate the flow rate through the converters.

Line 164: What type of passive sampler was used? What was the sampling duration?

Line 166: Is there a reference specific to the QCL NH3 measurements performed at this site?

Line 207: What was the typical magnitude of this correction to the total Nr flux?

Line 216: What caused the reduced sensitivity of the CLD and how was it identified?

Line 266: How was the quality of the DELTA measurements assessed?

Line 275: Has LAI been measured at this site? How variable is the LAI throughout the seasons, given the relative fractions of spruce and beech.

Line 285: Should be "gaps in micrometeorological".

Figure 1: The blue bar in the lower plot is incorrectly labeled NH3.

Line 296: The findings here relative to concentrations of NOx > 20ppb make me question the description of this site as being situated in a "low pollution" area. Some additional justification of this site characterization is needed.

Line 302: The figure numbering configuration for the Appendices (e.g., Figure B1) was not immediately clear to me. I believe the format for Biogeosciences is for such material to be included as "Supplemental Material".

Line 303: How does the sum of the concentrations measured by the DELTA compare to the TRANC Nr measurement?

Line 317: What fraction of the non gap-filled half-hourly fluxes exceeded the flux detection limit?

Line 350: Should "based on" be "are based on"?

Line 352: I might expect the sensor nearest the ground to remain "wet" later into the morning than the sensor closest to the top of the canopy.

Line 398: "It seems that....most likely driven by particulate Nr compounds." Is this supported by the particulate measurements? Do the DELTA measurements show relatively higher concentrations of particulate NH4NO3 during this period? Given the lower Vd of particles relative to gases, the concentrations would need to be much higher to

drive the high total N deposition during this period, correct?

Lines 404 and 405: So a linear interpolation is used? Please clarify.

Line 411: "..significantly higher lower and upper.." I understand what you mean here but it is a little confusing. Consider rewording for clarity.

Line 425: "4.6 kg N/ha/a are determined as a lower estimate." Please clarify how this estimate was determined.

Line 426: Should this section heading read "Sensitivity of deposition estimates to measured vs. modeled input parameters"?

Line 428: Remove comma after "class" and add "the" after "considering".

Line 432: Specify that you are referring to the apoplastic ratio of NH4+ to H+. I would also suggest you clarify that you are referring to the stomatal compensation point in the latter part of this sentence. Have any measurements of the soil and vegetation chemistry been conducted at this site such that compensation points could be estimated?

Line 435: "...global radiation enhances the opening of the width of the stomata" It may be more straightforward to say that the stomatal resistance is influenced by global radiation.

Line 439: I would suggest that reporting the bias (absolute percent) in the modeled values relative to the measured values is more informative than the correlation in this context.

Lines 450 - 455: A table comparing the measured Nr species concentrations (Delta compounds, NOx, QCL NH3, passive NH3) to LOTOS_EUROS would help clarify this section.

Line 458: "... are very low compared to other studies" This statement is true relative to the three references cited but perhaps not so for Nr flux studies in a global sense. Some additional context is required for this statement, e.g., low relative to sites influenced by

agricultural activities, previous studies in European ecosystems, etc.

Line 464: Consider modifying sentence to ". . .higher ground-level concentrations. . .".

Line 470: "Values. . .." This sentence seems incomplete.

Line 472: ". . .confirm the seasonal pattern of Nr". Do you mean that those studies show patterns consistent with the current study?

Line 473: "Obviously, measured concentration levels were significantly higher since the observed ecosystems were subject of agricultural management or in close proximity to industrial or agricultural emissions." Are the authors referring here to the studies listed in Line 471? At least for the Geddes study, NOx was lower than in the present study. Please clarify and correct this statement as needed.

Line 477: This sentence should include references for the "few studies focusing on Nr".

Line 483: Please consider changing "their flux pattern" to "the flux pattern observed by Ammann et al. (2012). . .."

Line 486: Please reference the "few studies" that measured fluxes Nr compounds mixed forests.

Lines 488 – 497: The discussion of the high emission fluxes observed in December requires some additional detail and clarification. The authors refer to decomposition of fallen leaves beneath a snow layer. Are the authors suggesting that the decomposition is enhancing emissions of NH3 or NO or both? Decomposition rates typically decrease at low temperatures. The authors mention that they "discovered an increase in nitrogen concentration in the investigated samples". Samples of what? Soil? How were these samples taken and analyzed and for which compounds? How frequently were they collected and at what depths? How much did the N concentrations increase and over what time period? The statement regarding the influence of the freeze-thaw cycle on the emission fluxes is interesting but very speculative. Can a soluble gas like NH3 diffuse through a partially wet snow layer to the atmosphere? Do the fluxes correlate

with air temperature in a pattern that would support this statement? Looking more closely at the December diurnal profiles in Figure 3 it appears the emission fluxes were mostly observed in 2017, which also had much higher variability in general than 2016. Were there more periods of snow cover in 2017? Did the two years differ in other ways in terms of meteorology or air concentrations that might help explain the emissions observed in 2017?

Line 508: Change "proposed" to "reported".

Line 511: Change "by DELTA" to "by the DELTA".

Line 528: Change "high" to "large".

Line 529: Change "at less" to "at a less".

Line 530: Change "It shows" to "These studies indicate"

Line 539: Please consider splitting up this long sentence for clarity.

Lines 544 – 547: The last two sentences of this paragraph seem more appropriate for the conclusions section.

Section 4.2: In general this discussion would benefit from some reorganization and a more thorough quantitative evaluation of relationships between flux, micrometeorology, and air concentrations. The authors discuss radiation/photosynthesis, air concentration, dryness/RH/temperature, and precipitation as important variables. Perhaps these can be discussed in sequence, rather than jumping back and forth among them throughout the section, to make the discussion read more smoothly and to eliminate redundancies. For example, the role of air concentration is mentioned in numerous places, as are relative humidity and temperature. Some care should be given to revising this section as it will be of particular interest to readers seeking a better understanding of the processes driving Nr fluxes above forests.

Line 548: Section heading 4.2 only mentions micrometeorology but much of the following discussion involves the relationship between flux and air concentrations. Consider rewording.

Line 549: What is the proposed mechanistic relationship between Nr flux and global radiation? What about the diurnal pattern of turbulent mixing and its role in air-surface exchange?

Line 551: The authors discuss the relationship between air concentration and flux in several places in Section 4.2. Can the authors be a bit more quantitative in this analysis? What is the relationship (scatterplot) between concentration and flux if, for example, the dataset is filtered to include only mid-day fluxes (i.e., periods of high global radiation and friction velocity)? Is a clear relationship observed? What are the observed diurnal patterns in concentration? Do these patterns confound the relationship with global radiation mentioned in line 549? The authors should consider adding figures similar to figures 2 and 3 but for TRANC Nr concentration in supplemental material.

Line 553: What do the authors mean by "favor" in this sentence?

Line 556: How is the last sentence in this paragraph justified by the preceding sentence? I must be missing something here.

Line 558: The authors compare March and April of 2017 and 2018 as an example of the potential role of photosynthesis in the interannual variability of fluxes. The explanation cites the role of temperature in stomatal function (and therefore the stomatal resistance) but what about the role of radiation? Are there differences in radiation between the two years that would also support this explanation?

Line 562: "...confirmed by the similar daily cycle for May 2017 and 2018." Similar daily cycle of what? Please specify.

Line 567: "Almost the same average....". This sentence is out of place relative to the rest of the paragraph. Please consider removing or consolidating with analysis of relative humidity and temperature in next paragraph.

Line 571: The first sentence of this paragraph should either be removed or reworded. The use of "Therefore" implies a missing introductory sentence.

Line 572: What is the proposed mechanism by which dry conditions enhance Nr deposition? Are the authors proposing that the stomatal processes are a larger overall source of variability in the net canopy-scale flux than the cuticular processes? It is unclear from this paragraph, which seems to include multiple lines of analysis the connections or which are unclear as currently written. Please see my previous comment regarding the organization and clarity of section 4.2

Line 573: The sentence "Higher concentrations values lead to higher deposition values through the entire daily cycle." seems out of place. How does this statement relate to the preceding sentence?

Line 576: "Higher temperatures increase the opening size of the stomata leading to increased photosynthetic activity." What do the authors mean by "photosynthetic activity" in the context of the Nr fluxes?

Line 580: The role of wetness and RH with respect to cuticular deposition of NH3 are reviewed by Massad et al. 2010 https://acp.copernicus.org/articles/10/10359/2010/ and Flechard et al. 2013. https://bg.copernicus.org/articles/10/5183/2013/.

Line 584: "Thus we examined the influence of precipitation on fluxes." Would it not be more straightforward to compare fluxes during wet versus dry conditions as indicated by the leaf wetness sensors, perhaps binning by day versus night or air concentration to examine the relationship while controlling for other sources of variability? I'm not sure what precipitation rate in figure F1 is telling us about the relationship between flux and canopy wetness. Is the canopy any less wet (or leaf water layers thinner) after a prolonged 0.5 mm/h rainfall compared to short duration 5 mm/h rainfall? To clarify, are these flux measurements conducted during active precipitation? What is the quality of the EC fluxes during such periods? Please add another figure to F1 similar to plot b) but for the fluxes and include in discussion.

Figure F1: Please begin the caption by describing plot a) rather than plot b).

Line 587: "It has to be considered that the catchment, in which the flux tower is located, has a size of approximately 0.69 km2 (Beudert and Breit, 2010) and is larger than the catchment of Wyers and Erisman (1998). Also, the surrounding forested area is much larger and the entire area is mountainous. The forest stand is relatively young since it is recovering from a bark beetle outbreak in the 1990s and 2000s (Beudert and Breit, 2014)." Please clarify how these statements are relevant to discussion of the relationship between surface wetness and flux.

Line 592: "Presumably, if NH3 concentrations are low, Nr dry deposition seems to be favored by dry conditions." Please clarify how this conclusion follows from the analysis of the Wyers and Erisman (1998) and Woff et al (2010) studies. What would be the underlying leaf-level mechanism?

Lines 595-598: It is unclear how the sentences on wet deposition relate to the rest of the paragraph. Please consider removing.

Line 609: "the implementation of Nr species like HNO3 is relatively straightforward compared to NH3" is out of place in this sentence. Consider removing.

Line 618: Change "uncertainties sources" to "sources of uncertainty".

Line 633: Change "much needed approach" to "much improved approach"

Line 663: "most of the studies. . ." Please indicate which studies the authors are referring to.

Line 667: "and the inclusion of exchange mechanisms for NO3 and NH4 should be considered in-situ modeling approaches." Please clarify what is meant here.

Line 671: As a general question, how well does the DEPAC total Nr flux reflect the relationships between measured TRANC Nr flux and radiation, temperature/RH/dryness described in section 4.2?

Line 682: And at sites with sparse vegetation.

Line 685: Change "almost similar" to "similar".

Line 688: Has VDI been explained/defined?

Line 689-690: The two sentences here related to NH3 should be move to the preceding paragraph.

Line 696: The use of "positive" to describe the deposition velocity is not necessary.

Line 712: Why is CBT mentioned here in the discussion of LOTOS-EUROS?

Line 720: As previously mentioned, a summary and comparison of the various measurement techniques would be helpful to this discussion. Could the authors add a table summarizing the statistics of QCL, DELTA, and passive measurements, along with the LOTOS_EUROS predictions, as supplemental material? How well did the measurement techniques agree?

Line 722: "The difference to LOTOS-EUROS NH3 concentrations was highest during periods with significant amount of NH3 in the atmosphere like in spring and autumn, which is caused by emissions from fertilizer leading to a high load of modeled concentrations." Please reword this sentence, avoiding the use of "like" and "load".

Line 726: I encourage the authors to revisit the point and usefulness of this paragraph. As written I can't see that it adds anything to the discussion.

Line 760: "The deposition event in February 2018 seen by the TRANC seems to be driven by particulate Nr." Do the DELTA measurements reflect higher NH4+ and NO3- concentrations during this period compared to other months? These data should be presented.

Line 775: The details here (i.e., "were selected from a matured tree stand") highlight that more information is needed in the method section regarding CBT as it was specifically applied at this site.

Line 783: And to CBT.

Line 779: Conclusions section. Much of the information contained in this section is a direct recap of the preceding results and discussions. The length of this section could be significantly reduced.

---

## Referee Comment (RC2) · Anonymous Referee #2 · 18 Dec 2020

The paper presents a 2.5-year long dataset of dry deposition of total reactive nitrogen (Nr) to a forest site, interpretation of the results in the light of measurements of Nr speciation, and a comparison of the results with alternative approaches: the prediction of a chemistry and transport model, a site-specific inferential model and a canopy budget technique. Direct measurements of Nr dry deposition is rare and such a long dataset of Nr dry deposition measurements to forest is unique and important, and thus generally publishable in Biogeosciences. I had high hopes for this paper, especially because the Nr flux measurements were accompanied by NH3 flux measurements (by QCL), which I hoped would have been used to elucidate the non-NH3 component of the Nr flux. However, I was let down in various aspects: the NH3 fluxes are not used in this

paper (only concentrations). It is not stated whether they just did not work or whether they are left for another paper. However, this paper speculates a lot about the nature of the NH3 exchange and its impact on the total Nr flux and with NH3 flux data presumably available to explore this explicitly, this seems rather odd. In addition, the Discussion section is quite long and lacks structure and aim. The advantage of the TRANC is that it captures most of the Nr flux with one instrument. The disadvantage is that it does not shed light on the behaviour of the individual Nr components. Yet, much of the discussion is dedicated to relating the measured flux to the behaviour of individual compounds reported in the literature. I do not think this adds to the manuscript and should be shortened. Instead the paper should be more focussed on describing the flux in its totality. For example, the Nr dry deposition budget is not discussed in the context of the additional wet deposition which could be taken either from nearby measurements (if available) or the LOTOS-EUROS prediction. A number of serious concerns need to be addressed as raised below before the manuscript can be accepted for publication. This will require significant reworking and refocussing of the manuscript.

Main scientific comments:

1. As mentioned above, if the NH3 fluxes could be worked into the manuscript this would strengthen the analysis a lot.

2. The paper confuses the rate of deposition (deposition velocity) and the actual deposition. Ignoring the effects of compensation points on NH3 exchange and the contribution of soil NO emissions to the net flux of NO and NO2, and also changes in the relative contribution of different compounds to Nr, the deposition of Nr is expected to scale approximately with its concentration. This is trivial and fundamentally also the way the deposition is calculated in LOTOS-EUROS and DEPAC-1D. Changes in concentration therefore mask the mechanisms that regulate the deposition rate. Thus, the analysis would be much more meaningful if the authors examined the controls of the deposition velocity rather than of the flux. This is what is done in the literature for the different compounds and, currently, comparisons are not correct. For example, it is

stated that NH3 fluxes are largest under wet conditions. In fact most studies report that Vd is larger for wet conditions, but at the same time the concentration may be reduced. For this reason statements like "dry conditions seem to favour nitrogen dry deposition (line 13, also line 793f)" are maybe not incorrect, but certainly misleading. Throughout the analysis it is rarely clear whether an association between the flux and drivers is due to their effect on concentration or Vd. For example, Fig. 4 would be more meaningful if presented for Vd. In fact, an analysis in terms of Rc would be even more meaningful as it would normalise for the effect of turbulence on Ra and Rb both of which contribute to Vd. Because particles are not really subject to a boundary-layer resistance in the way it is applied to gases, Rc is not really meaningful. However, the analysis could be done in terms of $Vds = Vd(z0)$, i.e. after normalising at least for Ra.

3. The interpretation of the measurements is not helped by the lack of showing absolute concentrations. The relative composition of total Nr (Figs. B1 and E1) is useful, but also the absolute concentrations are needed to interpret the results. Again, because fluxes are discussed in terms of their magnitude and not their Vd the reader is left wondering whether whether it is really the change in relative composition that changes the flux or whether it is just the overall Nr concentration. By the way, it is unclear what time periods are shown by each pie chart and what frequency this maximum refers to (Caption and text Line 305ff). Presumably, these are monthly results given that the lowest data resolution (from the DELTA) is monthly? Indeed, I would find a figure showing monthly stacked bar graphs of the individual Nr components very useful. This would convey how the total and their contribution to total Nr changed seasonally. Also, an assessment of how well the sum of the individual N compounds compares with the total Nr concentration needs to be added as quality control.

4. The measurements are compared to those made over other ecosystems and differences are explained by differences in ecosystems. Again, this is only part of the story, mainly the part that affects Vd. The pollution climate the ecosystem is in is equally important and does not necessarily correlate with the ecosystem type (think of an urban

woodland or a heavily grazed pasture in otherwise pristine environment). The comparison needs to be reworded. Generalisation that Nr fluxes always behave above natural vegetation as they do at this particular site is not tenable (e.g. line 13 and throughout).

5. The analysis of the effect of precipitation on the flux (Fig. F1a and associated text) is particularly problematic. During rain the eddy-covariance flux measurement of water soluble compounds (and many Nr compounds are) is highly uncertain because fluxes cannot be assumed to be constant with height due to the washout process. An increased Vd during rain may just reflect the presence of an additional sink (the washout process) below the measurement height. Rain episodes should potentially be filtered out, but certainly no process understanding should be derived from data taken during rain. How do the measurements demonstrate that wet deposition is important (Line 595)?

6. The paper does not distinguish different types of error (e.g. lines 617f and 652f). The flux error according to Finkelstein and Sims describes a random error, whereas the uncertainty in the DEPAC-1D estimate is more likely to be systematic and thus provide a bias. The input parameters are considered the largest uncertainty in DEPAC-1D (lines 619f), but actually different inferential models give very different results which highlights their uncertainty (e.g. Flechard et al., 2011).

7. This then also relates to an apparent contradiction between the discussion of the importance of stomatal exchange (Line 575) which is temperature dependent but mainly regulated by PAR and the statement that the canopy resistance is mainly driven by water solubility (Line 702).

8. Still on the topic of drivers of the exchange, a similarity in the diurnal cycle between global radiation and flux is no proof of causality (line 549ff). A lot of parameters are driven by the radiation: turbulence, photochemistry etc.. Neural networks also do not derive causalities or 'drivers', only associations and determinants.

9. The filtering criteria will have removed preferentially the smaller fluxes during low

turbulence conditions and the remaining dataset will therefore be biased. Whilst this is not an issue if a model is used for gap filling that accounts for changes in turbulence, it does impact the straight averages of the fluxes (Figure 2) the value of which then becomes questionable and also the MDV gap filling method. These issues and implications need to be discussed.

10. The use of monthly mean concentrations for some of the compounds (DELTA measurements) adds significant uncertainty to DEPAC-1D model results. The first mention that the DELTA measurements are monthly seems to come in line 303 and the uncertainties are not mentioned until Line 622 (and there without references to, e.g., Schrader et al. 2018). The limitations of this approach should be more visible earlier on. Was the gap-filling of NH3 (Line 257) done in a mass-conserved way, i.e. was the available data removed from the long-term NH3 average to work out what the average concentration during the gaps might have been? I suppose this would lower the uncertainty somewhat? Was a diurnal cycle superimposed on the long-temporal resolution measurements?

11. I do not follow the introduction of the DEPAC algorithm (Section 2.4.1). Erisman et al. (1994) does not describe a bidirectional resistance model (Line 224). Similarly, the references in lines 230-231 all describe deposition parameterisations, but most are almost certainly not the ones used in this version of DEPAC and contradict each other. The most correct description probably comes in Lines 243-247. Much of the description of the DEPAC-1D (Section 2.4.3), including the resistance parameterisations, probably also apply to the DEPAC version implemented in LOTOS-EUROS? It is all a little confusing. I did not realise until the Discussion section that DEPAC-1D does not treat the aerosol. This is a major and seemingly unnecessary shortcoming. My understanding was that DEPAC-1D is a stand-alone version of the deposition scheme implemented in LOTOS-EUROS and surely the latter treats the aerosol components. This seems hardly justifiable.

12. I am confused throughout about the use of a compensation point for NH3 in the

versions of LOTOS-EUROS and DEPAC-1D used. What is its magnitude for the forest types under consideration and where does it come from? Line 264 says that the DELTA concentrations were used for determining compensation points and additional deposition corrections? How was this done? Does this mean the models were not run with the standard scheme for these ecosystem types? Monthly concentrations do not lend themselves to deriving compensation points. Lines 671ff discuss uncertainties around cuticular compensation points. This would suggest that this was somehow adjusted based on the measurements?

13. Given all this discussion about compensation points it is then highly surprising that Vd for HNO3 and NH3 are virtually identical (Line 374). How can this be? Apart from potential of evaporating NH4NO3 on leaf surfaces, HNO3 exchange is well understood and follows a near-zero Rc. NH3 does not.

14. I am similarly unclear about the discussion of the landcover (Lines 236-242). Given the resolution of LOTOS-EUROS of 7 x 7 km2 it is not surprising that the landcover of the grid cell containing the measurement site does not match that of the flux footprint which is much smaller. But I also do not see a big problem: is LOTOS-EUROS not based on a mosaic / tiling approach and predict fluxes to each landcover type separately? The associated description of the LAI values (Lines 273-279) is also unclear. Surely DEPAC-1D and LOTUS-EUROS simulate the deposition to all landuse types in a gridcell and from those a landcover-weighted average can then be calculated? In general, it should be made clearer what is identical and what is different between the LOTOS-EUROS and the DEPAC-1D simulation. What measurements were used for DEPAC-1D? Concentrations, meteorological parameters, canopy characteristics?

15. The December emission fluxes are insufficiently explained. Were temperatures really sufficiently high to drive NH3 emissions from decomposition (Line 489)? Is there any evidence of freeze-thaw cycles affecting NH3 fluxes (Line 496)? Possibly, freeze-thaw cycle effects on soil NO are a more likely explanation? However, does the flux direction actually correlate with freeze-thaw events? Could it be caused by a problem

with the measurement setup for a period of time given that December measurements differed between the two years?

Minor scientific comments

The abstract seems overly long and should be shorted. This can be done linguistically (e.g. remove phrases such as "We further showed that") and in terms of content. For example, it is sufficient to list the results in terms of annual deposition inputs and remove the numbers for the 2.5-year timeframe (line 19ff).

In Section 2.2 I am missing a fuller statement on the response of the TRANC to Nr compounds in the aerosol phase. What is the size-cut? What is the response to nitrate other than ammonium nitrate (e.g. sodium nitrate, calcium nitrate, . . .)? Presumably they are not volatilised?

Line 33ff. I am not aware that deposition of Nr components threatens human health. They do so by acting as precursors to PM2.5 and O3.

Line 80f. The critique of the MDS method is difficult to understand because it is not explained what it is. The introduction of CTM approaches is a little messy. Line 90 explains their workings by needing meteorological data and land-use information. Emissions and chemistry are only mentioned much further down.

The introduction of the principle of operation of the TRANC is also not very logical. First reduced N is oxidised and then NH3 is formed from NH4NO3? Surely this happens before the oxidation (or in the same step).

The description of turning the leaf wetness value into a boolean value needs to be improved (line 158ff). At present, a value of 10 in arbitrary units is meaningless.

Line 166ff. Please state the temporal resolution of the DELTA measurements. Also, later the text refers to ammonia diffusion samplers and NOx measurements, which do not appear to be mentioned in Section 2.2.

[Figure]

Line 199. Does the flux loss depend on the chemical composition of Nr?

Line 207. Please state the relative magnitude of the water correction. What is its uncertainty?

Line 211. Removal of fluxes outside a certain range appears to be arbitrary and subjective. Are these extreme fluxes not caught by the other tests, e.g. Foken's stationarity test or testing for stochastic significance via the random flux error? I presume the latter is what the "threshold of two times 1.96sigma" (Line 213) refers to? Currently, sigma is not defined and its calculation remains unexplained.

Line 264f. How were compensation points derived from long-term measurements of SO2 and NH3? This would seem problematic.

Line 266. Why was the LAI modelled for a site-based application? Why was this not based on a measured value?

Line 390. How do the diurnal cycles compare between measurements and model results? Does this shed add additional light on model deficiencies?

Line 434. No, concentration is not proportional flux. The flux is proportional to the concentration. The concentration is the driver.

Line 468. What do the concentration ranges refer to?

Line 501. Both NO and NO2 contribute to Nr. So even if soil NO is converted to NO2 it will still contribute to the Nr flux except for the fraction that is removed by the canopy.

Line 507. The DELTA samplers does not measure NOx.

Line 514. There is a range of coatings available for the DELTA denuders. Clarify here and possibly also in the Methods section that carbonate coating was indeed used.

Line 551. Presumably in addition to total Nr concentration, its speciation also affects the net deposition rate and thus the flux.

Line 721. Is it worth adding DELTA, QCL and passive sampler data all to the graph to have an intercomparison between measurements? How do HNO3 compare between model and measurement? The modelled values of NH3 could also be too high because HNO3 in the model is too low (thus forming less NH4NO3).

Line 739. The model presumably calculates u* from the ascribed canopy height and does not know about the complexity of the terrain. Are you saying that the measured u* is elevated because of topography? Would this not imply that the conditions for eddy-covariance are not met?

Line 754. "input NH3 concentrations" Do you refer to emissions or long-range transport?

Line 763. If the deposition event wasn't measured it maybe did not exist. I suggest to rephrase: "All models predicted at 2nd emission event which was not confirmed by the measurements."

Line 793f. But you say the Vd of NH3 is very high almost as high as HNO3. Thus, a large relative contribution of NH3 should give you large deposition fluxes.

Line 795f. The wash-out could have occurred upwind and not contributed to the local wet deposition.

Line 798f. The good agreement seems entirely fortuitous given aerosol was not included in DEPAC-1D . . .

Line 803f. Maybe the gap filling methods are designed for compounds whose fluxes are actively regulated by production and consumption processes rather than the consequence of turbulence and concentrations such as deposition.

Technical corrections / suggestions:

The English needs to be improved throughout the manuscript, especially the use of articles. The list of corrections suggested here is not complete and since major changes

to the text will follow when addressing the points above it should be checked again anyway.

General: avoid starting sentences with numbers. E.g. line 23 could better read "Deposition of 16.8 kg N ha-1 was calculated"

General: there are numerous places where an article is missing. E.g.

- line 86: "due to the low number"

- Line 146: "as a reducing agent"

- Line 179: "on an annual basis"

General: there are several instances where the word "after" seems to be a mistranslation from German and needs to be replaced.

- Line 105: "were taken following the approaches of the International . . ."

- Line 108: " nitrogen deposition using the canopy budget technique"

- Line 179: "bases following the CBT approach"

General: in many cases units are incorrectly combined. For example ms-1 should read m s-1 and $\mu$gm-3 should read $\mu$g m-3.

Line 7. I was surprised to see Nr concentration given in ppb rather than $\mu$g N m-3, especially since Nr contains aerosol compounds for which the use of ppb is rather unusual.

Line 62. Better "EC studies of . . ."

Line 69 refers to "that site", but it is not clear which site is meant at this point.

Line 96. "validation with flux measurements" (or "against").

Line 116. "Measurements were carried out in". Actually, the authors should consider the alternative "Measurements were made" here and elsewhere.

Line 117. Remove "and".

Line 130. Remove "which is remote from significant sources of emissions." This is repeating what was said before.

Line 139. "which was housed in an"

Line 142. "oxidation"

Line 145. "during which remaining oxidised Nr species"

Line 219. "was caused by"

Line 249. "filling the gaps in the flux data."

Line 274. "weighted using the actual land-use fractions" ?

Line 275. "when considering only deciduous"

Section 3.1. Much of the section here and elsewhere should be put into past tense.

Line 303 and elsewhere. Please add charges to NO3- and NH4+ (NO3 is a radical).

Line 305. Redundant "with"

Line 308. "the relative contribution of NH3 is significantly higher"

Line 310 and elsewhere. A colon is followed by lower case in English.

Line 311 "done following the criteria mentioned"

Line 380 & 447. Should be "consequently" instead of "consequentially"

Line 384. Should the units here be "kg N ha-1 a-1"?

Line 391. "Clearly, . . ."

Figure 6. The colours between upper and lower CBT estimate seem to be reversed.

Line 417 and also line 816. "the range of . . ."
Line 450. "LOTOS-EUROS states out NH3 . . ." – meaning unclear.

Line 479. "Apart from management events, fluxes above the arable . . ."

Line 528. "Munger et al. (1995) also made NOy flux measurements . . ."

Line 607. "sometimes lead to non-stationarities"

Line 612 "under certain circumstances"

Conclusions. Re-introduce all acronyms, including Nr.

---

## Referee Comment (RC3) · Anonymous Referee #3 · 21 Jan 2021

Wintjen et al. present an interesting and valuable data set on total nitrogen deposition to a forest spanning multiple years. The paper will be a worthy addition to N deposition literature, but would be improved by providing a few additional details and considering some additional analysis and interpretation.

Page: 8 line 252-254.It would be helpful to provide a little more detail on the calculation of resistances beyond just giving a reference.The actual equation itself would be ideal, but at least note what input variables are used in the parameterizations so that readers can know what the calculations are based on without having to consult multiple sources from the literature.

[Figure]

line 257. Here it notes that alternate data sources are used for missing NH3 and HNO3. Is it stated anywhere how the data sources compare to one another when there are simultaneous measurements? Readers need this to assess whether there is any bias in the gap filling? Showing or mentioning a direct comparison would complement the plots showing cumulative deposition computed from different approaches. The direct comparison of simultaneous concentrations removes any confounding influence of other inputs to the calculated fluxes

Page: 23 Line 449.Here it concludes that radiation is the primary driver affecting the diel cycle of N deposition. How have you discounted the role of wind speed/turbulence intensity, which will covary to radiation, as an alternative? If you account for the turbulence contribution to deposition velocity based on resistance model and thus compute an apparent canopy resistance from the residual is there still a dependence on radiation?

Page: 24 line 574 Do you consider the role of humidity and temperature on the partitioning between gaseous NH3 and NH4 aerosol? The patterns imposed by stomatal opening and NH3 partitioning might be difficult to distinguish. The observed pattern would be consistent with shifting the equilibrium toward gaseous NH3 during the warm and dry daytime conditions.

---

## Author Comment (AC1) · 3 Feb 2021

**Author comment to reviewers' comments - manuscript *BG-2020-364*
"Forest-atmosphere exchange of reactive nitrogen in a low polluted area
– temporal dynamics and annual budgets"**

We thank the reviewers for their constructive comments. We appreciate their suggestions and agree that including analyses about deposition velocities and canopy resistance brings further insight into the exchange patterns of reactive nitrogen species. We further agree that a better streamlining of the discussion will improve

readability and quality of the paper. After carefully reflecting on the given suggestions and how these could be implemented in the best way given the fact that the current version is already relatively long, we decided to split the manuscript into two parts.

The first part will address the TRANC measurements, the second part will be focusing on the modeling of reactive nitrogen dry deposition. Therefore, a revised version of the submitted preprint will solely concentrate on the evaluation of the TRANC measurements. We will extend our analysis and will show diurnal cycles of both deposition velocities and canopy resistances stratified by meteorological drivers and concentration, as was done in the former Figure 4. The discussion will be updated accordingly. We will further show the impact of flux filters and different statistical gap-filling approaches on the dry deposition budget. A point-by-point response will be provided to keep track of the changes made and how the individual comments were taken into account.

The second part will deal with the nitrogen deposition modeling and will be submitted to BG as a separate manuscript at a later stage. We plan to draw comparisons of measured and modeled deposition velocities and, similar to the measurement part, investigate the dependencies of modeled total reactive nitrogen dry deposition velocities on micrometeorology. We will include wet deposition results from model calculations to discuss the total nitrogen deposition. Comments exclusively dealing with modeling aspects will be taken into account in the second paper. We hope that our strategy for a revised version will be supported by the editor and the reviewers.

---

## Editor Comment (EC1) · Ivonne Trebs (Editor) · 4 Feb 2021

Dear authors,

I do acknowledge your decision on splitting the manuscript into two parts. However, for further processing of this paper it is required that you please provide a point-by-point response to the referee comments. Please also indicate in more detail which issues raised by the referees will be addressed in the second paper.

---

## Author Comment (AC2) · 16 Mar 2021

Please find attached PDF file containing our responses to all comments and suggestions from the three reviewers.

Please also note the supplement to this comment:
https://bg.copernicus.org/preprints/bg-2020-364/bg-2020-364-AC2-supplement.pdf

---

## Author Response (AR1)

**Response to reviewers' comments - manuscript *BG-2020-364* "Forest-atmosphere exchange of reactive nitrogen in a low polluted area – temporal dynamics and annual budgets"**

We thank the reviewers for their constructive comments. As outlined in the author comment, comments to measurement part of the manuscript are answered in this review. Comments to the modeling part are highlighted in red. Referee comments are given in italic, the answers in standard font. The comments by Reviewer 1 are numbered from R1.1 to R1.81 titled as specific comments. The main scientific comments of Reviewer 2 range from R2.1 to R2.15, the additional comments start at R2.16 and end at R2.42, and the comment from R2.43 to 2.76 refer to technical corrections/suggestions. Comments of Reviewer 3 range from R3.1 to R3.5. The line and figure numbers in the answers, where we add the new information into the manuscript, refer in this document to the originally submitted version. The text which is enclosed by "..." is implemented in the revised manuscript.

**Response to Reviewer 1**

**General Comments** *This manuscript presents 2.5 years of measurements of total reactive nitrogen (Nr) fluxes above a mixed forest in Germany. The measurements are used to assess annual dry deposition budgets and are then compared to deposition estimates derived from a field scale model and a gridded chemical transport model. This study directly addresses the need for new Nr flux measurements to improve Nr deposition budgets, assess exceedances of critical loads of Nr, and improve models of reactive N deposition. The dataset developed is novel and should prove useful to the ecological and atmospheric chemistry communities interested in N deposition. Furthermore, the interpretation of the measurements in relation to micrometeorology and atmospheric chemistry sheds new light on the processes influencing air-surface exchange of Nr and the relative importance of Nr species to the dry Nr deposition budget. However, there are a number of technical details of the analysis and discussion, along with some organizational issues, that should be addressed before the paper is suitable for publication. In general, the paper would benefit from a more thorough quantitative analysis of the flux patterns and their relation to micrometeorology and atmospheric chemistry. Section 4.2 touches on these relationships but could be extended along the lines of several suggestions outlined below. As also suggested below, the current content of Section 4.2 could be reorganized and shortened by eliminating some redundancies, making it possible to expand the analysis without significantly lengthening the Section overall. Sections 4.3.2 and 4.3.3, which describe uncertainties in the modeling approaches, as well as the Conclusions section, could be significantly reduced in length. More specific comments are detailed below.*

We thank the Reviewer for his/her comments on this work. As mentioned in the author comment, we split the preprint into two parts. We appreciate your suggestions to the discussion of the measurements. We included a comparison of the TRANC measurements to results from DELTA samplers and investigated the effect of micrometeorology on the deposition velocity and resistances of $\Sigma N_r$. As outlined in the author comment, discussions on the modeling results will be shifted to the modeling manuscript. Questions related to the nitrogen modeling will not answered in this response in detail. Your suggestions will be included in the preparation of the modeling manuscript.

**Specific comments**

**Comment R1.1** *Line 83: Change "pattern" to "patterns".*
**Response to R1.1** Revised.

**Comment R1.2** *Line 90: Should the first word be "methods"?*
**Response to R1.2** Yes.

**Comment R1.3** *Line 94: CTM should be plural.*
**Response to R1.3** Revised.

**Comment R1.4** *Line 117: "site located" should be "site is located"*
**Response to R1.4** Done.

**Comment R1.5** *Line 145: For clarity, consider rewording this sentence to something like: "In a 2nd step, a gold tube passively heated to 300C catalytically converts the remaining oxidized Nr species to NO."*

**Response to R1.5** We reworded the sentence according to your suggestion.

**Comment R1.6** *Line 148: Is 2.1 L/min the flow rate through the converters (atmospheric pressure) or through the reduced pressure portion of the tubing downstream of the orifice? If the latter, please indicate the flow rate through the converters.*
**Response to R1.6** $2.1\,\mathrm{L}\;\mathrm{min}^{-1}$ was the flow rate after the critical orifice. The following lines were added to line 148: "The mass flow rate before the critical orifice was the same as after the critical orifice. Since mass flow was equal to both sides of the critical orifice, a difference in flow velocity was induced due to the reduction in pressure. Flow velocities were not measured for the different sections."

**Comment R1.7** *Line 164: What type of passive sampler was used? What was the sampling duration?*
**Response to R1.7** "Passive samplers of the IVL type (Ferm, 1991) were used for $NH_3$, and the exposition duration was approximately one month at a time". The information given here was added to line 164.

**Comment R1.8** *Line 207: What was the typical magnitude of this correction to the total Nr flux?*
**Response to R1.8** "The correction contributed approximately $132\,\mathrm{g}$ N $\mathrm{ha}^{-1}$ to two years of TRANC flux measurements if the Mean-Diurnal-Variation (MDV) approach was used as gap-filling approach. Half-hourly interference fluxes were between -3 and $+0.3\,\mathrm{ng}$ N $\mathrm{m}^{-2}$ $\mathrm{s}^{-1}$. Their random flux uncertainty ranged between 0.0 and $0.5\,\mathrm{ng}$ N $\mathrm{m}^{-2}$ $\mathrm{s}^{-1}$" We added the information given in this response to line 209.

**Comment R1.9** *Line 216: What caused the reduced sensitivity of the CLD and how was it identified?*
**Response to R1.9** "The reduction in sensitivity may be caused by reduced pump performance leading to an increase in sample cell pressure. If pressure in the sampling cell is outside the regular operating range, low pressure conditions needed for the detection of photons emitted by excited $NO_2$ molecules may not hold. Pump efficiency was controlled at least monthly, and tip seals were replaced if necessary. The sensitivity of the CLD could also be reduced by changes in the $O_2$ supply from gas tanks to ambient, dried box air if $O_2$ gas tanks were empty. Issues in the air-conditioning system of the box could also affect the sensitivity of the CLD. An influence of aging on the inlet, tubes, and filters may also affect the measurements. In order to minimize an impact on the measurements, half-hourly raw concentrations were carefully checked for irregularities like spikes or drop-outs by visual screening." We added the information given in this response to line 216.

**Comment R1.10** *Line 266: How was the quality of the DELTA measurements assessed?*
**Response to R1.10** "The denuder preparation and subsequent analyzing of the probes was identical to the procedure for KAPS denuders (Kananaskis Atmospheric Pollutant Sampler, (Peake, 1985; Peake and Legge, 1987)) given in Dämmgen et al. (2010) and Hurkuck et al. (2014). We controlled the pump flow to keep it at a constant level and checked the pipes for contamination effects before analyzing. Blank values were used as additional quality control." We added the description of the DELTA measurements to line 165.

**Comment R1.11** *Line 275: Has LAI been measured at this site? How variable is the LAI throughout the seasons, given the relative fractions of spruce and beech.*
**Response to R1.11** The leaf area index (LAI) was not measured at the site. It was modeled after the same scheme used for DEPAC (see Appendix B of van Zanten et al., 2010). A linear increase of the LAI was modeled from mid of April to begin of May, a linear decrease from October to begin of November. Values ranged between 4.1 and 4.8. Fig. R1 shows the modeled LAI for measured land-use classes.

[Figure]

Figure R1: Modeled LAI following van Zanten et al. (2010) for measured fractions of coniferous forest (81.1%) and deciduous forest (18.9%) within the flux foot print for a year.

Figure R1 was added to Sec 2.4 "Determining deposition velocity and canopy resistance of $\Sigma N_r$ from measurements".

**Comment R1.12** *Line 285: Should be "gaps in micrometeorological".*
**Response to R1.12** Yes, you are right.

**Comment R1.13** *Figure 1: The blue bar in the lower plot is incorrectly labeled NH3.*
**Response to R1.13** Corrected.

**Comment R1.14** *Line 296: The findings here relative to concentrations of NOx > 20ppb make me question the description of this site as being situated in a "low pollution" area. Some additional justification of this site characterization is needed.*
**Response to R1.14** We characterized the site as "low polluted" since average concentration level of $NO_x$ was comparatively low (see Fig. 1). $NO_x$ peaks above $10 \, \mu g \, N \, m^{-3}$ were observed only for short time periods during winter.

**Comment R1.15** *Line 302: The figure numbering configuration for the Appendices (e.g., Figure B1) was not immediately clear to me. I believe the format for Biogeosciences is for such material to be included as "Supplemental Material".*
**Response to R1.15** We agree that some figures are more suitable for "Supplemental Material". For the revised version, we prepared a supplemental file. Figure A1 of the appendix was moved to the supplemental file. Other figures of the appendix were deleted.

**Comment R1.16** *Line 303: How does the sum of the concentrations measured by the DELTA compare to the TRANC Nr measurement?*
**Response to R1.16** According to the suggestion of Reviewer 2, we added a stacked bar graph (Fig. R9) showing monthly concentrations of the DELTA measurements compared to the TRANC $\Sigma N_r$ concentrations. Latter were averaged to the exposition period of the DELTA samplers. The

comparison revealed significant underestimations of TRANC $\Sigma N_r$ from March to mid of May 2018 and from July to mid of August 2017. We found that the zero-air calibration value of the TRANC-CLD system was incorrect for the mentioned time periods by approximately $0.9\,\mu g$ N m$^{-3}$ compared to the uncorrected TRANC-CLD concentrations. Concentrations and fluxes were recalculated with the bias correction. Figures and evaluations shown in the response and in the revised manuscript were made with the bias-corrected data. Figures of the manuscript were updated accordingly. On average, the TRANC values were slightly higher by $0.3\,\mu g$ N m$^{-3}$ than DELTA+NO$_x$. The results of this comparison were added to line 302.

**Comment R1.17** *Line 317: What fraction of the non gap-filled half-hourly fluxes exceeded the flux detection limit?*
**Response to R1.17** The following sentences were added to line 318 "In total, 51% of the non gap-filled fluxes were higher than the flux detection limit. It shows that for large parts nitrogen dry deposition was close to detection limit of the used measuring device and that nitrogen exchange happened at a comparatively low level." Despite the low signal-to-noise ratio at the measurement site, we were able to investigate the exchange pattern of $\Sigma N_r$ and could estimate reliable dry deposition sums (see R2.9).

**Comment R1.18** *Line 350: Should "based on" be "are based on"?*
**Response to R1.18** Yes, you are right. Please note that sentence was deleted.

**Comment R1.19** *Line 352: I might expect the sensor nearest the ground to remain "wet" later into the morning than the sensor closest to the top of the canopy.*
**Response to R1.19** The statement made in line 352 was misleading. As an example, Fig. R2 shows diurnal patterns of the leaf wetness for all sensors on monthly basis for 2017. Since no difference was found between the spruce and beech tree, colors were chosen to highlight a potential effect of the measurement heights on the leaf surface wetness.

[Figure]

Figure R2: Daily cycles of the leaf wetness for 2017. Colors indicate installation heights of the sensors (red=top, green=middle, blue=bottom). Shaded areas represent the standard error of the mean.

Figure R2 shows diurnal patterns of the leaf wetness for all sensors on monthly basis for 2017. On monthly basis, the diurnal patterns of the sensors were almost the same for a season. From April, the start of the growing season, to September highest values were measured during dawn and lowest values during the day. During daylight, only slight differences in measurement height were visible. Considering the standard error, the differences in measurement heights diminished, especially between the lowest and middle sensor. Also, sensors from the mid and the top were within their uncertainty ranges. In conclusion, sensors at the lowest height seem to remain "wet" later during the morning, but effect is within the standard error range. Using only the top sensors for deriving the leaf wetness value, seems not to be appropriate with regard to the uncertainty ranges. Thus, we used all sensors for deriving a wetness boolean, which also lowered its uncertainty. Figure R2 and corresponding description were provided as supplemental material.

**Comment R1.20** *Line 398: "It seems that....most likely driven by particulate Nr compounds." Is this supported by the particulate measurements? Do the DELTA measurements show relatively higher concentrations of particulate NH4NO3 during this period? Given the lower Vd of particles relative to gases, the concentrations would need to be much higher to drive the high total N*

*deposition during this period, correct?*

**Response to R1.20** Unfortunately, we had no DELTA measurements during this period since the pump was not working properly. However, we found that $SO_2$ concentrations were remarkably high during the deposition period in February 2018. Passive sampler measurements showed a low $NH_3$ concentration level in February 2018. Presumably, ammonium sulfate or compounds formed at lower $NH_3$ concentrations, e.g., ammonium bisulfate were responsible for the observed $\Sigma N_r$ deposition. Please see Sec. 4.1 of the revised version for further details.

**Comment R1.21** *Lines 404 and 405: So a linear interpolation is used? Please clarify.*

**Response to R1.21** The deposition of the first half of 2018 was linearly interpolated to the end of 2018. As a second approach, we calculated the average deposition from the second half of 2016 and 2017 and used their average as assumption for the second half of 2018. Due to the deposition event in February 2018, the TRANC deposition estimated with MDV was significantly higher in case of linear interpolation. We will add this aspect to the second part.

**Comment R1.22** *Line 411: "..significantly higher lower and upper.." I understand what you mean here but it is a little confusing. Consider rewording for clarity.*

**Response to R1.22** Agreed. We will reword it.

**Comment R1.23** *Line 425: "4.6 kg N/ha/a are determined as a lower estimate." Please clarify how this estimate was determined.*

**Response to R1.23** Probably, you are referring to line 415. The given values are averages of the lower and upper canopy budget technique (CBT) estimates from 2016 to 2018. We will add it as explanation and think about more proper names for the calculated values.

**Comment R1.24** *Line 426: Should this section heading read "Sensitivity of deposition estimates to measured vs. modeled input parameters"?*

**Response to R1.24** We agree. We will change the section header within the preparation of the modeling study.

**Comment R1.25** *Line 428: Remove comma after "class" and add "the" after "considering".*
**Response to R1.25** Revised.

**Comment R1.26** *Line 432: Specify that you are referring to the apoplastic ratio of NH4+ to H+. I would also suggest you clarify that you are referring to the stomatal compensation point in the latter part of this sentence. Have any measurements of the soil and vegetation chemistry been conducted at this site such that compensation points could be estimated?*

**Response to R1.26** We appreciate the Reviewer's suggestion. Unfortunately, no measurements of soil and vegetation chemistry had been conducted at the site.

**Comment R1.27** *Line 435: "...global radiation enhances the opening of the width of the stomata" It may be more straightforward to say that the stomatal resistance is influenced by global radiation.*
**Response to R1.27** Agreed.

**Comment R1.28** *Line 439: I would suggest that reporting the bias (absolute percent) in the modeled values relative to the measured values is more informative than the correlation in this context.*

**Response to R1.28** We agree and will modify the description of the figure within the modeling part.

**Comment R1.29** *Lines 450 - 455: A table comparing the measured Nr species concentrations (Delta compounds, NOx, QCL NH3, passive NH3) to LOTOS-EUROS would help clarify this section.*

**Response to R1.29** We will add a stacked bar graph to similar to Fig. R9 but with the $\Sigma N_r$ concentrations from LOTOS-EUROS to the modeling manuscript. A time series showing monthly averages of $NH_3$ from the QCL, passive samplers, and LOTOS-EUROS is also planned. Please also note Fig. R13 showing a comparison of the different $NH_3$ measurement techniques.

**Comment R1.30** *Line 458: "... are very low compared to other studies" This statement is true relative to the three references cited but perhaps not so for Nr flux studies in a global sense. Some additional context is required for this statement, e.g., low relative to sites influenced by agricultural activities, previous studies in European ecosystems, etc.*

**Response to R1.30** We agree that the interpretation of measured concentrations and fluxes was misleading (see also R2.4). We modified deleted "are very low compared to other sites" and replaced it by "low relative to sites exposed to agricultural activities or urban environments."

**Comment R1.31** *Line 464: Consider modifying sentence to "...higher ground-level concentrations...".*

**Response to R1.31** Revised.

**Comment R1.32** *Line 470: "Values..." This sentence seems incomplete.*

**Response to R1.32** We changed it to "Concentration values of $NH_3$ and $NO_x$...".

**Comment R1.33** *Line 472: "...confirm the seasonal pattern of Nr". Do you mean that those studies show patterns consistent with the current study?*

**Response to R1.33** The corresponding lines were deleted in the revised version and were replaced by the following sentence "Studies like Wyers and Erisman (1998); Horii et al. (2004); Wolff et al. (2010) conducted measurements of $NH_3$ and $NO_2$ above remote (mixed) forests and reported similar concentrations for those gases."

**Comment R1.34** *Line 473: "Obviously, measured concentration levels were significantly higher since the observed ecosystems were subject of agricultural management or in close proximity to industrial or agricultural emissions." Are the authors referring here to the studies listed in Line 471? At least for the Geddes study, NOx was lower than in the present study. Please clarify and correct this statement as needed.*

**Response to R1.34** We thank the Reviewer for his/her recommendation. Yes, it should refer to publications listed in line 471. We modified the sentence accordingly.

**Comment R1.35** *Line 477: This sentence should include references for the "few studies focusing on Nr".*

**Response to R1.35** Agreed. Namely Ammann et al. (2012), Brümmer et al. (2013), Zöll et al. (2019), and Wintjen et al. (2020) measured $\Sigma N_r$ fluxes with the eddy-covariance method. We added the references to the corresponding line.

**Comment R1.36** *Line 483: Please consider changing "their flux pattern" to "the flux pattern observed by Ammann et al. (2012)…".*
**Response to R1.36** Agreed.

**Comment R1.37** *Lines 488 – 497: The discussion of the high emission fluxes observed in December requires some additional detail and clarification. The authors refer to decomposition of fallen leaves beneath a snow layer. Are the authors suggesting that the decomposition is enhancing emissions of NH3 or NO or both? Decomposition rates typically decrease at low temperatures. The authors mention that they "discovered an increase in nitrogen concentration in the investigated samples". Samples of what? Soil? How were these samples taken and analyzed and for which compounds? How frequently were they collected and at what depths? How much did the N concentrations increase and over what time period? The statement regarding the influence of the freeze-thaw cycle on the emission fluxes is interesting but very speculative. Can a soluble gas like NH3 diffuse through a partially wet snow layer to the atmosphere? Do the fluxes correlate with air temperature in a pattern that would support this statement? Looking more closely at the December diurnal profiles in Figure 3 it appears the emission fluxes were mostly observed in 2017, which also had much higher variability in general than 2016. Were there more periods of snow cover in 2017? Did the two years differ in other ways in terms of meteorology or air concentrations that might help explain the emissions observed in 2017?*
**Response to R1.37** First of all, we thank the Reviewer for his/her suggestions to this paragraph and we agree that clarifications and more details are needed. We did not take leaf or soil samples at the site. "discovered an increase in nitrogen concentration in the investigated samples" referred to Taylor and Jones (1990). As suggested, we took a closer look in the temperature, concentration, and snow fall measurements during the emission period in December 2017 and compared them to the same period in December 2016. Figure R3 shows recorded temperature, snow fall, concentrations, and estimated fluxes of $\Sigma N_r$ from 6 December to 15 December for 2016 and 2017. We deleted lines 488-497 and replaced the description as follows:

[revised manuscript text omitted]

**Comment R1.38** *Line 508: Change "proposed" to "reported".*
**Response to R1.38** Please note that the sentence was deleted.

**Comment R1.39** *Line 511: Change "by DELTA" to "by the DELTA".*
**Response to R1.39** Please note that the sentence was deleted.

**Comment R1.40** *Line 528: Change "high" to "large".*
**Response to R1.40** Please note that the sentence was deleted.

**Comment R1.41** *Line 529: Change "at less" to "at a less".*
**Response to R1.41** Please note that the sentence was deleted.

**Comment R1.42** *Line 530: Change "It shows" to "These studies indicate"*
**Response to R1.42** Please note that the sentence was deleted.

**Comment R1.43** *Line 539: Please consider splitting up this long sentence for clarity.*
**Response to R1.43** Please note that the sentence was deleted.

**Comment R1.44** *Lines 544 – 547: The last two sentences of this paragraph seem more appropriate for the conclusions section.*
**Response to R1.44** In principle, we agree that those lines would be more suitable for a conclusion section. However, we decided to leave out from the conclusions of this study.

**Comment R1.45** *Section 4.2: In general this discussion would benefit from some reorganization and a more thorough quantitative evaluation of relationships between flux, micrometeorology, and air concentrations. The authors discuss radiation/photosynthesis, air concentration, dryness/RH/temperature, and precipitation as important variables. Perhaps these can be discussed in sequence, rather than jumping back and forth among them throughout the section, to make the discussion read more smoothly and to eliminate redundancies. For example, the role of air concentration is mentioned in numerous places, as are relative humidity and temperature. Some care should be given to revising this section as it will be of particular interest to readers seeking a better understanding of the processes driving Nr fluxes above forests.*
**Response to R1.45** We appreciate the Reviewers' suggestions to Sec. 4.2. After carefully reflecting the Reviewer comments related to that section, we agree that this section needs to be

improved in readability and content. As outlined in the author comment, a discussion of the deposition velocity ($v_\text{d}$), the aerodynamic resistance ($R_\text{a}$), the boundary-layer resistance ($R_\text{b}$), and the effective canopy resistance ($R_\text{c,eff}$) of $\Sigma N_\text{r}$ was added. The discussion related to the effect of precipitation on the $\Sigma N_\text{r}$ exchange was deleted. For better readability, we separated the section in three subsections in which we discussed the effect of (global) radiation on $\Sigma N_\text{r}$ exchange (Sec. 4.2.1), the effect of concentration $v_\text{d}$ (Sec. 4.2.2), and the seasonal changes in the uptake capacity of $\Sigma N_\text{r}$ (Sec. 4.2.3)

**Comment R1.46** *Line 548: Section heading 4.2 only mentions micrometeorology but much of the following discussion involves the relationship between flux and air concentrations. Consider rewording.*

**Response to R1.46** As the content of section changed, we adjusted the header as follows "Influence of micrometeorology and nitrogen concentrations on deposition and emission".

**Comment R1.47** *Line 549: What is the proposed mechanistic relationship between Nr flux and global radiation? What about the diurnal pattern of turbulent mixing and its role in air-surface exchange?*

**Response to R1.47** The following lines were added to line 551 of the manuscript: "As shown by Zöll et al. (2019), $\Sigma N_\text{r}$ and $CO_2$ fluxes exhibited a similar daily cycle and showed a strong dependence on $R_\text{g}$ during summer. The latter controls the opening of the stomata (Jarvis, 1976), i.e. lowers the stomatal resistance. Thus, photosynthesis controlling the $CO_2$ exchange through stomatal pathway appears to be the mechanism for controlling the $\Sigma N_\text{r}$ exchange as compounds like $NO_2$ (Thone et al., 1996) or $NH_3$ (Wyers and Erisman, 1998) are taken up by the stomatal pathway, too. However, $\Sigma N_\text{r}$ compounds are not willingly absorbed by the plants as seen by the light response curves of Zöll et al. (2019, Fig. 5). The light response curve of $\Sigma N_\text{r}$ has a reversal instead of a saturation point as observed for $CO_2$ (Zöll et al., 2019). Consequently, a second mechanism, the stomatal compensation point firstly proposed by Farquhar et al. (1980) likely controls the uptake of the $\Sigma N_\text{r}$ compounds. Basically, if the stomatal concentration is lower than the ambient concentration, deposition is observed. Thus, both parameters, the stomatal resistance and the stomatal compensation point, which are regulated by $R_\text{g}$ and concentration, respectively, affect the uptake of $\Sigma N_\text{r}$. As further shown by Zöll et al. (2019), other parameters like $u_*$ were not identified as important drivers for $\Sigma N_\text{r}$. Photochemistry and stomatal control appear to be more important than turbulent mixing. Radiation changes the composition of $\Sigma N_\text{r}$ due to the formation of $O_3$. In addition, $R_\text{g}$ had an influence on $u_*$ as seen by their similar shapes in daily cycle (Fig. R5 and R6). The low correlations of $\Sigma N_\text{r}$ fluxes to concentration for most of the selected $u_*$ ranges show that atmospheric turbulence had a generally low influence on nitrogen deposition at the measurement site. Thus, $u_*$ adds almost no additional information to the $\Sigma N_\text{r}$ exchange and was not identified as important controlling factor for the $\Sigma N_\text{r}$ exchange from July to September by Zöll et al. (2019). Similar conclusions can be drawn for temperature and relative humidity. They are also affected by light/energy input into the ecosystem and follow a similar diurnal pattern. It shows that $R_\text{g}$ contains most of the information for the explanation of the $\Sigma N_\text{r}$ fluxes."

It has to be noted that the study was conducted for $\Sigma N_\text{r}$ at the same natural, unmanaged site from July to September. Micrometeorological parameters were controlled by natural processes. The low response to micrometeorological parameters may also related to other processes influencing the composition of $\Sigma N_\text{r}$, to opposing effects on $N_\text{r}$ species, or effects happened on a shorter time scale such as molecular interactions between the $\Sigma N_\text{r}$ compounds. $R_\text{g}$ was not identified as

primary controlling factor for $NH_3$ by Milford et al. (2001). Milford et al. (2001) measured $NH_3$ fluxes above moorland, which has a generally higher humidity level than our measurement site. They concluded that $NH_3$ exchange is mostly driven by canopy temperature, wetness, and ambient concentrations. Radiation was not identified as primary controlling factor by the authors. They found higher deposition of $NH_3$ through the cuticular than through the stomatal pathway. However, Zöll et al. (2019) found only minor improvements in their driver analysis if water vapor pressure deficit was considered as secondary driver. Additionally, we found that $v_d$ was reduced for high ambient humidity and wet leaf surfaces. Since we measured $NH_3$ indirectly by the TRANC and above an ecosystem characterized by lower humidity than a peatland, $R_g$ favoring the exchange through the stomatal pathway appears to be more important for $\Sigma N_r$ at the measurement site.

**Comment R1.48** *Line 551: The authors discuss the relationship between air concentration and flux in several places in Section 4.2. Can the authors be a bit more quantitative in this analysis? What is the relationship (scatterplot) between concentration and flux if, for example, the dataset is filtered to include only mid-day fluxes (i.e., periods of high global radiation and friction velocity)? Is a clear relationship observed? What are the observed diurnal patterns in concentration? Do these patterns confound the relationship with global radiation mentioned in line 549? The authors should consider adding figures similar to figures 2 and 3 but for TRANC Nr concentration in supplemental material.*

**Response to R1.48** We thank the Reviewer for his/her suggestion. We added plots similar to Fig. 2 and 3 but for the $\Sigma N_r$ concentration to the supplement. A scatterplot showing the dependency of $\Sigma N_r$ concentration on corresponding fluxes was added to Sec. 3.2. The following text was placed after line 348: "For visualizing the effect of turbulence on the fluxes, Fig. R4 shows the dependency of the measured fluxes on their concentrations for different $u_*$ classes and global radiation ($R_g$) higher than $50\,W\,m^{-2}$.

[Figure]

Figure R4: Dependency of measured concentrations on corresponding $\Sigma N_r$ fluxes shown as scatter plots during daylight ($R_g > 50\,W\,m^{-2}$). Colors indicate different $u_*$ classes. Linear regressions between concentrations and fluxes are made for each $u_*$ class indicated by black lines.

We found a decreasing slope with increasing $u_*$. The slope corresponds to $v_d$. Results of the linear regressions, $v_d$ and squared correlations ($R^2$), are listed in Table R1. In addition, numbers

of half-hours used for the regressions are given.

Table R1: Results of linear regressions from Fig. R4 for selected $u_*$ ranges. The slope of the linear function corresponds to $v_\mathrm{d}$, $R^2$ is the squared correlation of concentrations and fluxes, and $n$ is the number of half-hours used for the regression.

| $u_*$ range [m s$^{-1}$] | $v_\mathrm{d}$ [cm s$^{-1}$] | $R^2$ [-] | $n$ [-] |
|---|---|---|---|
| 0.0–0.3 | 0.61 | 0.07 | 9085 |
| 0.3–0.6 | 0.63 | 0.05 | 6124 |
| 0.6–0.9 | 1.20 | 0.14 | 2296 |
| 0.9–1.2 | 2.16 | 0.28 | 485 |
| > 1.2 | 4.34 | 0.51 | 79 |

For $u_*$ values lower than $0.6\,\mathrm{m\ s^{-1}}$, $v_\mathrm{d}$ was almost invariant. For $u_*$ values higher than $0.6\,\mathrm{m}$ s$^{-1}$ or even higher, an increase in $v_\mathrm{d}$ was found. Since $R_\mathrm{a}$ (Garland, 1977) and $R_\mathrm{b}$ (Jensen and Hummelshøj, 1995, 1997) decrease with increasing $u_*$, $v_\mathrm{d}$ increases. The highest $R^2$ was determined for $u_*$ higher than $1.2\,\mathrm{m\ s^{-1}}$. For other $u_*$ ranges, correlations were negligible. However, only 79 half-hourly concentrations and fluxes were available for $u_*$ values higher than $1.2\,\mathrm{m\ s^{-1}}$. Considering the number of half-hours, atmospheric turbulence had an influence on the deposition of $\Sigma\mathrm{N_r}$ but $u_*$ could not be solely responsible for the observed exchange of $\Sigma\mathrm{N_r}$. "

Figure R5 shows the daily cycle of concentration, $\mathrm{R_g}$, $\mathrm{u_*}$, air temperature ($\mathrm{T_{air}}$), and $v_\mathrm{d}$ for the period from May to September. Figure R6 is made for the same variables but for December, January, and February.

[Figure]

Figure R5: Daily cycle of $\Sigma\mathrm{N_r}$ (black) concentration, $\mathrm{R_g}$ (green), $\mathrm{u_*}$ (olive), air temperature $\mathrm{T_{air}}$ (orange), and $\mathrm{v}_d$ (red) for the period from May to September. Shaded areas represent the standard error of the mean.

[Figure]

Figure R6: Daily cycle of $\Sigma N_r$ (black) concentration, $R_g$ (green), $u_*$ (olive), air temperature $T_{air}$ (orange), and $v_d$ (red) for the period from May to September. Shaded areas represent the standard error of the mean.

From May to September, a clear diurnal pattern in $v_d$ was observed with largest values around noon and lowest values during the night. During winter, $v_d$ was almost equal and even lower during the day, which resulted in lower deposition of $\Sigma N_r$ during winter. The different shapes of $v_d$ were related to plant activity mainly controlled by $R_g$. Figures R5 and R6 were added to the supplement.

**Comment R1.49** *Line 553: What do the authors mean by "favor" in this sentence?*
**Response to R1.49** The word "favor" is confusing in this sentence. "influence" may be a more proper word. However, we decided to delete the sentence.

**Comment R1.50** *Line 556: How is the last sentence in this paragraph justified by the preceding sentence? I must be missing something here.*
**Response to R1.50** We agree that the last sentence is misleading. Please note the second paragraph of R1.47. We corrected the sentence with the information given there.

**Comment R1.51** *Line 558: The authors compare March and April of 2017 and 2018 as an example of the potential role of photosynthesis in the interannual variability of fluxes. The explanation cites the role of temperature in stomatal function (and therefore the stomatal resistance) but what about the role of radiation? Are there differences in radiation between the two years that would also support this explanation?*
**Response to R1.51** For March 2017 and 2018, we could not examine the reason for the slight difference median deposition. Micrometeorological conditions of both months were comparable. "The higher nitrogen deposition in April 2017 (Fig. 5) compared to April 2018 was mainly related to gaps in flux time series. In 2018, we had no flux measurements from mid of April to the beginning of May. During that time, foliage began in 2018 providing uptake of $\Sigma N_r$ compounds. Increased plant activity was caused by continuously, high radiation values during daylight ($> 400\,\text{W m}^{-2}$) leading to higher temperatures in April 2018 ($\sim 11.0°C$) than in April 2017 ($\sim 6.0°C$). We further observed high $NH_3$ concentrations measured by passive samplers and the DELTA system for the

same time. Elevated $NH_3$ concentrations were likely caused by emissions from agricultural management in the surrounding region. In 2017, leaf emergence began in early May. Thus, measured N deposition would have been higher in April 2018 than a year before presumably related to a lower stomatal resistance in 2018. Almost equal patterns of $v_d$ and $R_{c,eff}$ were determined for May 2017 and 2018. The conditions for uptake of $\Sigma N_r$ by the canopy were comparable. Consequently, the different contributions in $NH_3$ and conditions in radiation and temperature strongly affected $v_d$ and $R_{c,eff}$ and therewith the deposition of $\Sigma N_r$." The discussion of the April deposition was replaced by this response.

**Comment R1.52** *Line 562: "...confirmed by the similar daily cycle for May 2017 and 2018." Similar daily cycle of what? Please specify.*
**Response to R1.52** No deviations between the daily cycles of $v_d$ and $R_{c,eff}$ were found. Thus, conditions for surface uptake were comparable. We implemented the explanation in the revised manuscript.

**Comment R1.53** *Line 567: "Almost the same average...". This sentence is out of place relative to the rest of the paragraph. Please consider removing or consolidating with analysis of relative humidity and temperature in next paragraph.*
**Response to R1.53** We agree that the sentence is out of place. The sentence was deleted.

**Comment R1.54** *Line 571: The first sentence of this paragraph should either be removed or reworded. The use of "Therefore" implies a missing introductory sentence.*
**Response to R1.54** We agree. Please note that the sentence was deleted.

**Comment R1.55** *Line 572: What is the proposed mechanism by which dry conditions enhance Nr deposition? Are the authors proposing that the stomatal processes are a larger overall source of variability in the net canopy-scale flux than the cuticular processes? It is unclear from this paragraph, which seems to include multiple lines of analysis the connections or which are unclear as currently written. Please see my previous comment regarding the organization and clarity of section 4.2*
**Response to R1.55** As suggested by Reviewer 2, the analysis on the parameters regulating the fluxes should be made for $v_d$ and the $R_{c,eff}$. It shows if the association between fluxes and drivers is due to their effect on concentration or $v_d$. We further separated half-hours, which were influenced by precipitation, from the analysis since $\Sigma N_r$ compounds like $NH_3$, $HNO_3$ and $NH_4^+$ are affected by rain. Thus, the entire paragraph (lines 571 - 598) was rewritten. Updated versions of Fig. 4 for $v_d$ and $R_{c,eff}$ are shown in R2.2. Also, further details on $R_a$, $R_b$, and $R_{c,eff}$ are given in R2.2. The discussion regrading line 572 was modified as follows:

"Within the period of high incident radiation, in particular from May to September, a distinct diurnal pattern for $v_d$ was observed, and no precipitation, high temperatures ($> 14.6°C$), low relative humidity ($< 74.0\%$), and dry leaf surfaces, were found to enhance the surface uptake, presumably through the stomatal pathway, of nitrogen during daylight. The observed differences in $v_d$ for relative humidity and temperature were mostly related to $R_a$ and $R_b$. $R_{c,eff}$ showed only a slight response to lower air humidity. Responses to the chosen temperature threshold and to dry leaf surfaces were not found.

During the rest of the year, no diurnal pattern was found under dry conditions (no precipitation) since stomata were likely closed, or requirements for stomatal deposition were not fulfilled (stomatal compensation point). Since we still observed a low, non-zero $v_d$ but also short phases of $\Sigma N_r$

emission during seasons with lower radiation, cuticular, soil, and turbulent driven processes were likely to be responsible for the $\Sigma N_r$ exchange. In periods of reduced plant activity, for instance in winter and autumn, the uptake through the stomatal pathway was greatly reduced or even inhibited due to reduced radiation or leaf area surfaces. Besides stomatal deposition, cuticular deposition is also an important pathway for $\Sigma N_r$ compounds, which likely deposit on wet surfaces such as $NH_3$, $HNO_3$ or $NH_4^+$.

However, $v_d$ was lower under wet conditions. Presumably, requirements for cuticular deposition were not fully met. Measurements of $\Sigma N_r$ were conducted several kilometers away from nearby sources, and thus hydrophilic $\Sigma N_r$ components could be washed out before air masses reached the site. We showed that the contribution and concentrations of $N_r$ species, which can deposit on wet leaf surfaces, was comparatively low at the measurement site. Furthermore, those species were only indirectly measured, and wet leaf surfaces could be already saturated with water soluble $N_r$ species leading to a high cuticular compensation point. These issues may reduce the cuticular contribution to exchange processes with the canopy. Presumably, cuticular deposition was probably not as important as stomatal deposition during periods of high incident radiation, in particular from May to September. Stomatal deposition seems to be more important than other in-canopy uptake processes for the ecosystem in close proximity to the measurement site for those months."

**Comment R1.56** *Line 573: The sentence "Higher concentrations values lead to higher deposition values through the entire daily cycle." seems out of place. How does this statement relate to the preceding sentence?*
**Response to R1.56** Higher concentrations of $\Sigma N_r$ lead to a higher deposition as visualized by Panel (d) of Fig. 4. It is obvious that $\Sigma N_r$ deposition scales approximately with its concentration since several components are included in the $\Sigma N_r$ concentration signal. Thus, the statement was deleted in the revised version.

**Comment R1.57** *Line 576: "Higher temperatures increase the opening size of the stomata leading to increased photosynthetic activity." What do the authors mean by "photosynthetic activity" in the context of the Nr fluxes?*
**Response to R1.57** Higher temperatures lead to an increased plant activity and lower the stomatal resistance favoring $\Sigma N_r$ deposition up to an optimum. As shown by (Wichink Kruit et al., 2010), stomatal conductance decreases with increasing temperature after reaching its maximum. Moreover, the maximal stomatal conductance depends on several parameters such as vegetation, RH, $R_g$, T, etc. (see Appendix E of Wichink Kruit et al., 2010). Please note that this sentence was deleted.

**Comment R1.58** *"Thus we examined the influence of precipitation on fluxes." Would it not be more straightforward to compare fluxes during wet versus dry conditions as indicated by the leaf wetness sensors, perhaps binning by day versus night or air concentration to examine the relationship while controlling for other sources of variability? I'm not sure what precipitation rate in figure F1 is telling us about the relationship between flux and canopy wetness. Is the canopy any less wet (or leaf water layers thinner) after a prolonged 0.5 mm/h rainfall compared to short duration 5 mm/h rainfall? To clarify, are these flux measurements conducted during active precipitation? What is the quality of the EC fluxes during such periods? Please add another figure to F1 similar to plot b) but for the fluxes and include in discussion.*
**Response to R1.58** We agree that a differentiation into precipitation classes was less useful. As written before, we did a reanalysis of Fig. 4 by separating fluxes, $v_d$, and $R_{c,\text{eff}}$ in dry and wet

classes. With the improved versions of Fig. 4, Fig. F1 added no additional information and was removed. The quality of fluxes measured during rain was almost similar to flux measurements with no measured precipitation. For example, 15% of the "wet" fluxes were classified as flag two following the Mauder and Foken flagging system (Mauder and Foken, 2006). Also, 15% of the "dry" fluxes were classified as flag two.

**Comment R1.59** *Figure F1: Please begin the caption by describing plot a) rather than plot b).*
**Response to R1.59** Please see the previous answer.

**Comment R1.60** *Line 587: "It has to be considered that the catchment, in which the flux tower is located, has a size of approximately 0.69 km2 (Beudert and Breit, 2010) and is larger than the catchment of Wyers and Erisman (1998). Also, the surrounding forested area is much larger and the entire area is mountainous. The forest stand is relatively young since it is recovering from a bark beetle outbreak in the 1990s and 2000s (Beudert and Breit, 2014)." Please clarify how these statements are relevant to discussion of the relationship between surface wetness and flux.*
**Response to R1.60** These statements were deleted since they add no relevant information to the discussion of surface wetness and flux.

**Comment R1.61** *Line 592: "Presumably, if NH3 concentrations are low, Nr dry deposition seems to be favored by dry conditions." Please clarify how this conclusion follows from the analysis of the Wyers and Erisman (1998) and Woff et al (2010) studies. What would be the underlying leaf-level mechanism?*
**Response to R1.61** We agree that this assumption needs further clarification. "Wyers and Erisman (1998) measured highest $NH_3$ deposition if the canopy has a high water storage level (CWS) ($> 2\,\mathrm{mm}$ ). The deposition efficiency was reduced if CWS was higher than $0.25\,\mathrm{mm}$ but lower than $2\,\mathrm{mm}$. By comparing different measurement years, they found differences in the deposition efficiency even if the canopy was saturated with water. They attributed the effect to the solubility of $NH_3$ in the water film. If canopy gets drier, evaporation of water occurs and the concentration of $NH_3$ increases in the water film. The cuticular resistance increases and deposition of $NH_3$ is reduced. Even emission of $NH_3$ was observed by Wyers and Erisman (1998), especially during the day when the canopy was dry, and $NH_3$ exchange was bidirectional. They showed that stomatal resistance was higher than canopy resistance. The authors identified cuticular deposition as more important for $NH_3$ as stomatal deposition. They measured an average $NH_3$ concentration of $5.2\,\mu\mathrm{g}\,\mathrm{m}^{-3}$. We measured $0.65\,\mu\mathrm{g}\,\mathrm{m}^{-3}$ on average and found that the contribution of $NH_3$ to $\Sigma N_r$ was comparatively low at the measurement site. If contribution of $NH_3$ or other soluble $N_r$ species to $\Sigma N_r$ is comparatively low, cuticular deposition is most likely reduced under wet conditions. The authors proposed that even under low ambient humidity leaf surfaces can exhibit high humidity due to the accumulation of particles. In case of conifer needles, Burkhardt et al. (1995) showed that particles deposit close to their stomata. Most of them are hygroscopic. Therewith, cuticular deposition seems to be possible even under low ambient humidity. However, our measurement site was several kilometers away from potential (anthropogenic) emission sources. Concentrations of $NO_3^-$, $NH_4^+$, sulphur dioxide ($SO_2$), and $NO_x$ were comparatively low at the site, in particular during summer. Thus, stomatal deposition appears to be more important for $\Sigma N_r$ under high temperatures, low relative humidity, and no precipitation. This conclusion is valid for months with sufficient light/energy input leading to an increased plant activity, i.e. from May to September. Within the other seasons, aerosol concentrations originating from natural or

anthropogenic emission sources are probably higher resulting in a higher particle density on leaf surfaces promoting cuticular deposition.

Wolff et al. (2010) observed high deposition of tot-$NH_4^+$ and tot-$NO_3^-$ during sunny days. During rain or fog, tot-$NO_3^-$ exchange was almost neutral and emission was observed for tot-$NH_4^+$. They measured median concentration of 0.57, 0.12, 0.76, and 0.45 $\mu$g m$^{-3}$ for $NH_3$, $HNO_3$, $NH_4^+$, and $NO_3^-$, respectively. For the September months, we measured average concentrations of 0.76, 0.46, 0.50, and 0.78 $\mu$g m$^{-3}$ for $NH_3$, $HNO_3$, $NH_4^+$, and $NO_3^-$, respectively. Measured tot-$NO_3^-$ and tot-$NH_4^+$ of Wolff et al. (2010) exhibited a higher particle than gaseous contribution. At our measurement site, the gaseous contribution was higher than the values reported by Wolff et al. (2010). Median deposition velocities of tot-$NO_3^-$ and tot-$NH_4^+$ were higher than values measured for $\Sigma N_r$ at our site, and they found that deposition was mainly driven by aerodynamic resistance rather than by surface resistance, in particular during periods of high radiation. It shows that changes in the contribution of $N_r$ species to $\Sigma N_r$ lead to different deposition pathways." The sentence was replaced by this response.

**Comment R1.62** *Lines 595-598: It is unclear how the sentences on wet deposition relate to the rest of the paragraph. Please consider removing.*
**Response to R1.62** We agree. The sentences were removed.

**From R1.63 to R1.80, suggested modifications to the text and recommendations of the Reviewer are related to the modeling part and will be implemented in second manuscript.**

**Comment R1.63** *Line 609: "the implementation of Nr species like HNO3 is relatively straightforward compared to NH3" is out of place in this sentence. Consider removing.*
**Response to R1.63** The sentence will be removed.

**Comment R1.64** *Line 618: Change "uncertainties sources" to "sources of uncertainty".*
**Response to R1.64** Agreed.

**Comment R1.65** *Line 633: Change "much needed approach" to "much improved approach"*
**Response to R1.65** Agreed.

**Comment R1.66** *Line 663: "most of the studies.." Please indicate which studies the authors are referring to.*
**Response to R1.66** Agreed. We consider to remove that sentence in the modeling study.

**Comment R1.67** *Line 667: "and the inclusion of exchange mechanisms for NO3 and NH4 should be considered in-situ modeling approaches." Please clarify what is meant here.*
**Response to R1.67** Currently, deposition of $NO_3^-$ and $NH_4^-$ is not included in DEPAC-1D. We will include particle deposition in DEPAC-1D for the modeling study. DELTA measurements will be used as input data.

**Comment R1.68** *Line 671: As a general question, how well does the DEPAC total Nr flux reflect the relationships between measured TRANC Nr flux and radiation, temperature/RH/dryness described in section 4.2?*
**Response to R1.68** We appreciate the Reviewer's comment. This will be part of the modeling

study and compared with results from the TRANC-CLD system. A similar analysis to Fig. R7 and R8 will be made for DEPAC-1D.

**Comment R1.69** *Line 682: And at sites with sparse vegetation.*
**Response to R1.69** Will be added to end of the sentence.

**Comment R1.70** *Line 685: Change "almost similar" to "similar".*
**Response to R1.70** Agreed.

**Comment R1.71** *Line 688: Has VDI been explained/defined?*
**Response to R1.71** "Verein deutscher Ingenieure" (Association of German Engineers) is missing.

**Comment R1.72** *Line 689-690: The two sentences here related to NH3 should be move to the preceding paragraph.*
**Response to R1.72** We will move the sentences to the preceding paragraph.

**Comment R1.73** *Line 696: The use of "positive" to describe the deposition velocity is not necessary.*
**Response to R1.73** Agreed.

**Comment R1.74** *Line 712: Why is CBT mentioned here in the discussion of LOTOS-EUROS?*
**Response to R1.74** The sentence seems out of place here and will be deleted.

**Comment R1.75** *Line 720: As previously mentioned, a summary and comparison of the various measurement techniques would be helpful to this discussion. Could the authors add a table summarizing the statistics of QCL, DELTA, and passive measurements, along with the LOTOS-EUROS predictions, as supplemental material? How well did the measurement techniques agree?*
**Response to R1.75** We will add a figure similar to Fig. R9 but for the LOTOS-EUROS concentrations and a figure similar to Fig. R13 with $NH_3$ from LOTOS-EUROS.

**Comment R1.76** *Line 722: "The difference to LOTOS-EUROS NH3 concentrations was highest during periods with significant amount of NH3 in the atmosphere like in spring and autumn, which is caused by emissions from fertilizer leading to a high load of modeled concentrations." Please reword this sentence, avoiding the use of "like" and "load".*
**Response to R1.76** Agreed.

**Comment R1.77** *Line 726: I encourage the authors to revisit the point and usefulness of this paragraph. As written I can't see that it adds anything to the discussion.*
**Response to R1.77** A reduction in grid cell size may lead to improvements in the localization of the emission sources. In close proximity to the flux tower, only a few emission sources were located. Thus, a reduction of the size may reduce the modeled concentrations of grid cell, in which the measurement site was located. We will modify the paragraph accordingly.

**Comment R1.78** *Line 760: "The deposition event in February 2018 seen by the TRANC seems to be driven by particulate Nr." Do the DELTA measurements reflect higher NH4+ and NO3- concentrations during this period compared to other months? These data should be presented.*
**Response to R1.78** Please see R1.20.

**Comment R1.79** *Line 775: The details here (i.e., "were selected from a matured tree stand") highlight that more information is needed in the method section regarding CBT as it was specifically applied at this site.*

**Response to R1.79** Further information about the tree stand will be added to the description of the CBT approach. The description will be shifted to the modeling manuscript.

**Comment R1.80** *Line 783: And to CBT.*

**Response to R1.80** Will be added.

**Comment R1.81** *Line 779: Conclusions section. Much of the information contained in this section is a direct recap of the preceding results and discussions. The length of this section could be significantly reduced.*

**Response to R1.81** Due to the separation into two studies, the length of the conclusion of the revised manuscript was reduced. We stated the conclusions more precisely.

**Response to Reviewer 2**

**General Comments** *The paper presents a 2.5-year long dataset of dry deposition of total reactive nitrogen (Nr) to a forest site, interpretation of the results in the light of measurements of Nr speciation, and a comparison of the results with alternative approaches: the prediction of a chemistry and transport model, a site-specific inferential model and a canopy budget technique. Direct measurements of Nr dry deposition is rare and such a long dataset of Nr dry deposition measurements to forest is unique and important, and thus generally publishable in Biogeosciences. I had high hopes for this paper, especially because the Nr flux measurements were accompanied by NH3 flux measurements (by QCL), which I hoped would have been used to elucidate the non-NH3 component of the Nr flux. However, I was let down in various aspects: the NH3 fluxes are not used in this paper (only concentrations). It is not stated whether they just did not work or whether they are left for another paper. However, this paper speculates a lot about the nature of the NH3 exchange and its impact on the total Nr flux and with NH3 flux data presumably available to explore this explicitly, this seems rather odd. In addition, the Discussion section is quite long and lacks structure and aim. The advantage of the TRANC is that it captures most of the Nr flux with one instrument. The disadvantage is that it does not shed light on the behaviour of the individual Nr components. Yet, much of the discussion is dedicated to relating the measured flux to the behaviour of individual compounds reported in the literature. I do not think this adds to the manuscript and should be shortened. Instead the paper should be more focussed on describing the flux in its totality. For example, the Nr dry deposition budget is not discussed in the context of the additional wet deposition which could be taken either from nearby measurements (if available) or the LOTOS-EUROS prediction. A number of serious concerns need to be addressed as raised below before the manuscript can be accepted for publication. This will require significant reworking and refocussing of the manuscript.*

We thank the Reviewer for his/her comments and criticism on this work. The determination of the NH$_3$ fluxes with the eddy-covariance method was not possible (see R2.1). If NH$_3$ fluxes by the QCL were available, an investigation of the non-NH$_3$ component would be included in the manuscript. Up to now, publications about flux measurements of $\Sigma$N$_r$ are rare. Thus, we have not much comparison possibilities in case of $\Sigma$N$_r$. The discussion was extended to individual $\Sigma$N$_r$ compounds in order to show that the flux magnitude of the individual compounds is in agreement with our measurements for similar ecosystems. However, we agree that the discussion on that topic was too long and can be shortened. We deleted lines 485-545 and shortened the discussion on individual $\Sigma$N$_r$ compounds substantially. We plan to shift the discussion of the individual $\Sigma$N$_r$ compounds to the modeling manuscript. As stated in the author comment, we made an analysis on $v_d$ and $R_{c,eff}$ and determined the total nitrogen budget. We included measurements of wet deposition taken close to the tower by bulk and wet-only samplers and investigated the influence of micrometeorology on the nitrogen dry deposition sums using data-driven gap-filling methods.

   In the modeling study, individual flux components of DEPAC-1D will be compared to values reported in literature. As done for the TRANC-CLD measurements, an analysis of the micrometeorological parameters will be made for DEPAC-1D. The discussion of the dry deposition budgets will be improved, and wet deposition estimates from LOTOS-EUROS will be included. We will discuss the ecological impact of nitrogen deposition on forest ecosystems. A comparison to annual N budgets reported for other forest ecosystems will be carried out. We addressed all mentioned points related to the flux measurements and implemented your suggestions in the revised manuscript. Since we made a separation of the modeling part, a detailed reply to the $\Sigma$N$_r$ modeling results will

not be made yet.

**Main scientific comments**

**Comment R2.1** *As mentioned above, if the NH3 fluxes could be worked into the manuscript this would strengthen the analysis a lot.*

**Response to R2.1** As stated above, an evaluation of $NH_3$ fluxes with the eddy-covariance method was not possible. We added the following lines to the end of Sec. 2.3: "As outlined in Sec. 2.2, measurements of $NH_3$ were made with a QCL at high temporal resolution. In combination with the sonic anemometer, it gives the opportunity to determine $NH_3$ fluxes and to further investigate the non-$NH_3$ component of the $\Sigma N_r$ flux. However, a calculation of the $NH_3$ fluxes with the EC method was not possible in this study. No consistent $NH_3$ time lag was found making flux evaluation impossible. Due to regular pump maintenance, cleaning of the inlet and absorption cell, issues related to the setup of the QCL were unlikely to be the cause. We suppose that the variability in the measured $NH_3$ concentrations was not sufficiently detectable by the instrument. Significant short-term variability in the $\Sigma N_r$ raw concentrations were not found in the $NH_3$ signal even in spring or summer. Thus, no robust time lag estimation could be applied to the vertical wind component of the sonic anemometer and the $NH_3$ concentration. Recently, Ferrara et al. (2021) found large uncertainties for low $NH_3$ fluxes measured with the same QCL model. Cross-covariance functions had a low signal-to noise ratio indicating that most of the fluxes were close to the detection limit."

**Comment R2.2** *The paper confuses the rate of deposition (deposition velocity) and the actual deposition. Ignoring the effects of compensation points on NH3 exchange and the contribution of soil NO emissions to the net flux of NO and NO2, and also changes in the relative contribution of different compounds to Nr, the deposition of Nr is expected to scale approximately with its concentration. This is trivial and fundamentally also the way the deposition is calculated in LOTOS-EUROS and DEPAC-1D. Changes in concentration therefore mask the mechanisms that regulate the deposition rate. Thus, the analysis would be much more meaningful if the authors examined the controls of the deposition velocity rather than of the flux. This is what is done in the literature for the different compounds and, currently, comparisons are not correct. For example, it is stated that NH3 fluxes are largest under wet conditions. In fact most studies report that Vd is larger for wet conditions, but at the same time the concentration may be reduced. For this reason statements like "dry conditions seem to favour nitrogen dry deposition (line 13, also line 793f)" are maybe not incorrect, but certainly misleading. Throughout the analysis it is rarely clear whether an association between the flux and drivers is due to their effect on concentration or Vd. For example, Fig. 4 would be more meaningful if presented for Vd. In fact, an analysis in terms of Rc would be even more meaningful as it would normalise for the effect of turbulence on Ra and Rb both of which contribute to Vd. Because particles are not really subject to a boundary-layer resistance in the way it is applied to gases, Rc is not really meaningful. However, the analysis could be done in terms of Vds = Vd(z0), i.e. after normalising at least for Ra.*

**Response to R2.2** The $\Sigma N_r$ compounds have different exchange pattern and differ in their interaction and reaction pathways. Thus, it is difficult to show one deposition velocity for $\Sigma N_r$. However, we agree that the manuscript benefits from an analysis of $v_d$ in order to show if an influence of a driver on the flux is due to its effect on $v_d$ or concentration. Figure R7 was done in accordance to Fig. 4 in the manuscript but for $v_d$.

We further determined the aerodynamic resistance ($R_a$) following Garland (1977) and the boundary-layer resistance ($R_b$) following Jensen and Hummelshøj (1995, 1997). $R_b$ requires a

molecular diffusion coefficient of $\Sigma N_r$. We determined the molecular diffusion coefficient for $\Sigma N_r$ as the weighted average of the campaign-wise averages of $HNO_3$, $NH_3$, $NO$, and $NO_2$ multiplied with their individual molecular diffusivities adapted from Massman (1998) and Durham and Stockburger (1986). The effective canopy resistance $R_{c,eff}$ was determined by subtracting the maximum deposition velocity allowed by turbulence from the measured deposition velocity. In Sec. 2.4 "Determining deposition velocity and canopy resistance of $\Sigma N_r$ from measurements", equations needed for the calculation of $v_d$ and resistances are given. Figure 4 and the corresponding description (lines 353-360) were deleted and replaced by the figures and text shown in this response.

[revised manuscript text omitted]

cycles of each parameter shown in Fig. R7 and R8 are almost similar for the chosen threshold values and differ only in amplitude."

Due to the focus on $v_d$ and resistances, the interpretation of the results had to be rewritten. Please see also R1.47, R1.48 R1.50, R1.51, R1.55, and R1.61. For the second part of this study, we plan to add a discussion of resistances and $v_d$ calculated from the TRANC measurements compared to the results from LOTOS-EUROS and DEPAC-1D. Additionally, the investigation on micrometeorological controls will be applied to $\Sigma N_r$ fluxes modeled by DEPAC-1D.

**Comment R2.3** *The interpretation of the measurements is not helped by the lack of showing absolute concentrations. The relative composition of total Nr (Figs. B1 and E1) is useful, but also the absolute concentrations are needed to interpret the results. Again, because fluxes are discussed in terms of their magnitude and not their Vd the reader is left wondering whether whether it is really the change in relative composition that changes the flux or whether it is just the overall Nr concentration. By the way, it is unclear what time periods are shown by each pie chart and what frequency this maximum refers to (Caption and text Line 305ff). Presumably, these are monthly results given that the lowest data resolution (from the DELTA) is monthly? Indeed, I would find a figure showing monthly stacked bar graphs of the individual Nr components very useful. This would convey how the total and their contribution to total Nr changed seasonally. Also, an assessment of how well the sum of the individual N compounds compares with the total Nr concentration needs to be added as quality control.*

**Response to R2.3** We agree that a comparison of the absolute concentration values is helpful for interpreting differences in the flux pattern. The pie chart (c) covers the entire measurement period of the DELTA system. (a) and (b) show a pie chart with the lowest and highest concentration of TRANC $\Sigma N_r$ during the exposition periods of the DELTA samplers. Yes, the underlying time resolution is approximately monthly since the denuder were exchanged nearly every month. By the comparison of the absolute values, we found that the zero-air calibration value of the TRANC-CLD system was incorrect from July to September 2017 and from March to mid of May 2018 by approximately $0.9\,\mu g$ N m$^{-3}$ compared to the uncorrected TRANC-CLD concentrations. Concentrations and fluxes were recalculated with the bias correction. Figures shown in the response are made with the bias-corrected data. In the revised version, Figs. B1 and E1 were deleted since we found no significant deviations of the minimum and maximum TRANC $\Sigma N_r$ cases to average after the bias correction. The following lines including the figures were added to Sec. 3.1 after line 302.

[revised manuscript text omitted]

The changes in the composition of $\Sigma N_r$ were also affecting $v_d$. Only slight seasonal changes in the overall $\Sigma N_r$ concentration were observed. We measured 3.3, 2.6, 2.5, and $3\,\mu g$ N m$^{-3}$ for

spring, summer, autumn, and winter, respectively. Consequently, it was not only the change in the overall $\Sigma N_r$ concentration that influenced $v_d$. Please note the new subsection 4.2.2 "Influence of $N_r$ species on measured $v_d$."

**Comment R2.4** *The measurements are compared to those made over other ecosystems and differences are explained by differences in ecosystems. Again, this is only part of the story, mainly the part that affects Vd. The pollution climate the ecosystem is in is equally important and does not necessarily correlate with the ecosystem type (think of an urbanwoodland or a heavily grazed pasture in otherwise pristine environment). The comparison needs to be reworded. Generalisation that Nr fluxes always behave above natural vegetation as they do at this particular site is not tenable (e.g. line 13 and throughout).*
**Response to R2.4** We appreciate your comment and corrected the corresponding comparisons, e.g. line 458.

**Comment R2.5** *The analysis of the effect of precipitation on the flux (Fig. F1a and associated text) is particularly problematic. During rain the eddy-covariance flux measurement of water soluble compounds (and many Nr compounds are) is highly uncertain because fluxes cannot be assumed to be constant with height due to the washout process. An increased Vd during rain may just reflect the presence of an additional sink (the washout process) below the measurement height. Rain episodes should potentially be filtered out, but certainly no process understanding should be derived from data taken during rain. How do the measurements demonstrate that wet deposition is important (Line 595)?*
**Response to R2.5** Based on the suggestions of Reviewer 1 and your comment, we removed the corresponding text (lines 584-598) and Fig. F1. Yes, we agree that rainy episodes should be filtered out since water soluble $N_r$ such as $NH_3$, $HNO_3$, and $NH_4^+$ were probably washed out from air masses before reaching the measurement height. As written before, we did a reanalysis of Fig. 4 by separating fluxes, $v_d$, and $R_{c,eff}$ in dry and wet classes. Please note the responses to comments 1.55 and 1.58 to 1.62. The sentence "It shows that wet deposition is important for the uptake of $\Sigma N_r$ compounds at our measurement site." was certainly misleading and deleted. Wet deposition samplers were in close proximity to the flux tower. Thus, wash out processes also affected wet deposition measurements.

**Comment R2.6** *The paper does not distinguish different types of error (e.g. lines 617f and 652f). The flux error according to Finkelstein and Sims describes a random error, whereas the uncertainty in the DEPAC-1D estimate is more likely to be systematic and thus provide a bias. The input parameters are considered the largest uncertainty in DEPAC-1D (lines 619f), but actually different inferential models give very different results which highlights their uncertainty (e.g. Flechard et al., 2011).*
**Response to R2.6** The mixture of the different error types was not intended. In the revised version of the measurement part, the flux uncertainty of the gap-filled fluxes was calculated as the standard error of mean. The random uncertainty following Finkelstein and Sims (2001) was included in the discussion. Total uncertainty from random error estimates was calculated as square root of the sum of the squared random uncertainties according to Pastorello et al. (2020). Please note the substantial changes to Sec. 4.3, which was renamed to "Uncertainties in dry deposition estimates". The uncertainty discussion of DEPAC-1D and LOTOS-EUROS was deleted and will be moved to the modeling study and substantially improved.

**Comment R2.7** *This then also relates to an apparent contradiction between the discussion of the importance of stomatal exchange (Line 575) which is temperature dependent but mainly regulated by PAR and the statement that the canopy resistance is mainly driven by water solubility (Line 702).*

**Response to R2.7** We thank the reviewer for his/her hint to this contradiction. We improved the discussion on the uptake capacity of $\Sigma N_r$. Please note the new subsection 4.2.3 "Seasonal changes in $\Sigma N_r$ uptake capacity". The information related to the modeling part will be improved accordingly.

**Comment R2.8** *Still on the topic of drivers of the exchange, a similarity in the diurnal cycle between global radiation and flux is no proof of causality (line 549ff). A lot of parameters are driven by the radiation: turbulence, photochemistry etc.. Neural networks also do not derive causalities or 'drivers', only associations and determinants.*

**Response to R2.8** We appreciate your comment and reworded the corresponding lines. Please note R1.47 and the description to Fig. R5 and R6. Zöll et al. (2019) showed that global radiation and concentration added independent information to the variability of the $\Sigma N_r$ fluxes. Adding other parameters like temperature, $u_*$, or $CO_2$ as secondary driver resulted in lower values if global radiation was chosen as primary driver. Their investigation revealed that global radiation contained important information for the explanation of the $\Sigma N_r$ fluxes. The word 'driver' is a paraphrase of the expression controlling input variable (Moffat et al., 2010). Drivers are identified by their correlation with the flux. In general, correlations could also be influenced by other parameters, which have not or could not considered by Zöll et al. (2019), for example chemical interactions of components contributing to $\Sigma N_r$. We agree that the word driver could be misinterpreted without proper explanation. We implemented the explanations given in this response to line 551.

**Comment R2.9** *The filtering criteria will have removed preferentially the smaller fluxes during low turbulence conditions and the remaining dataset will therefore be biased. Whilst this is not an issue if a model is used for gap filling that accounts for changes in turbulence, it does impact the straight averages of the fluxes (Figure 2) the value of which then becomes questionable and also the MDV gap filling method. These issues and implications need to be discussed.*

**Response to R2.9** Yes, the application of the filtering criteria like Mauder and Foken or a friction velocity threshold could preferentially remove smaller fluxes, which occurred at night-time. Therefore, we introduced a new section "Sensitivity of $\Sigma N_r$ dry deposition sums to micrometeorological parameters". In this chapter beginning at line 362, we investigated possible dependencies of the $\Sigma N_r$ dry deposition sum on micrometeorological parameters if data-driven gap-filling methods like the Mean-Diurnal-Variation (MDV) method were used. Text and figures of this response were added to this chapter. We further calculated total annual depositions by using wet depositions measurements from wet-only samplers. Details about the wet deposition measurements were added to line 174.

[revised manuscript text omitted]

**Comment R2.10** *The use of monthly mean concentrations for some of the compounds (DELTA measurements) adds significant uncertainty to DEPAC-1D model results. The first mention that the DELTA measurements are monthly seems to come in line 303 and the uncertainties are not mentioned until Line 622 (and there without references to, e.g., Schrader et al. 2018). The limitations of this approach should be more visible earlier on. Was the gap-filling of NH3 (Line 257) done in a mass-conserved way, i.e. was the available data removed from the long-term NH3 aver-*

*age to work out what the average concentration during the gaps might have been? I suppose this would lower the uncertainty somewhat? Was a diurnal cycle superimposed on the long-temporal resolution measurements?*

**Response to R2.10** Yes, the usage of monthly mean values introduces a significant uncertainty to DEPAC-1D. We agree that the DELTA resolution has to be mentioned earlier. We added it to line 165 and will implement a detailed description of the usage of monthly DELTA concentrations and related uncertainties for the in-situ modeling in the second part.

**Comment R2.11** *I do not follow the introduction of the DEPAC algorithm (Section 2.4.1). Erisman et al. (1994) does not describe a bidirectional resistance model (Line 224). Similarly, the references in lines 230-231 all describe deposition parameterisations, but most are almost certainly not the ones used in this version of DEPAC and contradict each other. The most correct description probably comes in Lines 243-247. Much of the description of the DEPAC-1D (Section 2.4.3), including the resistance parameterisations, probably also apply to the DEPAC version implemented in LOTOS-EUROS? It is all a little confusing. I did not realise until the Discussion section that DEPAC-1D does not treat the aerosol. This is a major and seemingly unnecessary shortcoming. My understanding was that DEPAC-1D is a stand-alone version of the deposition scheme implemented in LOTOS-EUROS and surely the latter treats the aerosol components. This seems hardly justifiable.*

**Response to R2.11** We thank the Reviewer for his/her hints. We will improve the description of DEPAC and check the corresponding references within the preparation of the modeling study.

**Comment R2.12** *I am confused throughout about the use of a compensation point for NH3 in the versions of LOTOS-EUROS and DEPAC-1D used. What is its magnitude for the forest types under consideration and where does it come from? Line 264 says that the DELTA concentrations were used for determining compensation points and additional deposition corrections? How was this done? Does this mean the models were not run with the standard scheme for these ecosystem types? Monthly concentrations do not lend themselves to deriving compensation points. Lines 671ff discuss uncertainties around cuticular compensation points. This would suggest that this was somehow adjusted based on the measurements?*

**Response to R2.12** We appreciate the Reviewer's suggestions. Since the question relates to modeling part of the manuscript, this question will be answered in the modeling study.

**Comment R2.13** *Given all this discussion about compensation points it is then highly surprising that Vd for HNO3 and NH3 are virtually identical (Line 374). How can this be? Apart from potential of evaporating NH4NO3 on leaf surfaces, HNO3 exchange is well understood and follows a near-zero Rc. NH3 does not.*

**Response to R2.13** See above.

**Comment R2.14** *I am similarly unclear about the discussion of the landcover (Lines 236-242). Given the resolution of LOTOS-EUROS of 7 x 7 km2 it is not surprising that the landcover of the grid cell containing the measurement site does not match that of the flux footprint which is much smaller. But I also do not see a big problem: is LOTOS-EUROS not based on a mosaic / tiling approach and predict fluxes to each landcover type separately? The associated description of the LAI values (Lines 273-279) is also unclear. Surely DEPAC-1D and LOTUS-EUROS simulate the deposition to all landuse types in a gridcell and from those a landcover-weighted average can then be calculated? In general, it should be made clearer what is identical and what is different*

*between the LOTOS-EUROS and the DEPAC-1D simulation. What measurements were used for DEPAC-1D? Concentrations, meteorological parameters, canopy characteristics?*

**Response to R2.14** We thank the the Reviewer for his/her advice. We will implement your suggestions in the preparation of the modeling manuscript.

**Comment R2.15** *The December emission fluxes are insufficiently explained. Were temperatures really sufficiently high to drive NH3 emissions from decomposition (Line 489)? Is there any evidence of freeze-thaw cycles affecting NH3 fluxes (Line 496)? Possibly, freeze-thaw cycle effects on soil NO are a more likely explanation? However, does the flux direction actually correlate with freeze-thaw events? Could it be caused by a problem with the measurement setup for a period of time given that December measurements differed between the two years?*

**Response to R2.15** Yes, you are right that the emission fluxes were insufficiently explained. Previous conclusions regarding $NH_3$ being mainly responsible for the observed emission was most likely incorrect. Based on your suggestions and Reviewer 1, we improved the description. Please see R1.37. No issues with the instrument were found during the periods in December 2016 and 2017.

**Minor scientific comments**

**Comment R2.16** *The abstract seems overly long and should be shorted. This can be done linguistically (e.g. remove phrases such as "We further showed that") and in terms of content. For example, it is sufficient to list the results in terms of annual deposition inputs and remove the numbers for the 2.5-year timeframe (line 19ff).*

**Response to R2.16** Due to the separation of the manuscript, the abstract length was reduced. We removed redundant phrases and numbers for 2.5-year time frame.

**Comment R2.17** *In Section 2.2 I am missing a fuller statement on the response of the TRANC to Nr compounds in the aerosol phase. What is the size-cut? What is the response to nitrate other than ammonium nitrate (e.g. sodium nitrate, calcium nitrate, ...)? Presumably they are not volatilised?*

**Response to R2.17** Marx et al. (2012) conducted particle conversions test for sodium nitrate ($NaNO_3$), ammonium nitrate ($NH_4NO_3$), and ammonium sulfate (($NH_4)_2SO_4$) since they are the most common nitrogen aerosol compounds (e.g., Wexler and Seinfeld, 1991; Nemitz et al., 2009). Aerosols were produced by a collision-type atomizer (TSI, St. Paul, USA) with a 0.3 mm nozzle from aqueous solutions of 0.5 g $l^{-1}$, 1 g $l^{-1}$, and 0.5 g $l^{-1}$, respectively (Marx et al., 2012). Conversion efficiencies were 78%, 142%, and 91% for $NaNO_3$, $NH_4NO_3$, and ($NH_4)_2SO_4$, respectively. A comparison with a twin differential mobility particle sizer (TDMPS) (Birmili et al., 1999) showed similar conversion efficiencies for $NaNO_3$ and ($NH_4)_2SO_4$ but differences for $NH_4NO_3$ (Marx et al., 2012, Fig. 6). At higher temperatures (>20°C) and relative humidity (>50%), $NH_4NO_3$ is semi-volatile resulting in higher fraction of $NH_3$ and $HNO_3$. Since TRANC-CLD detects the gaseous forms, a higher conversion efficiency than the one recorded by the particle detector can be expected. Overall, the results indicate that the TRANC is able to convert aerosols efficiently to NO. We added the determined conversion efficiencies for aerosols to the manuscript (line 149). For further details we refer to the publication of Marx et al. (2012)

**Comment R2.18** *Line 33ff. I am not aware that deposition of Nr components threatens hu-*

*man health. They do so by acting as precursors to PM2.5 and O3.*

**Response to R2.18** We added "by acting as precoursors for ozone ($O_3$) and PM2.5" to line 34.

**Comment R2.19** *Line 80f. The critique of the MDS method is difficult to understand because it is not explained what it is. The introduction of CTM approaches is a little messy. Line 90 explains their workings by needing meteorological data and land-use information. Emissions and chemistry are only mentioned much further down.*

**Response to R2.19** MDS utilizes the temporal correlation of micrometeorological parameters with fluxes to estimate gap-filled fluxes. In other words, "MDS requires a short-term stability of fluxes and micrometeorological parameters. This condition is not necessarily fulfilled for $\Sigma N_r$ and its components. Their exchange patterns are characterized by a higher variability for different time scales leading to a lower autocorrelation and non-stationarities in flux time series compared to inert gases like $CO_2$." We replaced the lines related to MDS (lines 80-81) by the highlighted sentences given in this response. In addition, $\Sigma N_r$ is a combination of several $N_r$ species, which differ in physical and chemical properties and in their seasonal contribution. Thus, the application of data-driven gap-filling methods is suitable for gaps being a few days long. We appreciate the Reviewer's remarks to the introduction of CTMs. Since the manuscript was separated, the introduction changed. The paragraphs to CTMs and DEPAC were deleted, and a short paragraph to $v_d$ and resistance analysis was added to the introduction (line 74).

**Comment R2.20** *The introduction of the principle of operation of the TRANC is also not very logical. First reduced N is oxidised and then NH3 is formed from NH4NO3? Surely this happens before the oxidation (or in the same step).*

**Response to R2.20** We agree that the description of the conversion steps is confusing. We deleted "resulting in an oxidization of reduced $N_r$ compounds" (line 142) and generally improved the description (see R1.5 and R2.17).

**Comment R2.21** *The description of turning the leaf wetness value into a boolean value needs to be improved (line 158ff). At present, a value of 10 in arbitrary units is meaningless.*

**Response to R2.21** We agree that the explanation needs to be improved. We added the following sentences to line 158: "Due to a wetting of the sensor's surface, the electric conductivity of the material changes. This signal, the leaf wetness, was converted by the instrument to dimensionless counts. Based on the number and range of counts, different wetness states could be defined. Half-hourly leaf wetness values were in the range from 0 to 270. In this study, we defined the wetness states "dry" and "wet". The condition wet can be induced by the accumulation of hygroscopic particles extending the duration of the wetness state or water droplets. In order to classify a leaf as dry or wet, we determined a threshold value based on the medians of leaf wetness values." In order to clarify the determination of the threshold value used for classifying a leaf wetness sensor as wet or dry, we replaced the corresponding line 158 by the following sentences: "During daylight (global radiation $> 20\,\mathrm{W\ m^{-2}}$), medians ranged from 1.1 to 2.0 and were between 4.1 and 9.4 during nighttime. During nighttime, medians are higher due to dew formation. According to the values determined during daylight, we set the threshold value to 1.5 for all sensors."

**Comment R2.22** *Line 166ff. Please state the temporal resolution of the DELTA measurements. Also, later the text refers to ammonia diffusion samplers and NOx measurements, which do not appear to be mentioned in Section 2.2.*

**Response to R2.22** Please see R1.7 and R1.10. NO and $NO_2$ measurements are mentioned in

Sec. 2.2. Here, $NO_x$ was determined by adding NO to $NO_2$ concentrations.

**Comment R2.23** *Line 199. Does the flux loss depend on the chemical composition of Nr?*
**Response to R2.23** Wintjen et al. (2020) determined flux loss factors for two different ecosystems, which are different, for example, in the composition of $\Sigma N_r$. They assumed that the differences in flux losses are also related to the chemical composition of $\Sigma N_r$. We added the information to line 199.

**Comment R2.24** *Line 207. Please state the relative magnitude of the water correction. What is its uncertainty?*
**Response to R2.24** Please see R1.8.

**Comment R2.25** *Line 211. Removal of fluxes outside a certain range appears to be arbitrary and subjective. Are these extreme fluxes not caught by the other tests, e.g. Foken's stationarity test or testing for stochastic significance via the random flux error? I presume the latter is what the "threshold of two times 1.96sigma" (Line 213) refers to? Currently, sigma is not defined and its calculation remains unexplained.*
**Response to R2.25** We applied a limit filter for flux and concentration in order to filter out extreme outliers. Some of them were not identified by quality flags of Mauder and Foken (2006) or by the stochastic significance of the random flux error. $\sigma$ represents the standard deviation of the variance. Fluxes were filtered out if variances of concentration, vertical wind, or temperature exceed the respective average plus $3 \cdot 1.96\sigma$. However, an investigation on the effectiveness of the filters revealed that quality flag criteria of Mauder and Foken (2006), a concentration limit filter, and a manual screening for periods of insufficient instrument performance, which resulted in irregularities in the raw signals (line 214-216), were sufficient to identify high-quality fluxes of $\Sigma N_r$. Please also note the answer to comment R1.9. Filters not needed were left out for preparation of the revised manuscript. We deleted the information to the variance filter (lines 212-213). Since other filters were chosen, the limits of flux filter and half-hourly fluxes also changed (lines 211 and 315).

**Comment R2.26** *Line 264f. How were compensation points derived from long-term measurements of SO2 and NH3? This would seem problematic.*
**Response to R2.26** We agree that additional details are needed to justify the determination of compensation points following Wichink Kruit et al. (2010). In the modeling study, the derivation of compensation points will be added to the description of DEPAC-1D.

**Comment R2.27** *Line 266. Why was the LAI modelled for a site-based application? Why was this not based on a measured value?*
**Response to R2.27** The LAI was not measured at the site. Please also see comment R1.11.

**Comment R2.28** *Line 390. How do the diurnal cycles compare between measurements and model results? Does this shed add additional light on model deficiencies?*
**Response to R2.28** We appreciate the Reviewers suggestions. A comparison of measured and modeled diurnal cycles will be made for the modeling study.

**Comment R2.29** *Line 434. No, concentration is not proportional flux. The flux is proportional to the concentration. The concentration is the driver.*

**Response to R2.29** Yes, you are right.

**Comment R2.30** *Line 468. What do the concentration ranges refer to?*
**Response to R2.30** Values for $NO_2$ and NO refer to 1992 until the end of 2008, $NH_3$ was measured from mid of 2003 to 2005. We added the information to line 468.

**Comment R2.31** *Line 501. Both NO and NO2 contribute to Nr. So even if soil NO is converted to NO2 it will still contribute to the Nr flux except for the fraction that is removed by the canopy.*
**Response to R2.31** We agree. The sentence was deleted, but we the implemented the Reviewer's suggestion in Sec 4.1.

**Comment R2.32** *Line 507. The DELTA samplers does not measure NOx.*
**Response to R2.32** Agreed. It should be DELTA+$NO_x$. Please note that the sentence was deleted.

**Comment R2.33** *Line 514. There is a range of coatings available for the DELTA denuders. Clarify here and possibly also in the Methods section that carbonate coating was indeed used.*
**Response to R2.33** We agree. For basic denuders, sodium carbonate and glycerol dissolved in water and methanol was used as coating for capturing $HNO_3$, $SO_2$, and $NO_3^-$, and citric acid and glycerol and also being dissolved in water and methanol as coating for acid denuders used for $NH_3$ and $NH_4^+$. Please note the changes to line 165 and comment R1.10.

**Comment R2.34** *Line 551. Presumably in addition to total Nr concentration, its speciation also affects the net deposition rate and thus the flux.*
**Response to R2.34** Probably, yes. Please note the revised discussion following line 551.

**Comment R2.35** *Line 721. Is it worth adding DELTA, QCL and passive sampler data all to the graph to have an intercomparison between measurements? How do HNO3 compare between model and measurement? The modelled values of NH3 could also be too high because HNO3 in the model is too low (thus forming less NH4NO3).*
**Response to R2.35** Figure R13 shows $NH_3$ concentrations of the DELTA system, passive samplers, and the QCL. $NH_3$ concentrations of the QCL were averaged to the exposition periods of the samplers. Figure R13 was added to supplement.

[Figure]

Figure R13: Concentrations of NH$_3$ measured by the DELTA and passive samplers, and the QCL in $\mu$g N m$^{-3}$. NH$_3$ of the QCL was averaged to the exposition period of the long-term samplers. Colors of the passive samplers indicate different measurement heights.

Averaged NH$_3$ concentrations of the QCL agreed well with NH$_3$ from passive samplers and DELTA measurements (Fig. R13). Overall, the agreement in the annual pattern was good, but a bias between the QCL and the diffusion samplers was found. From passive sampler measurements, an increase in the NH$_3$ concentration with measurement height could be observed. At 10 m (in the canopy), the lowest NH$_3$ concentrations were measured. No systematic difference was found between 20 m and 30 m. At 50 m, NH$_3$ was slightly higher (0.1 $\mu$g N m$^{-3}$) than 30 m. During winter, the difference in measurement heights diminished. Slightly higher NH$_3$ concentration were observed at 10 m in winter. A similar figure will be prepared for the modeling part including LOTOS-EUROS NH$_3$. As written in R1.29, a stacked bar graph to similar to Fig. R9 but with LOTOS-EUROS concentrations instead of TRANC $\Sigma$N$_r$ will be made for the modeling part.

**From R2.36 to R2.42, suggested modifications to the text and recommendations of the Reviewer are related to the modeling part and will be implemented in second manuscript.**

**Comment R2.36** *Line 739. The model presumably calculates u* from the ascribed canopy height and does not know about the complexity of the terrain. Are you saying that the measured u* is elevated because of topography? Would this not imply that the conditions for eddy-covariance are not met?*
**Response to R2.36** The deviation in $u_*$ was not related to the topography. $u_*$ was calculated with the wind speed given at the reference height. As written in the manuscript, the reference height of LOTOS-EUROS was lower than the measurement height of the EC system. A single grid cell consists of various vegetation types, and all of them have different roughness lengths. We showed that the vegetation of the flux footprint differs significantly from the vegetation generated by the land-use classes for the grid cell. Thus, differences in $u_*$ could be expected.

**Comment R2.37** *Line 754. "input NH3 concentrations" Do you refer to emissions or long-range transport?*
**Response to R2.37** For our measurement site, the elevated NH$_3$ concentrations were most likely

caused by emissions from nearby agriculture.

**Comment R2.38** *Line 763. If the deposition event wasn't measured it maybe did not exist. I suggest to rephrase: "All models predicted at 2nd emission event which was not confirmed by the measurements."*
**Response to R2.38** Agreed.

**Comment R2.39** *Line 793f. But you say the Vd of NH3 is very high almost as high as HNO3. Thus, a large relative contribution of NH3 should give you large deposition fluxes.*
**Response to R2.39** In case of the modeled $v_d$, yes. $v_d$ of $\Sigma N_r$ was significantly lower than the modeled $v_d$ of $NH_3$ and closer to $v_d$ of $NO_2$. Figure R10 reveals that $NO_x$, in particular $NO_2$, was the dominant $N_r$ species and not $NH_3$. Presumably, a measured $v_d$ of $NH_3$ would have been lower than modeled values.

**Comment R2.40** *Line 795f. The wash-out could have occurred upwind and not contributed to the local wet deposition.*
**Response to R2.40** Agreed. The sentence will be removed as written in R2.5.

**Comment R2.41** *Line 798f. The good agreement seems entirely fortuitous given aerosol was not included in DEPAC-1D ...*
**Response to R2.41** Currently, we are working on including of $NH_4^+$ and $NO_3^-$ in DEPAC-1D for the modeling study.

**Comment R2.42** *Line 803f. Maybe the gap filling methods are designed for compounds whose fluxes are actively regulated by production and consumption processes rather than the consequence of turbulence and concentrations such as deposition.*
**Response to R2.42** We appreciate the Reviewer's suggestion for rephrasing and will modify the sentence accordingly.

**Technical corrections / suggestions:**

**Comment R2.43** *General: avoid starting sentences with numbers. E.g. line 23 could better read "Deposition of 16.8 kg N ha-1 was calculated"*
**Response to R2.43** We changed the beginning of the corresponding sentences.

**Comment R2.44** *General: there are numerous places where an article is missing. E.g. line 86: "due to the low number", Line 146: "as a reducing agent", Line 179: "on an annual basis"*
**Response to R2.44** We went carefully through the text and add articles if necessary.

**Response to R2.44** General: there are several instances where the word "after" seems to be a mistranslation from German and needs to be replaced. Line 105: "were taken following the approaches of the International ...", Line 108: "nitrogen deposition using the canopy budget technique", Line 179: "bases following the CBT approach"

**Response to R2.44** We checked corresponding lines and replace "after" by appropriate words.

**Comment R2.45** *General: in many cases units are incorrectly combined. For example ms-1 should read m s-1 and µgm-3 should read µg m-3.*
**Response to R2.45** We improved the notation of the units and separate them correctly.

**Comment R2.46** *Line 7. I was surprised to see Nr concentration given in ppb rather than µg N m-3, especially since Nr contains aerosol compounds for which the use of ppb is rather unusual.*
**Response to R2.46** Previous studies on measurements of $\Sigma N_r$ by the TRANC also used ppb as unit for concentrations (e.g., Ammann et al., 2012; Brümmer et al., 2013; Zöll et al., 2019). In the TRANC, $N_r$ species are converted to NO. The measured $\Sigma N_r$ signal is basically NO, which is in a gaseous state under standard conditions. Therefore, the unit ppb seems to be appropriate for $\Sigma N_r$. In order to avoid switching between units, we changed the unit ppb to $\mu g$ N $m^{-3}$. For comparing the measured concentrations to reported concentrations from other publications (lines 458-470), we changed the unit to ppb.

**Comment R2.47** *Line 62. Better "EC studies of …"*
**Response to R2.47** Replaced by "Prior EC studies of..."

**Comment R2.48** *Line 69 refers to "that site", but it is not clear which site is meant at this point.*
**Response to R2.48** It will be replaced by "conducted with the same instrumentation at the measurement site".

**Comment R2.49** *Line 96. "validation with flux measurements" (or "against").*
**Response to R2.49** Revised.

**Comment R2.50** *Line 116. "Measurements were carried out in". Actually, the authors should consider the alternative "Measurements were made" here and elsewhere.*
**Response to R2.50** Revised. We made a rephrasing of the corresponding lines.

**Comment R2.51** *Line 117. Remove "and".*
**Response to R2.51** Revised.

**Comment R2.52** *Line 130. Remove "which is remote from significant sources of emissions." This is repeating what was said before.*
**Response to R2.52** Agreed.

**Comment R2.53** *Line 139. "which was housed in an"*
**Response to R2.53** Changed.

**Comment R2.54** *Line 142. "oxidation"*
**Response to R2.54** Word was deleted.

**Comment R2.55** *Line 145. "during which remaining oxidised Nr species"*
**Response to R2.55** Sentence was deleted.

**Comment R2.56** *Line 219. "was caused by"*
**Response to R2.56** Changed.

**Comment R2.57** *Line 249. "filling the gaps in the flux data."*
**Response to R2.57** Will be changed.

**Comment R2.58** *Line 274. "weighted using the actual land-use fractions" ?*
**Response to R2.58** Agreed.

**Comment R2.59** *Line 275. "when considering only deciduous"*
**Response to R2.59** Will be changed.

**Comment R2.60** *Section 3.1. Much of the section here and elsewhere should be put into past tense.*
**Response to R2.60** Agreed.

**Comment R2.61** *Line 303 and elsewhere. Please add charges to NO3- and NH4+ (NO3 is a radical).*
**Response to R2.61** Charges were added.

**Comment R2.62** *Line 305. Redundant "with"*
**Response to R2.62** Removed.

**Comment R2.63** *Line 308. "the relative contribution of NH3 is significantly higher"*
**Response to R2.63** Revised.

**Comment R2.64** *Line 310 and elsewhere. A colon is followed by lower case in English.*
**Response to R2.64** Revised.

**Comment R2.65** *Line 311 "done following the criteria mentioned"*
**Response to R2.65** Revised.

**Comment R2.66** *Line 380 & 447. Should be "consequently" instead of "consequentially"*
**Response to R2.66** Will be changed.

**Comment R2.67** *Line 384. Should the units here be "kg N ha-1 a-1"?*
**Response to R2.67** Yes, a N is missing here.

**Comment R2.68** *Line 391. "Clearly, ..."*
**Response to R2.68** Will be changed.

**Comment R2.69** *Figure 6. The colours between upper and lower CBT estimate seem to be reversed.*
**Response to R2.69** We agree. Colors will be switched.

**Comment R2.70** *Line 417 and also line 816. "the range of ..."*
**Response to R2.70** Will be changed.

**Comment R2.71** *Line 450. "LOTOS-EUROS states out NH3 ..." – meaning unclear.*

**Response to R2.71** Sentence will be modified as follows: "LOTOS-EUROS determines NH$_3$ as the main contributor to $\Sigma$N$_r$".

**Comment R2.72** *Line 479. "Apart from management events, fluxes above the arable ..."*
**Response to R2.72** Changed.

**Comment R2.73** *Line 528. "Munger et al. (1995) also made NOy flux measurements ..."*
**Response to R2.73** Sentence was deleted.

**Comment R2.74** *Line 607. "sometimes lead to non-stationarities"*
**Response to R2.74** Will be changed.

**Comment R2.75** *Line 612 "under certain circumstances"*
**Response to R2.75** Will be changed.

**Comment R2.76** *Conclusions. Re-introduce all acronyms, including Nr.*
**Response to R2.76** In the revised version, acronyms were mentioned in conclusions.

**Response to Reviewer 3**

**General Comments** *Wintjen et al. present an interesting and valuable data set on total nitrogen deposition to a forest spanning multiple years. The paper will be a worthy addition to N deposition literature, but would be improved by providing a few additional details and considering some additional analysis and interpretation.*

We thank the Reviewer for his/her comments and suggestions on this work. Since your comments and recommendations are discussed in the responses to Reviewer 1 and 2, we will add references to the given answers.

**Comment R3.1** *Page: 8 line 252-254.It would be helpful to provide a little more detail on the calculation of resistances beyond just giving a reference. The actual equation itself would be ideal, but at least note what input variables are used in the parameterizations so that readers can know what the calculations are based on without having to consult multiple sources from the literature.*
**Response to R3.1** We agree. We added a new chapter called "Determining deposition velocity and canopy resistance of $\Sigma N_r$ from measurements" to the revised manuscript. In this chapter, equations needed for calculating the deposition velocity and canopy resistance of $\Sigma N_r$ are given.

**Comment R3.2** *line 257. Here it notes that alternate data sources are used for missing NH3 and HNO3. Is it stated anywhere how the data sources compare to one another when there are simultaneous measurements? Readers need this to assess whether there is any bias in the gap filling? Showing or mentioning a direct comparison would complement the plots showing cumulative deposition computed from different approaches. The direct comparison of simultaneous concentrations removes any confounding influence of other inputs to the calculated fluxes.*
**Response to R3.2** Figure R13 shows a comparison of the $NH_3$ measurement techniques (see R2.35). In R2.3, a discussion is made on the agreement between the TRANC $\Sigma N_r$ and $\Sigma N_r$ derived from the DELTA samplers (see Fig. R9 and R10). We discuss the influence of micrometeorology on the MDV approach in R2.9.

**Comment R3.3** *23 Line 449.Here it concludes that radiation is the primary driver affecting the diel cycle of N deposition. How have you discounted the role of wind speed/turbulence intensity, which will covary to radiation, as an alternative? If you account for the turbulence contribution to deposition velocity based on resistance model and thus compute an apparent canopy resistance from the residual is there still a dependence on radiation?*
**Response to R3.3** Please see R1.47, R1.55, R2.2 and also the publication by Zöll et al. (2019). Turbulence was not identified as important driver for the $\Sigma N_r$ flux by the authors. In R2.2, a discussion of the resistances and $v_d$ is made (see Fig. R7 and R8).

**Comment R3.4** *Page: 24 line 574 Do you consider the role of humidity and temperature on the partitioning between gaseous NH3 and NH4 aerosol? The patterns imposed by stomatal opening and NH3 partitioning might be difficult to distinguish. The observed pattern would be consistent with shifting the equilibrium toward gaseous NH3 during the warm and dry daytime conditions.*
**Response to R3.4** $NH_4^+$ and $NH_3$ concentrations were obtained from DELTA measurements. During the warmer month, $NH_3$ concentrations were higher than concentrations measured for $NH_4^+$. From November to February, the situation was vice-versa. However, due to the denuder's low time resolution, we had no possibility to derive an influence of $NH_4^+$ and $NH_3$ on stomatal

processes, which happened on a shorter time-scale.

[revised manuscript text omitted]

---

## Editor Decision (ED1)

**Title:** Using "remote region (area)" instead of "low polluted area" would sound much better to my opinion.

Line 5: ….at a mixed forest exposed to low air pollution levels.

Line 15: … high solar radiation…

Line 19: No significant influence of temperature, humidity, friction velocity, or wind speed on ΣNr dry deposition sums were found.

> ➔ This is somewhat in contrast to what was mentioned in other places of the paper. These variables determine the deposition velocity and, hence, at the end also the total deposition. Maybe reformulate or delete.

Line 21: …half-hourly value…

Line 25: … to a remote forest ecosystem.

Line 37: nitric oxide (NO)

Line 39: …aerodynamic gradient method (AGM), please change throughout

Lines 79-80: These few long-term micrometeorological measurements of Nr species above forests were made more than 20 years ago and no recent reports on long-term flux measurements of Nr are currently available.

Line 83-84: As stated above, the outstanding benefit of TRUNC is…..

Line 85-86: I would combine this sentence with the scientific objectives mentioned in lines 100-104: please reformulate / list points (1), (2), (3) as scientific objectives.

Lines 105-108: Please delete these lines a they are not required here and can be misleading. I would just mention one sentence that a follow up paper will deal with…

Line 196: Additionally, fast-response measurements…. (delete…, too).

Line 251: … and associated descriptions are based on…

Line 293: … replace "or nitrogen aerosols" with "or related aerosol compounds"…

Lines 305-306: Please delete: "Further details about the implementation of these resistances in surface-atmosphere models can be found in van Zanten et al. (2010)."

Line 312: replace "A breakdown…" by "The contribution of individual nitrogen compounds to the total ΣNr concentration pattern is shown in Fig. 2, which…..

Line 315: NOx also showed a… (delete "too")

Line 318: The ΣNr concentration was 3.1….

Line 321: …in the annual pattern was reasonable…

Line 322: …with measurement height was observed.

Line 323-324: At 50 m the $NH_3$ concentration exceeded that at 30 m by 0.1 µg N m$^{-3}$.

Line 339:

I propose to use the expression diurnal cycles instead of daily cycles throughout the MS.

Line 325: The seasonal variations of the half-hourly ΣNr concentrations are represented by box-and-whisker plots including monthly medians in Fig. S3. (delete: Figure S3 shows monthly box plots of the concentrations.)

Line 327: Medians ranged between…

Figure 2 caption: …Missing $NH_3$ values from the DELTA measurements…. Numbers above the bars indicate the relative coverage of TRANC measurements during each exposure period.

Line 349: $NH_3$ also featured seasonal variations with….

Line 353: As shown in Fig. 2, ΣNr…

I would split the first results section in two parts:

3.1 Measured concentrations of individual reactive nitrogen compounds

Including Figures 1-3 and S1-S4

3.2 Measured exchange fluxes of total reactive nitrogen

Starting on page 14 (break at line 355)

Line 355-356: …on a monthly timescale….

Line 359: on a half-hourly basis…., On a monthly basis…

Line 360: According to Langford et al. (2015), the limit of detection (LOD) is calculated by multiplying the random flux error (95% confidence limit) with 1.96.

Line 364: This indicates that emission fluxes….

Line 365: In general, median deposition was within the same range for the entire campaign with only small seasonal differences.

Line 367-368: Median deposition was significantly increased from June 2016 till September 2016 than for the same period in 2017 and IQR and whisker also covered a wider range in 2016.

Line 374: Fig. 5 shows averaged daily cycles of measured ΣNr fluxes for every month.

Figure 5: Mean diurnal cycle of ΣNr fluxes (ng N $m^{-2}$ $s^{-1}$) based on half-hourly measurements for every month from June 2016 to June 2018. The shaded….

Line 374-375: In general, the ΣNr diurnal cycle exhibited low deposition or fluxes close to zero during nighttime/evening and increasing deposition during daytime. Deposition fluxes were…

Line 378: …with near-zero or small negative fluxes…

Line 379: … months were comparable.

Line 381-382: …was close to zero one year later.

Line 386: Again, the average standard error…

Line 390: The meaning of "From May to September, the curve was approximately bell-shaped." is unclear. Please clarify.

Line 393: 3.3 Controlling factors…

Line 396: "leading to a constantly low vd during the day (Fig. S10)." From Fig. S10 it is evident that vd even strongly decreases during midday, this should be mentioned (and explained in the discussion).

Line 399: and the concentration of ΣNr, especially changes in the concentration of the individual nitrogen compounds….

Line 410: For visualizing the impact of the concentration on $v_d$ (Fig. 6),…

Line 412: …increments of the ΣNr concentration…

Line 413: … on the ΣNr concentration…

Line 414: It demonstrates that the ΣNr concentration…

Line 419: …$v_d$ was more influenced by micrometeorological variables than by the ΣNr concentration.

Line 425-426: Combine 2 sentences, they should read as: "During winter (December, January, and February), vd was almost equal and even lower during the day, which resulted in a lower deposition of ΣNr."

Line 426-428: The sentence should read as:

The different shapes of the diurnal variations of vd could be induced by micrometerological variables, which change the composition of available ΣNr compounds during the day  and promote photosynthesis (e.g. stomatal uptake or release of $NO_2$ and $NH_3$).

Seinfeld and Pandis, 2006 is for sure not the appropriate literature here, please choose other more specific references (as in the discussion section).

Figure 7. Mean diurnal cycle of $v_d$ from May to September for low and high temperature (a), relative humidity (b), and concentration (c). Median….

Line 435: … lower relative humidity….

Line 437: During dawn/nighttime, deposition velocities exhibited no significant difference between the applied thresholds…. I can see a difference for the dry/wet leaf surface. Please double check this statement.

Line 439: …compared to the May to September period.

Section 3.3 should be changed to:

3.4 Dependence of ΣNr dry deposition sums on micrometeorological variables

Figure 8. … represented by box-and-whisker plots…

Eq. (3) (Pastorello et al., 2020) →please refer to the discussion section here.

Line 450: median deposition of  ΣNr  with….

Line 451: median deposition…→ please correct all instances

Figure 9. Annual ΣNr dry deposition shown as bar graphs

Line 459: … dry depositions sums…

Introduce new section after Line 466:

3.5 Wet and total nitrogen deposition

Line 473: In the second year, the contribution of dry deposition…

Line 476-477: Which was probably related to high $NH_3$ concentrations… For sure it was, you measured them, please refer to the corresponding Figure here.

Line 509: Thus, their influence on NOx measurements was most likely small.

515-516: DELTA measurements further suggested that the ΣNr concentration pattern was mainly influenced by gaseous Nr.

Line 521-522: Due to the reaction of $NH_3$ with $HNO_3$ and sulphuric acid particulate $NH_4^+$ is formed, available as $NH_4NO_3$ or $(NH_4)_2SO_4$.

➔  I would change the order of compounds here:

Explanation:

In chemical systems composed of $NH_3$, $HNO_3$ and $H_2SO_4$, the formation of non-volatile $(NH_4)_2SO_4$ is preferred. Only when $NH_3$ is available in excess of $H_2SO_4$ and when favourable meteorological conditions (low to moderate T and/or high RH) prevail, neutralization of $HNO_3$ vapor with $NH_3$ occurs (Trebs et al., 2005).

*Trebs, I., Metzger, S., Meixner, F.X. et al., 2005. The NH4+-NO3--Cl--SO42--H2O aerosol system and its gas phase precursors at a pasture site in the Amazon Basin: How relevant are mineral cations and soluble organic acids? Journal of Geophysical Research-Atmospheres, 110(D07303): doi:10.1029/2004JD005478.*

Line 523: … fine mode and associated with aerodynamic diameters….

Line 537: …, but were probably…

Line 553-554: …for instance by bidirectional exchange of $NH_3$ leading to both periods of net emission and deposition of ΣNr.

Line 567: Also, the $SO_2$ concentration was much larger…

Line 577: …resulting in a high vd, which is due to efficient turbulent mixing. Hence, even at low concentrations…

Line 578-579: In conclusion, particulate $NH_4^+$ was mainly responsible for the large ΣNr deposition due to its excess over aerosol $NO_3^-$.

I propose that the section on ΣNr emission and the influence of snow can be shortened. The English writing of this section must be improved (Lines 590-609).

Line 616: I think it is anyway highly unlikely that the concentration drives the deposition velocity.

However, the impact of increasing concentrations on….

Line 624: … was nearly zero and emission…

Line 631: … contribution of individually compounds do show a seasonal cycle. Since the ΣNr compounds differentiate in their vd,…

Line 635: …than of NO$_2$, but… than of NO$_2$ for woodland.

Line 636: …and 2.2 cm s$^{-1}$  (see Schrader…

Lines 637-643: Rewrite to:

However, variations in the composition of ΣNr may correlate with micrometerological parameters. For example, the formation of HNO$_3$ is correlated with Rg. The solar radiation responsible for the stomatal opening also promotes the formation hydroxyl radicals, which react with NO$_2$ to form HNO$_3$ (Seinfeld and Pandis, 2006). Tair influences the diurnal pattern of NH$_4$NO$_3$, which may also volatilize close to the surface due to the depletion of its precursors and in case the temperature gradient is large enough (Wyers and Duyzer, 1997; Van Oss et al., 1998). Thus, part of the NH$_4^+$ and NO$_3^-$ in the aerosol phase may be converted to NH$_3$ and HNO$_3$, which deposits faster to surfaces than aerosols.

Line 646: In conclusion, the variability…

Line 648-649: Delete: .

Line 656: …measured half-hourly values…

Line 657: … low-quality half-hourly values were effectively…

Line 660: Was there any footprint analysis performed or required due to fetch limitations? Could you comment on that? Maybe refer to previous publications.

Line 669: of turbulent motions…

Line 675: As shown in Fig. 8…

Line 679: …a certain half-hourly value was…

Line 688: …estimated dry deposition for…

Line 693: …has a distinct diurnal cycle.

Lines 701-702: Please delete:

Line 705: …total N deposition…

Lines 708-709: It suggests that the forest is currently not in a critical state in relation atmospheric N input.

→ I think this statement is incorrect. The N input was 10 and 12 kg N ha$^{-1}$ a$^{-1}$, which is within the range of the critical load.

According to the OECD, the critical load is defined as:

*Critical Load is the quantitative estimate of the level of exposure of natural systems to pollutants below which significant harmful effects on specified sensitive elements of the environment do not occur.*

According to my understanding, the forest is just at the limit of receiving too much Nitrogen from the atmosphere. This implies that N inputs should not increase in the next years.

Line 713: …above a protected temperate mixed forest, that is located in a remote area.

Line 721: …throughout the year.

Line 726: …periods of high solar radiation…

Line 727: seasonal changes in the concentrations of the ΣNr compounds,..

➔ Before is was written that ΣNr does not influence vd….

Please double check.

Line 728: From May to September,  vd was….

Line 732-733:

➔ This sentence does not make sense, please delete.

Line 735: No significant influence of micrometeorological parameters on estimated dry depositions sums was found.

➔ This sentence does not make sense and is in contrast to what was written before. (micrometeorology influences vd and therefore also the total N deposition)

Line 736-737:

➔ Please delete, note relevant here.

Please add information to the conclusion that dry deposition contributed 1/3 to the total N deposition.

**Rephrase Supplement:**

A1 Description of wet deposition measurements

Figure S3: …shown as box-and-whisker plots….

Figure S4. Mean diurnal cycle of ΣNr concentrations ($\mu$g N m$^{-3}$) based on half-hourly measurements for every month from June 2016 to June 2018.

Figure S5: … presented by box-and-whisker plots…

Figure S6. Mean diurnal cycle of $v_d$(ΣNr) (cm s$^{-1}$) based on half-hourly measurements for every month from June 2016 to June 2018.

Figure S11. Diurnal cycles…

Figure S13. ….Wind direction corresponds to values measured in three-hourly intervals.

---

## Author Response (AR2)

**Response to reviewers' comments – manuscript BG-2020-364 Forest-atmosphere exchange of reactive nitrogen in a low polluted area – Part I: Measuring temporal dynamics**

We thank again the anonymous referees for their comments to revised version. We recognized that the discussion about (effective) canopy resistances ($R_{c,eff}$) and deposition velocities ($v_d$) was too speculative and shortened it substantially. We added results on wet deposition measurements and clarified misleading statements to the DELTA denuders. Finally, we improved the discussion about the deposition occurred in February 2018 with own measurements on particulate nitrogen.

Comments of Referee 1 range from R1.1 to R1.26, Comments of Referee 2 range from R2.1 to R2. 73. Line numbers in the answers, where new information was added to the manuscript, refer to the original submitted version. The text which is enclosed by "…" is implemented in the manuscript.

**Response to Referee 1**

**General Comment:** *Based on initial reviews, the authors have eliminated the modeling component, which will be covered in a forthcoming separate manuscript, to focus only on the measurement component of the study. Many of initial review comments regarding the measurements were addressed but some technical weaknesses remain, as outlined below. Second, the overall writing of the manuscript, specifically the grammar and sentence structure, has not significantly improved. While some examples for improvement of the writing are included below, there are many more instances, the comprehensive correction of which is outside the scope of a scientific technical review. While I believe this dataset and its analysis can make a contribution to the literature, and is appropriate for Biogeosciences, treatment of the remaining technical issues (as outlined below) and readability must be addressed before the paper is suitable for publication.*

We thank the Reviewer for his/her comments on the revised version. We addressed all of your remarks and implemented your suggestions in the manuscript

**Comment R1.1:** *Section 2.2: The authors note different heights for the various measurements: 30m for ∫Nr and Delta, various heights for passive NH3, 30m for QCL NH3, and 50m for NO and NO2. The NO and NO2 measurements were used for assessment of Nr speciation. Did the authors investigate the potential magnitude of the NOx gradients between 30m and 50m? Since NOx appears to be the primary Nr component, some mention of the importance of differences in measurement height is warranted.*

**Response to R1.1:** Since no $NO_2$ and NO measurements were conducted at 30 m, no concentration gradient was calculated between 50 m and 30 m. "Seok et al. (2013) found highest $NO_x$ concentrations above the canopy but their concentrations differences were negligible. Since both measurement heights were above the canopy, no correction was applied to $NO_2$ and NO concentration measurements." We added these sentences to line to 195 and discussed the influence of the different heights with regard to $NO_2$ and NO at line 527ff.

**Comment R1.2:** *Line 155: Can the authors give an explanation for the NH4NO3 efficiency of 142%?*

**Response to R1.2:** Marx et al. (2012) provided reasons for their results. $NH_4NO_3$ is semi-volatile under ambient conditions and can evaporate to $NH_3$ and $HNO_3$. Both gases are detected by the TRANC and influence the conversion efficiency of $NH_4NO_3$. Ambient air was used to clear the analysis chamber of the CLD and to transport the aerosol, resulting in a $N_r$ background ranging from 20 to 39 ppb. The latter was corrected by the authors. For further details we refer to the publication of Marx et al. (2012).

**Comment R1.3:** *Line 182: Change "exposition" to "exposure".*

**Response to R1.3:** Done.

**Comment R1.4:** *Line 198: A brief description of the methods for wet and bulk deposition is needed, including organic nitrogen. This can be added to Supplemental.*

**Response to R1.4:** "Wet-only and bulk deposition were collected by four samplers, one wet-only and three bulk samplers, at an open site. The measurements took place in southwest direction of the tower (approx. 1.3 km). Bulk samplers had a funnel opening of 321 $cm^2$ at 1.25 m above ground. The automatic wet-only sampler (NSA 181K – cooled, Eigenbrodt, Königsmoor, Germany) had a funnel opening of 500 $cm^2$ at 2 m above ground. During the weekly sampling intervals, precipitation samples were kept dark and cool (<4°C). After sampling they were filtered (< 0.45 µm, Whatman) and cooled at 2 to 4°C without chemical preservation/treatment until analysis.  No biocides were used during sampling because denitrification was unlikely due to the short exposure time and permanent cooling. In fact, we found very low carbon concentrations and no nitrite as an intermediate product of denitrification in the precipitation samples.

$NH_4^+$ and $NO_3^-$ were analyzed following DIN EN ISO 10304-1. Determination of total wet N was done according to DIN 38409-27 and EN 12260. Dissolved organic nitrogen is calculated by subtracting $NH_4^+$-N and $NO_3^-$-N from total wet N." We provided this description as a supplement and named it A1.

**Comment R1.5:**  *Line 238: "Pump efficiency was controlled...". Do the authors mean "Pump efficiency was assessed.."? What does it mean to control the efficiency monthly?*

**Response to R1.5:** "We checked the pressure in the sample cell of the CLD during each, at least monthly, site visit. If the sample cell pressure was outside the allowed range, tip seals of the pump were replaced." We rephrased the sentence according to this response.

**Response to R1.6:** *Lines 308-320: Consider combining into a single paragraph.*

**Comment R1.6:** We decided to delete those lines since they were no longer needed for the discussion.

**Comment R1.7:** *Line 339: Change the phrasing of "almost similar". "Almost" appears in other places in the manuscript and should be avoided.*

**Response to R1.7:** We deleted the word "almost" at several places or changed the phrasing.

**Comment R1.8:** *Caption Figure 3: Change "exposition" to "exposure". Consider indicating the year along the x-axis.*

**Response to R1.8:** Agreed. We changed the word and labels of the x-axis for better readability.

**Comment R1.9:** Line 379: *I think this sentence is not necessary.*

**Response to R1.9:** Agreed. We deleted this sentence.

**Comment R1.10:** *General comment: Words such as "mostly", "almost", "slightly", "mainly" are used throughout the manuscript. In general, they should be used very infrequently in the context of data reporting/interpretation.*

**Response to R1.10:** Agreed. We reduced the usage of these words.

**Comment R1.11:** *Figure S8: The caption says the period is May-September. Please correct.*

**Response to R1.11:** Corrected. It should be December to February.

**Comment R1.12:** *Line 411: "The analysis of vd and corresponding fluxes show that their diurnal pattern was characterized by lower deposition during the night and highest values around noon, in particular from May to September (Fig. 6 and Fig. S6)" But this was not the pattern during winter, which showed low vd at mid-day (Fig. S8). Please clarify.*

**Response to R1.12:** "From May to September, a clear diurnal pattern was found for $v_d$ and their corresponding fluxes (Fig. 6 and Fig. S6). It was characterized by lower deposition during the night and highest values around noon (Fig. S9). During winter, deposition fluxes were close to zero and showed no diurnal variation leading to a constantly low $v_d$ during the day (Fig. S10)." We changed the sentence according to this response.

**Comment R1.13:** *Line 438: "vd and Rc,eff determined during rain were treated separately." Reviewers 1 and 2 both questioned the quality of the TRANC flux measurements during rain and the suitability of the EC method (or any micromet flux method) during rain as the washout process introduces an additional sink below the EC measurement height and therefor a height dependent flux. I believe the*

*EC measurements conducted during rain should not be included in this analysis. If the authors retain them, the validity of the EC method during rain must be justified.*

**Response to R1.13:** We agree that a height dependent flux could be induced due to precipitation since many $N_r$ compounds are water soluble. In order to show the impact of precipitation on measured $\Sigma N_r$ deposition velocities, we made the analysis of controlling factors for deposition velocities and effective canopy resistances during active and no precipitation separately. Possibly, there was a misunderstanding in treating the suggestions to precipitation in the first review round. We only applied a precipitation filter on Figs. 8 and 9 in order to strengthen process understanding of $\Sigma N_r$ flux measurements.

"In order to avoid uncertainties due to the washout process as it introduces an additional sink below the measurement height leading to a height dependent flux, we applied a precipitation filter on $\Sigma N_r$ flux measurements (IV)." We deleted discussions and subplots related to the impact of precipitation on the $\Sigma N_r$ exchange. Since a precipitation filter was introduced (line 232), annual budgets were higher since lower $\Sigma N_r$ fluxes were excluded from analysis.

**Comment R1.14:** *Figure 7 caption: "Dependency of measured concentrations on corresponding ΣNr fluxes…". Consider changing to "Relationships between measured concentrations and corresponding ΣNr fluxes…".*

**Response to R1.14:** We applied your suggestion to Figs. 4, 5, and 6. Please note that Fig. 7 of the original submitted manuscript was deleted.

**Comment R1.15:** *Table 2 and associated paragraphs. Because the TRANC is not Nr species specific, its primary strength is in quantifying the total Nr dry deposition to facilitate a total deposition calculation when combined with wet deposition. The authors have done this but some additional detail on the variability of the relative fractions of wet versus dry deposition would be helpful. Can the authors add a table or pie charts showing the total wet and dry fluxes and fractional contributions of wet versus dry deposition by year and summarized by overall annual and seasonal periods?*

**Response R1.15:** We appreciate the Reviewers suggestion. Figure 1 shows relative fractions of wet and dry deposition to total deposition for each season and for both measurement years.

[Figure]

**Figure 1** Contribution of dry and wet deposition to total deposition for each season and both measurement years labeled from (a) to (f).

In case of seasonal contributions, dry deposition contributed approximately one third to total deposition except for winter. In the second year, contribution of dry deposition was higher than in the first year. Higher fractions of dry deposition were related to the large dry deposition occurring in late February 2018. Table 2 shows seasonal and yearly deposition sums of dry and wet deposition measurements. Please note that the sum of all seasons corresponds to the sum of both measurement years.

**Table 1** Annual and seasonal sums of dry deposition estimates (DD) and $NH_4^+$-N, $NO_3^-$-N, dissolved organic nitrogen (DON), and the resulting total wet deposition (TWD) from wet deposition samplers (bulk (BD) and wet-only (WD)) in kg N ha$^{-1}$ period$^{-1}$. Uncertainties in dry deposition estimates are related to applied gap-filling technique.

| Time | DD [kg N ha$^{-1}$ period$^{-1}$] | WD [kg N ha$^{-1}$ period$^{-1}$] | | | | BD [kg N ha$^{-1}$ period$^{-1}$] | | | |
|---|---|---|---|---|---|---|---|---|---|
| | | $NO_3^-$-N | $NH_4^+$-N | DON | TWD | $NO_3^-$-N | $NH_4^+$-N | DON | TWD |
| Winter | 2.0 | 1.5 | 0.9 | 0.4 | 2.8 | 1.7 | 1.3 | 0.5 | 3.5 |
| Spring | 2.2 | 1.8 | 2.3 | 0.1 | 4.2 | 1.9 | 2.4 | 0.1 | 4.4 |
| Summer | 2.0 | 1.9 | 2.6 | 0.2 | 4.7 | 1.6 | 2.2 | 0.6 | 4.4 |
| Autumn | 1.7 | 1.5 | 1.4 | 0.6 | 3.5 | 1.4 | 1.4 | 0.6 | 3.4 |
| June 16 – May 17 | 3.8 | 3.8 | 4.2 | 0.4 | 8.4 | 3.5 | 4.2 | 1.0 | 8.7 |
| June 17 – May 18 | 4.0 | 2.9 | 3.1 | 0.9 | 6.9 | 3.0 | 3.1 | 0.9 | 7.0 |

Small seasonal and annual differences in dry deposition were determined (approx. 200 g N ha$^{-1}$ period$^{-1}$). Total seasonal and annual uncertainties related to gap-filling (Eq. (3)) were between 7 and 21 g N ha$^{-1}$ period$^{-1}$. Due to the large fluxes in late February 2018, dry deposition and its uncertainty were remarkably high during winter. Total wet deposition (TWD) was highest in spring and summer. During those periods, $NH_4^+$-N contributed most to TWD which was probably related to high $NH_3$ concentrations. Interseasonal differences for $NO_3^-$-N were found but were lower compared to changes in $NH_4^+$-N. DON deposition was lowest and was between 0.1 and 0.6 kg N ha$^{-1}$ a$^{-1}$. Overall, differences in TWD for both sampler types were less than 300 g N ha$^{-1}$ a$^{-1}$ except for winter.

We added Fig. 1 to the Supplemental Material, replaced Table 2 of the manuscript by Table 1 of the response, and rephrased the description beginning at line 500.

**Comment R1.15:** *Line 507: "In total, we got a total nitrogen deposition of…" Please consider changing to "Total wet + dry deposition was equivalent to…".*

**Response to R1.15:** Done.

**Comment R1.16:** *Line 510: Change the section title to reflect the structure of the section, which is ordered as concentrations, fluxes, then deposition velocity.*

**Response to R1.16:** Agreed.

**Comment R1.17:** *Line 523: Please consider using a word other than "neutral" in this sentence, as neutral is commonly used to characterize atmospheric stability and is confusing as currently used.*

**Comment R1.17:** Supposedly, you refer to line 533. We replaced "closer to neutral conditions" by "closer to zero".

**Comment R1.18:** *Line 547: The explanation of high deposition in Feb 2018 being driven by NH4+ is not convincing (i.e., high SO2 corresponding to high ammonium sulfate/bisulfate concentrations). In other winter periods, NH4+ makes a relatively small contribution to Nr (Figure 3). Additionally, given the lower deposition velocity of particles a very large concentration of NH4+ would be needed to explain this much larger deposition flux. Such a large increase in NH4+ should have a regional signal. Can data from other monitoring sites be used to assess larger scale patterns in atmospheric chemistry during this period that could shed light on what could be driving the large increase in Nr at the authors study site? Some additional analysis may be possible here.*

**Response to R1.18:** After carefully reflecting the reviewer's comments to that aspect, we agree the explanation of the deposition in February 2018 is insufficient. For January 2017 (only $NH_3$) December 2017, March 2018, and April 2018 no DELTA measurements were available since the denuder pump was not working properly. For those months, averages from previous years were used to fill the gaps. Actually, we had measured delta concentrations for February 2018. Thus, Figures 2, 3, S1 and their corresponding descriptions were adjusted. We searched for air pollution stations, which are in close proximity to measurement station and are exposed to a similar pollution climate. Our measurement

site is located in a remote area and represents a rural background station in the air pollution network hosted by the German Environment Agency. However, there is no nearby measurement site in a radius of 50 km representing rural background and conducting $NH_4^+$ measurements in the network. In 20 km distance, a station integrated in the Czech Air pollution network is measuring $NH_4^+$. Unfortunately, they provide no measurements of $NH_4^+$ during February 2018.

"During the exposure period of the DELTA samplers, we found 0.96, 0.17, 0.37, 0.27, and 1.70 µg N m$^{-3}$ for $NH_4^+$, $NH_3$, $NO_3^-$, $HNO_3$, and $NO_x$, respectively. The aerosol concentrations were exceptionally large in February 2018, which have affected these averages considerably. Average $NH_4^+$ concentration during winter excluding February 2018 was only 0.38 µg N m$^{-3}$ in comparison to 0.96 µg N m$^{-3}$ for February 2018. The concentration in this month results in a $NH_4^+$ concentration 2.5 times higher than the average. Also, $SO_2$ was much larger concentrations (1.54 µg m$^{-3}$) in this month compared to the other winter month (0.37 µg m$^{-3}$). Figure 2 shows the relative contributions of each $N_r$ compound for February 2018 compared to averaged fractions during winter excluding February 2018.

[Figure]

**Figure 2** Relative contribution of concentrations for $NO_x$, $NH_3$, $HNO_3$, $NO_3^-$, and $NH_4^+$ to $\Sigma N_r$ estimated from DELTA and $NO_x$ measurements for winter and separately for February 2018.  $NO_x$ measurements are averaged to exposure periods of the DELTA samplers.

During February 2018, $NH_4^+$ made a significant contribution to the $\Sigma N_r$ concentration. The measured $NH_4^+$ value is an integrated value over approximately one month. Thus, daily contributions of $NH_4^+$ could have been even higher. Earlier studies by e.g. Wolff et al. (2010) report events with large aerosol deposition. During their campaign, wind speeds were relatively high. Largest aerosol deposition occurred during dry conditions, e.g. low $RH$, no rain, and high visibility. Figure 3 shows micrometeorological parameters, deposition velocities, and gap-filled $\Sigma N_r$ fluxes from the 12 February to 6 March. Large deposition fluxes were accompanied by high $wsp$ (wind speed) and $u_*$ values, high $R_g$ indicating high visibility, and low $RH$. The observed conditions are typical for cold air streams with high aerosol loads coming from North east and led to a reduction in turbulent resistances resulting in a high $v_d$, which is allowed by turbulence. Hence, at low concentrations of $NH_4^+$ significant aerosol deposition is possible if $R_a$ and $R_b$ are reduced.

In conclusion, $NH_4^+$ aerosols, ammonium sulfate and nitrate, were most responsible for $\Sigma N_r$ deposition due to their excess over $NO_3^-$. Since we had no high-resolution flux measurements of any $\Sigma N_r$ compound during that time, we have no evidence which aerosol predominated the $\Sigma N_r$ flux. "

We rephrased the discussion about the deposition event from lines 548 to 565 by this response. Figure 2 was implemented in Sec 4.1 and Figure 3 was added to the Supplemental Material.

[Figure]

**Figure 3** Recorded air temperature ($T_{air}$.), relative humidity (RH), global radiation ($R_g$), wind speed (*wsp*), friction velocity ($u_*$), $v_d$, and gap-filled $\Sigma N_r$ flux as 3-h running mean from 16 February to 6 March 2018. Wind direction corresponds to values measured at 3-h time stamps.

**Comment R1.19:** *Line 578: Change "been happened" to "been occurring".*

**Response to R1.19:** Agreed.

**Comment R1.20:** *Line 599: Section 4.1 is structured as a summary of concentrations, fluxes, then deposition velocity. The paragraph beginning on line 599 describes concentrations and should be moved to follow (or integrated with) the first paragraph in the section (beginning at line 511).*

**Comment R1.20:** We agree. We integrated the corresponding lines 599-616 to the first paragraph of Section 4.1 (line 527).

**Comment R1.21:** *Line 614: Regarding the DELTA, the authors include these statements: "Concentration peaks could not be collected sufficiently by the coated surfaces. The latter are exposed to environmental influences like temperature and moisture, and their sensitivity may reduce over time." The authors appear to be saying that the DELTA measurements are biased low because short term peaks in concentration are under sampled (inefficiently collected by denuder coatings) and due to loss of denuder collection efficiency over time. I have not read this previously about the DELTA denuders. The authors need to support these statements with evidence or remove them.*

**Response to R1.21:** In this paragraph, we mentioned possible reasons for differences between denuder and TRANC measurements. "In addition, higher oxidized compounds like $N_2O_5$ or peroxy acetyl nitrates could not be collected by DELTA, but probably converted by the TRANC. Issues in the temperature stability or CO supply leading to instabilities in the conversion efficiency of the TRANC may be responsible for disagreements to the collection efficiency of the denuders. The difference between TRANC and DELTA concentrations is also related to their aerosol cut-off sizes (R2.13 and R2.18). A key uncertainty was the data coverage of the TRANC, which was at 78% on average during the exposure periods."

Still, the coating of denuders can be washed off if water is sucked in by the pump. In order to avoid that, we mounted a funnel at the opening of the denuder. During the measurement campaign, no water was found in the denuders. Thus, a loss of collection efficiency due to water can be excluded. Following the Reviewers suggestion, we decided to delete these lines (614-616) and added the information given in this response to that line 614. Please note that the discussion was shifted to line 527.

**Comment R1.22:** *Line 631: Change "not considered" to "not be considered".*

**Response to R1.22:** Done.

**Comment R1.23:** *Line 639: "Consequently, a second mechanism, the stomatal compensation point firstly proposed by Farquhar et al. (1980) likely controls the uptake of the Nr compounds." This statement is false as written. The stomatal compensation point does not control the exchange of all Nr compounds. The authors should be specific about which Nr compounds they are referring to here (i.e., NO2 and NH3?). If the authors are suggesting a compensation point driven exchange of NO2 then an appropriate citation is needed.*

**Response to R1.23:** We agree that the stomatal compensation point is not responsible for all $N_r$ compounds and further clarification is needed. Zöll et al. (2019) examined $\Sigma N_r$ fluxes at the same site during summer and found a high contribution of $NH_3$ to the $\Sigma N_r$ concentration. Since $v_d$ of $NH_3$ is higher than $NO_2$ above forest, the saturation point in the light response curve of $\Sigma N_r$ is probably caused by the stomatal compensation point of $NH_3$.

Please note that line 639 was deleted.

**Comment R1.24:** *Line 652: "Micrometeorological parameters were controlled by natural processes." This statement is unnecessary.*

**Response to R1.24:** Agreed. We deleted the sentence.

**Comment R1.25:** *Line 709: A paragraph should not begin with "However".*

**Response to R1.25:** Agreed.

**Comment R1.26:** *Line 803: "It shows that deposition of sedimenting organic and inorganic particles is not relevant at the site." What does this statement mean? It is unclear as written.*

**Response to R1.26:** We found that differences in wet deposition estimated from bulk and wet-only samplers were negligible. Small differences could be induced by the sedimentation of organic and inorganic dusts or by the dry deposition of gases like $NH_3$ or $HNO_3$ (Staelens et al. 2005). We rephrased the statement according to this response.

**Response to Referee 2**

**General Comment:** *The separation of the paper into a paper mainly reporting the measurements and a paper on modelling of the exchange has made this paper more accessible. However, the revised paper now covers much of the same ground in the discussion as the paper by Zoll et al. (2019). I found it rather disingenuous that the paper does not point out right at the beginning that it presents a continuation of the time-series of Zoll et al. (2019), taken at the same site, which is not spelt out until line 651. There is little point in re-iterating the findings of Zoll et al. to the extent done here and much of the Discussions section is rather speculative. I suggest that the analysis and discussion gets more focussed on the presentation of the annual budgets, cutting the paper by about 1/3. This is particularly the case as I do not believe that either the present paper nor the analysis by Zoll et al. provides insights into the mechanistic controls. Whilst the neutral network analysis (and the correlation analysis of the present papers) can identify associations, they cannot identify causality.*

We wrote in the introduction that Zöll et al. (2019) conducted flux measurement of $\Sigma N_r$ with the same instrumentation at the measurement site. They selected a short time period from the 2.5 measurement campaign (14 July to 30 September 2016) for identifying links $\Sigma N_r$ between $CO_2$ using artificial neural network approach. In addition, their post-processing and quality-selection was different. Thus, the term continuation may be misleading. Still, we agree that arguments in Sec 4.2 are similar to Zöll et al. (2019) and the subsequent discussion on canopy resistances, stomatal, and non-stomatal deposition is rather speculative. Thus, we shortened Sec. 4.2 substantially according to your comments.

We agree that the analysis of mechanistic controls is done for individual compounds and compared to results of resistance models. Based on the reviewers' suggestions, we decided to remove the discussion about the resistance analysis, integrated your suggestions to the dependence of deposition velocities on their concentrations to the manuscript, and shifted the focus of this manuscript on the determination of annual budgets.

We addressed all of your remarks and implemented your suggestions to scientific and technical issues in the manuscript.

**Major comments**

**Comment R2.1:** *At several places (starting with the opening two sentences and final sentence of the abstract) the paper overstates the utility of the Nr measurements for the development of model improvements / parameterisations. Because it is unknown which Nr compounds dominate the flux during any particular 30-minute flux period, any parameterisation of the net exchange is not transferable to other situations subject to another compound mix. I am not saying the analysis of the Vd and Rc of the net exchange is not worth doing but the paper should point out more clearly (e.g. lines 60-66) that the main utility of the TRANC is to quantify net dry deposition inputs of Nr with a single instrument (rather than a suite of instruments for each compound individually) and that the analysis of the net exchange parameter is a by-product.*

**Response to R2.1:** We agree that measuring only the $\Sigma N_r$ exchange provides no information about the actual flux contribution of each compound – a main drawback of the TRANC. Thus, a parametrization of the net exchange is not very meaningful. However, a comparison of modeled and measured deposition velocities of $\Sigma N_r$ could hint on deficits in deposition modeling. For example, it is possible to compare Fig 9. to $\Sigma N_r$ deposition velocities determined with inferential modeling. Thus, the analysis of the net exchange should be a point of interest but treated as an additional outcome. We agree that the corresponding lines in the abstract provide a false impression about the utility of TRANC measurements.

We added the main utility of the TRANC, which is the determination of the annual dry deposition and temporal dynamics of $\Sigma N_r$ with one instrument, to lines 60-66 and rephrased corresponding lines in the abstract (lines 1-3 and 32-33). We further focused the evaluation of the TRANC measurements on annual budgets and removed the discussion on effective canopy resistances ($R_{c,eff}$). The latter was removed since the quasi-laminar resistance ($R_b$) is not known (see R2.4). Still, we found that $v_d$ was influenced by micrometeorological parameters but not driven by the overall concentration (see R2.3).

**Comment R2.2.** *As mentioned above, the paper does not sufficiently clearly distinguish between associations and correlations on the one hand and causes / drivers on the other. Radiation, turbulence and temperature (and sensible heat flux) are highly correlated with each other (Figure S7) and it is impossible to decide which is the mechanistic driver and I am not convinced that vd is controlled by the plant activity rather than u\* (Line 435-436) and other drivers that correlate with Rg. From what we know of the exchange of the individual Nr compounds, stomatal exchange will be important for NO2 and its importance is highly variable for NH3 as pointed out in the paper. However, it is not so important for HNO3 and NH4+/NO3-. However, Rg will change Nr composition over the day with HNO3, a particularly fast depositing compound, typically peaking at midday (again related to Rg, this time via photochemical production), and it will produce a diurnal pattern on the effect of NH4NO3 volatilisation, which deposits fast only during daytime when temperature gradients are large. I therefore can't see that the measurements prove that stomatal conductance is the main controller of the Nr flux (Section 4.2.1). Stating that u\* does not affect the flux would be saying that Ra and Rb do not exist. If this analysis were done on Rc or 1/Rc at least the influence of turbulence would have been removed.*

**Response to R2.2.** We agree that the differentiation in associations and correlations was not done clearly. We totally agree that it is impossible to state global radiation, turbulence or temperature as the mechanistic driver of the $\Sigma N_r$ flux due to their high correlation with each other. We carefully checked the manuscript for corresponding cases and corrected them. We further agree that the role of $u_*$ as a control for $v_d$ and the flux has to be rephrased. The analysis of $v_d$ vs concentration showed that $\Sigma N_r$ concentrations did not correlate with their $v_d$ (see R2.3). We corrected the corresponding lines about plant activity as control (line 435-436). From an analysis of the net flux, we cannot examine stomatal or non-stomatal controls since fluxes of individual compounds are not known and highly variable during the day. Thus, we removed sections (Sec. 4.2.1) and corresponding lines about stomatal control and non-stomatal controls (lines 687-747) on the $\Sigma N_r$ flux. As written in R2.1 and R2.4, the discussion about $R_{c,eff}$ was removed.

**Comment to R2.3:** *In this context I do not find the analysis of the controlling factors for the flux very helpful. The flux would be expected to be affected by u\* (Fig. 7, Table 1 and associated text) as it still*

*contains the control via Ra and Rb. Also, the authors seem to try to convey that the slope changes with u\*. Except for the possibility of a non-zero intercept, the slope is actually vd. Thus, the authors should either plot vd vs Nr concentration or the ratio Vd/u\* against Nr concentration. They might also want to consider binning data according to y-values rather than showing raw data to convey a clearer message. This has implications for the discussion in Section 4.2.1. I fail to see why the lack of correlation between flux and concentration within a u\* class suggests that u\* is not a driver (line 645ff). Surely, it would suggest that the concentration is not the driver.*

**Response to R2.3:** We highly appreciate the Reviewers' suggestions to Fig. 7. Actually, the plot flux vs concentration was requested by Reviewer 1 during the first review round. Still, we note that the conclusions drawn from Fig. 7 were incorrect and we agree that a plot of $v_d$ vs concentration or the ratio $v_d/u_*$ vs concentration surely shows if the concentration is a driver of the $\Sigma N_r$ exchange. We removed lines 417-430 and replaced them by the text and figures given below. Figures 4 and 6 were provided as supplemental material. Figure 5 was added to the manuscript.

"In order to investigate the influence of $u_*$ on the $\Sigma N_r$ exchange, Fig. 4 illustrates the dependency of $v_d$ on $u_*$ for deposition and emission fluxes during day and night. The $R_g$ threshold for day and nighttime fluxes was set to 10 W m$^{-2}$. For better visibility, we binned data in 0.1 m s$^{-1}$ increments of $u_*$. Since bins are not equal in size, we added corresponding half-hourly fluxes to the plots. Red dots represent averages of each bin and error bars correspond to their standard error.

[Figure]

**Figure 4** Relationships between measured $u_*$ and corresponding $\Sigma N_r$ $v_d$ separated in emission and deposition during day ((a) and (c)) and night ((b) and (d)). Half-hourly data is displayed in black, red dots represents averages binned in increments of 0.1 m s$^{-1}$. Error bars indicate the standard error of the averages. The threshold for identifying day and nighttime $v_d$ was set to 10 W m$^{-2}$. $r$ represents the measure of correlation evaluated for the binned data.

We found that $v_d$ increased slightly with $u_*$ due to dependency of $v_d$ on $R_a$ and $R_b$. The latter are proportional to the inverse of $u_*$ suggesting that the increase with $u_*$ should follow a power law. In case of particles, linear relationships between $u_*$ and $v_d$ were found by Gallagher et al. (1997); Lavi et al. (2013); Donateo and Contini (2014). A relationship between $v_d$ and $u_*$ seems to exist as suggested by the correlations ($r$), but no clear functional relationship could be identified due to the large scattering of half-hourly $v_d$.

For visualizing the impact of concentration on $v_d$ (Fig. 5), we plotted $\Sigma N_r$ concentration against the ratio $v_d/u_*$ in order to reduce the influence of $R_a$ and $R_b$ on $v_d$. The threshold for $R_g$ was set to 10 W m$^{-2}$, and we binned data in 0.5 µg N m$^{-3}$ increments of $\Sigma N_r$ concentration.

[Figure]

**Figure 5** Relationships between measured $\Sigma N_r$ concentrations and corresponding ratios $v_d/u_*$ separated in emission and deposition during day ((a) and (c)) and night ((b) and (d)). Half-hourly data is displayed in black, red dots represents averages binned in increments of 0.5 µg N m$^{-3}$. Error bars indicate the standard error of the averages. The threshold for identifying day and nighttime $v_d$ was set to 10 W m$^{-2}$. $r$ represents the measure of correlation evaluated for the binned data.

It is obvious that $v_d/u_*$ exhibited no significant dependence on $\Sigma N_r$ concentration as shown by the low values for $r$. The ratio appeared to be constant across the (entire) concentration range. It demonstrates that $\Sigma N_r$ concentration had no significant influence on their $v_d$. In case of particles, the ratio $v_d/u_*$ depends on Obukov-Length ($L$) and particle size according to Gallagher et al. (1997) and Lavi et al. (2013). In case of deposition fluxes measured during daytime, we found that the ratio decreased for $-0.2 > L^{-1} < 0$ up to a minimum if $L^{-1}$ reaches zero (neutral stratification) (Fig. 6).

[Figure]

**Figure 6** Relationships between $L^{-1}$ and corresponding ratios $v_d/u_*$ separated in emission and deposition during day ((a) and (c)) and night ((b) and (d)). Half-hourly data is displayed in black, red dots represents averages binned in increments of 0.02 $m^{-1}$. Error bars indicate the standard error of the averages. The threshold for identifying day and nighttime $v_d$ was set to 10 $W\ m^{-2}$. $r$ represents the measure of correlation evaluated for the binned data.

This relationship was observed by Gallagher et al. (1997) and Lavi et al. (2013). Although the scattering of half-hourly ratio is large, the decrease of the ratio with increasing $L^{-1}$ as well as the dependence of $v_d$ on $u_*$ demonstrate that $v_d$ had a higher affinity to micrometeorological parameters than to the $\Sigma N_r$ concentration.

From the analysis of the figures 4, 5, and 6, it is impossible to state $u_*$ or $L$ as the controlling variable of the $\Sigma N_r$ exchange since turbulence, stratification, $R_g$, sensible heat flux, air temperature, and relative humidity are highly correlated with each other" as visualized by Fig S7. Thus, the dependence on $u_*$ could also be related to effects of the sensible heat flux, $T$, or $R_g$ and it is impossible to decide which is the mechanistic micrometeorological driver of the $\Sigma N_r$ flux.

According to these results, we removed the entire discussion of Sec 4.2.

**Comment R2.4.** *Although it theoretically provides more insights, the Rc analysis is quite uncertain due to the calculation of Rb. On the one hand the authors attempt to calculate an Rb that is weighted by the different compounds, on the other hand they set Rb for particles to 0 so that their full interaction with the canopy enters Rc. This is a crude approximation because the authors only have long-term information on composition (rather than half-hourly) and, numerically, the weighting should be done according to the compound contribution to the flux rather than the concentration. Moreover, the statement that particles are not subject to an Rb term (line 285) is incorrect. Rb describes the resistance posed by the laminar sublayer resistance and this is in fact larger for particles than it is for gases. However, the concept behind the terminology of Rb is that of Brownian diffusion, whilst particles have other mechanisms (interception, impaction, gravitation settling) to overcome this boundary layer in addition to diffusion (which is very ineffective for all but the smallest particles). Thus, for particles, the concept of Rb is usually replaced with that of Vds = Vd(z0) = 1/(1/Vd-Ra). The current approach followed*

*in the paper therefore derives an Rc that is a combination of different elements that mean different things for different compounds. This highlights again the limitations of the total Nr flux for mechanistic analysis (point 1 above).*

**Response to R2.4:** We thank the Reviewer for his/her remarks to the implementation of $R_b$ in the resistance analysis. Based on the analysis of the DELTA denuders, we set $R_b$ for particles to zero since generally their contribution to $\Sigma N_r$ concentrations was relatively low compared to gases. We agree that setting $R_b$ to zero for particles is not very meaningful based on observations on aerosol deposition and its implementation in current resistance models. We agree that the weighting should be done for the contribution of each compound to the $\Sigma N_r$ flux. Due to these limitations, we decided to remove the entire analysis and discussion on $R_c$ (Sec 4.2.1 to Sec. 4.2.3) and related paragraphs in the introduction (lines 66-91) and theoretical background (lines 272-320). As written in R2.1, the analysis of the net exchange of $\Sigma N_r$ is an additional product. Therefore, we shifted the focus towards the annual budgets and shortened the discussion about micrometeorological influences on $v_d$.

**Comment R2.5**. *It is similarly incorrect that the flux pattern of "nitrogen aerosols … is driven by Ra" (line 317). In fact vd for particles tends to be more reduced compared with 1/Ra than that the vd of gases.*

**Response to R2.5:** We agree that the statement is incorrect since it ignores the contribution of the surface to $v_d$ of aerosols. We deleted the sentence.

**Other scientific comments**

**Comment R2.6:** *Throughout the manuscript it is not clear to me which results/ figures are based on u\* filtered data and which not. Please clarify throughout. For deposition u\* filtering introduces a bias to the remaining data, removing preferentially small fluxes.*

**Response to R2.6:** In line 231, we wrote that a $u_*$-filter was applied to the measured fluxes. Thus, figures 5, 6, 8, 9, S5, S6, S9, S10, S12, S13, and associated text are based on $u_*$ filtered data. In Section 3.3, we discussed the effect of a turbulence filter on nitrogen dry deposition estimates. In the first review round, it was mentioned that a $u_*$-filter will remove preferentially small fluxes. We calculated annual dry deposition for two flux data sets with and without $u_*$-filter. On both datasets, the Mean-Diurnal-Variation (MDV) technique was applied as gap-filling approach. As visualized in Fig. 10, the effect of the $u_*$-filter is present but within the uncertainty range of the gap-filled fluxes. The comparison of different MDV approaches was done for both flux data sets as written in the caption of Fig. 11. We clarified the description of the flux filtering (line 233 and 251).

**Comment R2.7**: *I cannot fully follow the alternative implementation of the MDV in which you consider temperature, humidity and precipitation (lines 485ff). The introduction of the approach is not very clear and should probably be moved the methods section anyway. Could you please go through the English. "Dry deposition without restriction" (line 485) is not very meaningful. You probably mean "agreed within +/- 3C" rather than "varied by". It would probably make the section more readable if you gave this implementation a name. What about "stratified MDV" or "conditional MDV". Overall, I wonder*

*whether it would make more sense to apply the MDV gap filling to vd rather than fluxes as it is the exchange mechanism that is impacted by the meteorology rather than the concentration. Clearly, this would only work for periods for which you have concentration data.*

**Response to R2.7**: We agree that the description of the alternative MDV approach needs to be clarified. Yes, the wording "agreed within +/- 3°C" is meant here. In the revised version, we moved the description to the method section (line 251), improved its readability, and entitled the MDV approach with additional micrometeorological criteria as "conditional MDV" (CMDV) according to your suggestion.

Since we had substantial gaps of different sizes in the $\Sigma N_r$ concentration time series, an application of the MDV method to $v_d$ seems to be less useful.

**Comment R2.8:** *Line 17ff and line 826. Ra and Rb do not make a contribution to vd, but to Rt=1/vd. Alternative reword to say that Ra and Rb make a negligible contribution to limiting vd.*

**Response to R2.8:** We agree. Please note that the discussion related to Figure 9 was deleted. Thus, we removed corresponding sentences in the abstract and conclusion.

**Comment R2.9:** *It is well established that closed path sensors lead to a dampening of the fluctuations and thus the fluctuations induce the artificial flux due to quantum mechanical quenching are reduced compared with the true latent heat flux and as a result Eq. (1) will overestimate the correction by analogy to the impact on the density correction (e.g. Ibrom et al., 2007). Because the relative correction is small this is not a major issue, but the authors should acknowledge the uncertainty and clarify that the correction is an upper estimate.*

**Response to R2.9:** We agree. "Since we measured $H_2O$ fluxes with an open-path system and used them for correcting $\Sigma N_r$ fluxes, density corrections following the Webb-Pearman-Leuning correction for $H_2O$ fluxes measured with closed-path systems (Ibrom et al. 2007) were not accounted for. The impact on the correction is likely small, but the determined interference flux correction should be seen as an upper estimate." We added these details to line 229.

**Comment R2.10:** *The paper incorrectly states that the aerosol detected by the TRANC is NH4NO3 (line 46; line 144). In fact it detects the sum of NH4+ and NO3-, with the former also representing ammonium sulfates and the latter also sodium and calcium nitrate. Figure 4 very clearly demonstrates the presence of excess NH4+ over NO3- at this site.*

**Response to R2.10:** We replaced "particulate ammonium nitrate ($NH_4NO_3$)" by particulate ammonium ($NH_4^+$) and nitrate ($NO_3^-$) (line 46). Yes, different $NH_4^+$ and $NO_3^-$ aerosols are converted by the TRANC. We rephrased line 144 as follows: [..] "leading to a split up of $NH_4^+$ and $NO_3^-$ aerosols such as ammonium sulfate, ammonium nitrate, sodium and calcium nitrate into their subcomponents. In case of $NH_4NO_3$, it is thermally converted to $NH_3$ and $HNO_3$ (Marx et al., 2012)".

**Comment R2.11:** *The review of previous studies (lines 45 to 91) is incomplete and inconsistent. Firstly, it is worth mentioning that other micrometeorological methods do exist, beyond EC. In fact the references in line 69 refer partly to flux gradient measurements although the paragraph starts with "Prior EC studies of …". Secondly, there are not as few flux studies of Nr compounds to remote sites as stated. I could probably easily list 30, but many only cover short campaign periods. I therefore suggest starting the sentence in Line 67 with "Only a few long-term studies have been conducted to derive annual inputs at remote locations." and then focus on listing the long-term studies which can be done more exhaustively. This is also consistent with the true benefit of the TRANC system and this dataset as outlined above.*

**Response to R2.11:** We appreciate your suggestion to the introduction. Yes, there are other micrometeorological methods for estimating biosphere-atmosphere exchange of reactive nitrogen, for example the flux gradient method. Here, the focus is on eddy-covariance since it is the "common method for estimating greenhouse gas fluxes (Aubinet et al., 1999; Baldocchi, 2003) in flux monitoring networks (FLUXNET (Baldocchi et al., 2001), ICOS (Heiskanen et al., 2021)) and also suitable for reactive nitrogen compounds" as shown by the listed references. "However, the EC method requires fast-response analyzers." These sentences were added to line 48.

From line 45 to 59, we wanted to show that eddy covariance method has been applied to several $N_r$ compounds, but a simultaneous operation of individual devices using EC is challenging. We focused on listing EC studies in these lines since the coupling of the TRANC to a fast-response detector for NO allows the application of the EC method for total reactive nitrogen.

We appreciate your suggestions to the second paragraph since it strengthens the significance of TRANC measurements for the derivation of annual nitrogen dry deposition. We rephrased lines 67-91 as follows:

"Only a few long-term studies have been conducted to derive annual inputs with micrometeorological methods at (remote) forest ecosystems. Munger et al. (1996) conducted EC measurements of $NO_y$, which refers to the sum of all oxidized $N_r$ compounds, e.g., NO, $NO_2$, $HNO_3$, dinitrogen pentoxide ($N_2O_5$), peroxyacyl nitrates (PAN), aerosol nitrates, above a mixed deciduous forest for five years. Averaged $NO_x$ concentrations were at 0.62 and 4.26 ppb (0.36 and 2.44 µg N m$^{-3}$) during summer and winter, respectively, if wind was blowing from Northwest. During southwesterly winds, mean $NO_x$ concentrations were 1.25 and 9.48 ppb (0.72 and 5.43 µg N m$^{-3}$) during summer and winter, respectively, indicating a varying pollution climate. The authors reported an annual net dry deposition of $NO_y$ covering 1990 to 1994 of 2.49 kg N ha$^{-1}$ a$^{-1}$. Munger et al. (1998) reported an annual reactive N deposition of wet + dry deposition measurements of 6.4 kg N ha$^{-1}$ a$^{-1}$ for the period 1990 to 1996 at the same site. Dry deposition of $NO_y$ contributed 34% to total deposition. Wet deposition of $NH_4^+$ was comparatively low estimated to 1.1 kg N ha$^{-1}$ a$^{-1}$.

Neiryck et al. (2007) and Erisman et al. (1996) conducted GM measurements in order to estimate dry deposition of $NO_x$ and $NH_3$. Neiryck et al. (2007) published GM measurements from July 1999 to November 2001 above mixed coniferous/deciduous forest, which was in close proximity of a highway and the city of Antwerp leading to mean $NO_2$ and $NH_3$ concentrations of 8.7 and 3.0 µg N m$^{-3}$, respectively. The authors determined an annual $NH_3$ dry deposition of 19.6 kg N ha$^{-1}$ a$^{-1}$ and $NO_x$ emission of 2.7 kg N ha$^{-1}$ a$^{-1}$. $NO_x$ emissions were probably related to a strong contribution of soil-emitted NO. Erisman et al. (1996) reported $NO_x$ and $NH_3$ fluxes above a Douglas Fir stand of 2.5 ha surrounded by a larger forested area of 50 km$^2$ for 1995. Mean $NH_3$ concentration was 4.5 µg N m$^{-3}$

possibly related to livestock farming in the surroundings of the site. They estimated annual dry depositions of 17.9 kg N ha$^{-1}$ a$^{-1}$ and 2.8 kg N ha$^{-1}$ a$^{-1}$ for $NH_3$ and $NO_x$, respectively.

These were the few micrometeorological measurements of $N_r$ species above forests. No recent reports on long-term flux measurements of $N_r$ were found. Since several $N_r$ compounds contribute to $\Sigma N_r$ each with different chemical and physical properties, a complex arrangement of different, highly specialized measurement devices would be needed for quantifying $\Sigma N_r$ exchange. To our knowledge, there is no publication available reporting annual $\Sigma N_r$ deposition at (remote) forest ecosystems using micrometeorological methods. As stated above, the true benefit of the TRANC is that the most relevant $N_r$ species are converted, and a single instrument is sufficient for determining dry nitrogen deposition. Therewith, we were able to determine annual dry deposition and show seasonal changes in the $\Sigma N_r$ flux pattern."

**Comment R2.12** *Section 2.2: Please add horizontal and vertical displacement between TRANC inlet and anemometer, as well as the pressure downstream of the critical orifice and the turbulent Reynolds number in this low pressure region.*

**Response to R2.12:** The horizontal and vertical displacement heights were 32 and 20 cm, respectively (Wintjen et al., 2020). "The pressure gradient from the critical orifice to the CLD was not measured. Thus, only assumptions about the turbulent flow regime can be made. Considering tube length and lag time minus residence time in the converter, the latter assumed to 2 sec at maximum due to tube length and platin mesh as an additional flow resistance, flow speed was at 2.7 ms$^{-1}$ at maximum. Using an inner diameter of 4.4 mm and a kinematic viscosity at 15°C (1.485*10$^{-5}$ m$^2$/s), we calculated a Reynolds number of 800 indicating an overall laminar flow. We cannot provide a reasonable explanation to the low Reynolds number since pressure gradient was not measured.

Generally, the flow type inside the tube affects high-frequency attenuation (Massman, 1991; Lenshow and Raupach, 1991; Moncrieff et al., 1997). High-frequency attenuation was corrected with an empirical method based fully on measured cospectra (Wintjen et al., 2020). Since an empirical approach was used to estimate the high-frequency damping, effects originating from the low Reynolds number and from physical and chemical processes occurred after the critical orifice were considered in the flux analysis. "

We replaced line 149-152 by this response.

**Comment R2.13:** *Some more information on how the DELTA denuders were operated would be helpful without having to look up the quoted references. What filters were used and which coating for the denuders? The use of paper filters has been found to result in an aerosol underestimation of about 30%, which is not an issue for PTFE filters. K2CO3 coating results in a positive artefact on HNO3 from other NOy compounds, while NaCl coating is more selective. It is only in Section 4.1 that the paper seems to imply that K2CO3 coating was used. It is also worth stating that the cut-off of the DELTA denuder is approximately PM4.5 (see Tang et al., 2015; https://uk-air.defra.gov.uk/library/reports?report_id=861 ). The implications should be discussed also when comparing the TRANC and the sum of the Nr compounds (line 353). Mention also that the APNA-360 NO2 measurement was (presumably) made with a thermal converter and is therefore cross-sensitive to other oxidised nitrogen compounds.*

**Response to R2.13:** We thank the Reviewer for hinting on the publication of Tang et al. (2015). The positive artifact of carbonate coated denuders on $HNO_3$, the consequence of using paper filters, and the DELTA denuder cut-off size of 4.5 µm were added to line 187.

"Basic denuders were coated with sodium carbonate to collect $HNO_3$, $SO_2$, and HCl. Citric acid was applied to acid denuders for removing $NH_3$. Two cellulose filter papers (Whatman No. 1, 25mm diameter) were used for collecting aerosols. The first filter was prepared with potassium carbonate in glycerol, the second filter with citric acid."

For measuring NO and $NO_2$, the used APNA-360 was equipped with a thermal $NO_x$ converter resulting in cross-sensitivity to higher oxidized nitrogen compounds (line 194).

We considered these implications of the chosen coating, the aerosol cut-off size of DELTA, and the thermal NOx converter in the discussion (line 599 to 616, shifted to line 527). In comparison to the DELTA denuders cut-off size, we assume that TRANC cut-off size is higher. Due to the high temperatures in the TRANC (≥ 870°C), coarse $NO_3^-$ aerosols are probably decomposed. For example, sodium nitrate originating from sea salt is converted by the TRANC as shown by Marx et al. (2012). Thus, coarse fractions of nitrogen aerosols were converted in the TRANC implicating a higher cut-off size than the DELTA samplers. We added the discussion about the different cut-off sizes to line 527.

**Comment R2.14.** *Lines 189-192. I suggest you state already in this context that you were not able to calculate NH3 fluxes.*

**Response to R2.14:** We added the following sentence to line 191. "In contrast to Zöll et al. (2016), we were not able to calculate $NH_3$ fluxes with the QCL using the EC method (see Sec. 2.2)".

**Comment R2.15:** *Please add some details or reference with respect to the wet/bulk deposition measurements. Was a biocide used to avoid denitrification?*

**Response to R2.15:** Please see the response to R1.4.

**Comment R2.16:** *Line 305ff. Strictly speaking a "compensation point" is defined at the concentration at which (biological) consumption equals production. Thus, when talking about compensation points in a context other than "stomatal compensation point" it may be better to use the term "emission potential" or "equilibrium concentration", depending on context.*

**Response to R2.16:** We agree. Please note that lines 298 to 320 were removed as the discussion about $R_c$ was deleted.

**Comment R2.17:** *Line 315. It is worth noting that the evaporation of NH4NO3 during the deposition process also implies that some of the NH4 and NO3 measured as aerosol does not reach the surface as aerosol but as NH3 and HNO3 and can therefore deposition faster than particles.*

**Response to R2.17:** We agree. Please note that lines 315ff were deleted.

**Comment R2.18:** *Section 3.1. Figure 3 actually conveys the relative contribution of NH3 and NOx to total Nr more clearly than Figure 2. Maybe refer forward to Figure 3 when you discuss Fig. 2. In my mind Figure 3 does two things: (a) it shows the best estimate of the relative breakdown of Nr into the different species and (b) it acts as a quality control of the total Nr measurement. However, to interpret the figure in terms of (b), the reader would need to know which stacked bars are fully based on real data and which rely on gap-filling and also the % coverage of the Nr measurement for each data period. Could both pieces of information be added to the figure? With this additional information February 2018 could then be re-added to the figure: it reflects real data, but the gap filling does not work well on this data point. It would also indicate the years to which the sampling periods refer.*

**Response to R2.18:** We appreciate your suggestions to the discussion of figures 2 and 3. We added the following sentence after line 325. "A breakdown of $\Sigma N_r$ in compounds contributing most to its concentration pattern is shown in Fig. 3 illustrating a comparison of $\Sigma N_r$ concentrations with DELTA denuder and $NO_x$ measurements on monthly basis."

We added coverage of valid $\Sigma N_r$ concentration measurements in % for each exposure period. After carefully reviewing DELTA concentrations and replacing missing $NH_3$ values by passive samplers, we had data gaps for January 2017 (only $NH_3$), December 2017, March 2018, and April 2018. Remaining gaps averages were replaced by monthly averages estimated from other years. We rephrased the caption of Fig. 3 and indicate gap-filled bars as hatched. We added the year to the x-axis labels. Coverages of TRANC measurements during each exposure period were added if TRANC data was available. In the previous version of Fig. 3, TRANC concentrations were shown for June 2017. During the exposure period of the denuders, we had less than 1% coverage of TRANC measurements. We decided to remove that bar.

**Comment 2.19:** *Line 335. No systematic difference in NH3 between 20 and 30 m would indicate that NH3 showed no flux. Or is the uncertainty just too large to resolve gradients?*

**Response to 2.19:** For the entire campaign, "we found no systematic difference between $NH_3$ concentrations within the canopy and just above the canopy. Only for short time periods, for example summer 2016 and 2017, differences in passive samplers were found indicating a small $NH_3$ flux. Considering the LOD for IVL passive samplers for $NH_3$ of 0.4 µg N m$^{-3}$ determined by Dämmgen et al. (2010), shows that passive sampler measurements were conducted close to their LOD. It suggests that the uncertainty of the passive samplers was too large to resolve flux gradients." We added these details to the discussion (line 527).

**Comment R2.20:** *I have some comments regarding the assessment of the limit of detection and positive and negative fluxes (lines 371 to 390). The Finkelstein and Sims (2001) algorithm returns a different random error (and hence detection limit) for each 30-minute flux value. It is fine to state the average / median of this detection limit, but does it not make more sense to evaluate the fraction of data points for which the LOD is exceeded against individual LODs rather than the average LOD. For a near-zero*

*flux below the LOD one would expect about half of the flux values to be positive and half to be negative, but this does not really carry much information on the actual contribution of emission events as many of the positive fluxes would not be significantly different from zero. It would therefore be useful to add what fraction of the flux values above the LOD shows emission and deposition. The LOD is a function of instrument signal-to-noise, but also of turbulence and would be expected to be larger over forest. This needs to be taken into account when comparing LODs between studies (lines 538ff). As with other parts of the manuscript it is not very clear whether the median deposition figures (lines 381ff) refer to the filtered or the gap-filled data.*

**Response to R2.20:** According to your suggestion, we did the analysis of the LOD on half-hourly basis. "The comparison of half-hourly fluxes with their individual LOD revealed that 79% of the measured fluxes were above their detection limits. Deposition fluxes contributed with 84% to fluxes above the LOD. The fraction of emission was estimated to 16%. The relative contribution of emission fluxes to measured fluxes decreased under the consideration of the LOD. It shows that emission fluxes were closer to the flux detection limit of the instrument." We added these sentences to line 374ff.

We agree that micrometeorological factors like turbulence influence the flux detection limit, too. We rephrased the statements in line 538ff. As noted in comment R2.6, measured fluxes refer to filtered fluxes.

**Comment R2.21:** *Figure 9: Fluxes scale with gc = 1/Rc rather than with Rc and thus mean values of Rc should be calculated by averaging 1/Rc values and then turning back into Rc (or presenting as gc). The resulting pattern can look quite different. Was a filter applied for maximum that was allowed for Ra+Rb? At large values of Ra+Rb, Rc potentially becomes a small difference of two large numbers and thus quite uncertain.*

**Response to R2.21:** We applied filters for $R_a$ and $R_b$. Possibly, they were not strict enough as seen by the outliers in Fig. 9. As outlined in R2.1 to R2.4, we removed the discussion on $R_c$ since an accurate calculation of $R_b$ was not possible for $\Sigma N_r$.

**Comment R2.22:** *It would be worth discussing the annual N input (line 507) in relation to the Critical Loads for woodland.*

**Response to R2.22:** As written in lines 114 to 115, the discussion about critical loads will included in part II. However, we agree that critical loads for woodland should be mentioned. Published critical loads for *Picea abies* and *Fagus sylvatica* ranged from 10 to 15 kg N ha$^{-1}$ a$^{-1}$ and 10 to 20 kg N ha$^{-1}$ a$^{-1}$, respectively (Bobbink and Hettelingh, 2011). Since the forest stand consists to approximately 80% of Norway spruce in the footprint and the surrounding forest stand is predominated by Norway spruce, the critical load for the forest stand is probably closer to the limits of *Picea abies*. Estimated annual N input was 12.2 and 10.9 kg N ha$^{-1}$ a$^{-1}$ for the measurement years and were found to be at lower estimate of critical loads. It suggests that the forest is currently not in a critical state in relation atmospheric N input. We integrated details of this response to line 842.

**Comment R2.23:** *Line 549. Did February 2018 stand out in any other way? Was the wind direction unusual? Do the reports of the federal and state measurement networks report anything unusual?*

**Response to R2.23:** Please note the response to R1.14.

**Comment R2.24:** *Lines 562-565. I am not convinced there is a threshold NH3 concentration for ammonium sulfate formation. I thought any free NH3 would be pulled into the aerosol phase by the presence of sulphuric acid or bisulfate. Also, I am not sure this analysis works well with monthly data. The high concentrations could have been due to a short event during which no NH3 was present.*

**Comment R2.24:** As written above, please note the response to R1.14. We agree that a threshold analysis did not work well with monthly data. Therefore, we decided to delete these lines.

**Comment R2.25:** *Lines 566 to 598. It is fine to point out the difference in flux during the winter periods 2016 and 2017, and their relationship to snow cover. However, it would be prudent to show in Fig. 12a the fluxes during periods with snowfall as dotted lines as they are highly uncertain (it could be argued they should be removed completely) and I find the section overly lengthy and speculative. I am highly sceptical that NH3 would be able to diffuse through a 60 cm snow layer without being re-captured. Is there literature evidence that this might be possible? Also, please select colours to be readable by people with red/green blindness (Figure 12).*

**Response to R2.25:** In Fig. 12, we showed fluxes smoothed with a 3-h running mean. In case of averaging the uncertainty of the fluxes is significantly reduced. Still, uncertainties are not reduced completely. Due to visibility reasons, we decided to show fluxes and concentrations as solid lines and changed the colors according to your suggestion.

We decided to remove the discussion about $NH_3$ since the diffusion of $NH_3$ through a snow layer of a large depth (60 cm) seems improbable. We found no literature evidence for diffusion of $NH_3$ through a snow/ice layer. NO could be responsible for the observed $\Sigma N_r$ emission fluxes but different observations were made about correlations of NO with snow cover, micrometeorological parameters, and about sources of NO emissions as stated in the manuscript. Since we had no measurements of NO close to the forest floor or measurements of the mass loss rate of litter under snow cover, we can only made assumptions about the origin of the $\Sigma N_r$ emission fluxes. As mentioned in lines 596 zo 598, NO emitted from the forest soil is rapidly converted to $NO_2$ (Rummel et al., 2002). Thus, the measured $\Sigma N_r$ flux probably had a high $NO_2$ contribution during that time. In the revised version of the manuscript, we shortened the discussion of Fig. 12.

**Comment R2.26:** *Line 604f. I don't understand this sentence. Are you trying to say that the DELTA measurements suggested that gaseous compounds made a significant contribution to the Nr concentration?*

**Response to R2.26:** Yes. We rephrased the sentence.

**Comment R2.27***: Line 605. Which slight increase in HNO3 and decrease in NH4+?*

**Response to R2.27:** The increase in the relative contributions of $HNO_3$ from spring to summer compared to the decrease of $NH_4^+$ and $NO_3^-$ (Fig. 4) can be related to the evaporation of $NH_4NO_3$. We rephrased the sentence according to this response.

**Comment R2.28:** *Line 610ff. The NOx analyser was likely a thermal analyser and cross-sensitive to*

*other NOy compounds? Worth mentioning here also the likely difference in cut-off diameters between DELTA and TRUNC for aerosol.*

**Response to R2.28:** As written in R2.13, the $NO_x$ analyzer was a thermal analyzer and likely cross-sensitive to other $NO_y$ compounds. "However, measured concentrations of $HNO_3$ or $NO_3^-$ were comparatively low as seen in Fig. 3. Thus, their influence on $NO_x$ measurements appeared to be negligible." As written in R2.13, we integrated the possible difference in cut-off sizes of TRANC and DELTA in the discussion (line 527ff).

**Comment R2.29.** *Line 614f. What is your evidence that the DELTA suffered break-through at high concentration peaks? Or are you just speculating that this might be a possibility. Maybe the use of the word "could" is not quite right? Also, a key uncertainty originates from the TRUNC measurement likely not covering 100% of the DELTA sampling time. See comment 18 above.*

**Response to R2.29:** Reasons for differences between TRANC and DELTA are the cut-off diameters, issues in the conversion efficiency, and the data coverage of the TRANC, which was 78% on average for the exposure periods. In addition, $N_2O_5$ or peroxy acetyl nitrates are not collected by denuders, but probably converted by the TRANC. We rephrased line 614f accordingly.

**Comment R2.30:** *Lines 617-624. Deposition velocities of NH3 are highly variable and would be expected to decrease for semi-natural forests that are subject to high Nr input (because the stomatal compensation points would go up; see Massad et al., 2010) and with decreasing ambient concentration (away from sources). The importance of the NH4NO3 evaporation effect that likely affected the summer measurements of Wolff et al. (2010) would likely be much smaller during cooler periods resulting in smaller deposition rates at other times of the year. So I am not sure the conclusions hold.*

**Response to R2.30:** After carefully reflecting your comment, we found a comparison of a $v_d$ for $\Sigma N_r$ with individually measured $v_d$ as less useful since we have no information about their relative contribution to the $v_d$ of $\Sigma Nr$. Thus, we decided to remove the lines 617 to 624.

**Comment R2.31:** *Line 639. A stomatal compensation point has only been shown to exist for NH3 and has in some studies been indicated for NO2. There is no such thing as a stomatal compensation point for total Nr. And there is also no canopy compensation point for Nr (Line 685). The concept of a canopy compensation point has not been introduced in the paper anyway.*

**Response to R2.31:** We agree that the sentence is incorrect (line 639) and the canopy compensation point was not introduced. Please note the lines were removed and the substantial changes to Sec 4.2.

**Comment R2.32**: *Line 645. HNO3 is formed by reaction of NOx with OH not O3.*

**Response to R2.32:** Corrected.

**Comment R2.33:** *Lines 664 to 669. This paragraph seems to mix up the effects of concentration on the flux and vd. Clearly, for a depositing compound, the flux increases with concentration. For vd this may or may not be the case.*

**Response to R2.33:** We agree that in these lines the effect of concentration on flux and $v_d$ are mixed up. The analysis of Fig 5 has shown that $v_d$ was not dependent on concentration. We corrected the statements accordingly.

**Comment R2.34:** *Lines 729 to 738. I am not sure the water holding capacity of leaves at intermediate relative humidity is governed only by NO3-/NH4- in air. Any hygropscopic aerosol from dry, wet and fog deposition could contribute to this. As the authors show their measurement site is by no means pristine. See also Sutton et al. (1998) and Flechard et al. (1999).*

**Response to R.34:** These lines belong to the Fig 9 and $v_d$ measured during precipitation. The discussion about Fig. 9 was removed and fluxes determined during rain were filtered out from analysis. Thus, we deleted lines 729 to 738.

**Comment R2.35:** Line 801. What do you mean by 'canopy outflow'? Do you mean "catchment outflow" or "throughfall"? How can those measurements distinguish between dry and wet deposition?

**Response to R2.35:** We meant here the canopy budget technique (Beudert et al. 2014). We corrected the sentence.

**Comment R2.36:** *Line 803. Which two sampler types? Positive artefacts on bulk deposition gauges (if this is what you are referring to here) can also originate from dry deposition of gas phase NH3 and HNO3.*

**Response to R2.36:** We referred here wet deposition from wet-only and bulk samplers. As shown in Table 1, differences between bulk and wet-only deposition were negligible. Small differences could be induced by organic and inorganic dusts or related to dry deposition of $NH_3$ and $HNO_3$ as you mentioned. The effects were not relevant for the annual nitrogen deposition at the measurement site. We rephrased line 803 according to the Reviewers' comments.

**Comment R2.37:** *Line 834. I can see that inferential modelling would extrapolate fluxes mechanistically to low turbulence conditions, however I fail to see how this is possible with neutral flux networks if they are trained with u\* filtered data.*

**Response to R2.37:** If neural networks are trained with $u_*$ filtered data and subsequently used for gap-filling, biases to the $u_*$ threshold are introduced. We decided to remove neural networks from the sentence.

**Technical comments**

**Comment R2.38:** *The papers uses different units in different places. When discussing the Nr components these are in µg-N m-3, which is logical, but when referring to previous measurements (e.g. line 121, 512, 521, …) they change to ppb. I suggest adding values in µg-N m-3 in brackets here so the reader can compare more easily.*

**Response to R2.38:** We thank the Reviewer for his suggestion. We changed the unit to ppb because the cited references provided their results in ppb. We adapted your suggestion and added values in μg N m$^{-3}$ in brackets.

**Comment R2.39:** *Line 10. "was observed for the contribution of NH3 …"*

**Response to R2.39:** Changed.

**Comment R2.40:** *Line 15. "changes in composition of ΣNr and radiation"*

**Response to R2.40:** Changed.

**Comment R2.41:** *Line 23. "During these periods, cuticular or soil …"*

**Response to R2.41:** Sentence was deleted.

**Comment R2.42:** *Line 38. Correct subscript on PM2.5.*

**Response to R2.42:** Corrected.

**Comment R2.43:** *Line 45. The community tends to use "nitric oxide" over "nitrogen monoxide" for NO.*

**Response to R2.43:** We changed the order.

**Comment R2.44:** *Line 74. "… radiation as the primary driver for …"*

**Response to R2.44:** Please note that lines 66-91 were removed.

**Comment R2.45:** *Line 76. "… as a secondary driver …"*

**Response to R2.45:** see above.

**Comment R2.46:** *Line 134. "the dominating Norway spruce is recovering"*

**Response to R2.46:** Corrected.

**Comment R2.47:** *Line 149. The sentence starting "The mass flow rate …" seems redundant.*

**Response to R2.47:** We agree. The sentence was removed.

**Comment R2.48:** *Line 208. The sentence "Figures with the notation …" seems over the top. I suggest you just write "(see Fig. S1 of the Supplementary Material)"*

**Response to R2.48:** We deleted the sentence and adapted your suggestion.

**Comment R2.49:** *Line 266. Missing parenthesis "2010))."*

**Response to R2.49:** Corrected.

**Comment R2.50:** *Figure 2. I think the figures would be more readable if they all used the same y-scale, possibly capped at 15 ug N m-3.*

**Response to R2.50:** We adjusted the y-limit of all subplots to 15 µg N $m^{-3}$.

**Comment R2.51:** *Figures 2 and 5. The whiskers do not look like they scale with the IQR. Please state sampling intervals for both figures as the statistics depend on it.*

**Response to R2.51:** In Fig. 2, the boxplots cover the entire measurement campaign. As noted in Fig. 5, boxplots refer to monthly intervals. The whiskers extend to the outermost points within the Q1 - 1.5*IQR and Q3 + 1.5*IQR range. Thus, whiskers have different lengths.

**Comment R2.52:** *Line 329. "reached values of up to"*

**Response to R2.52:** Corrected.

**Comment R2.53:** *Figure 5. Rather than the last sentence in the caption, the authors could use arrows with values to indicate the magnitude of the three points that fall outside the y-range.*

**Response to R2.53:** "The whiskers in February 2018 cover the range from -191 to 105 ng N $m^{-2}$ $s^{-1}$, the upper whisker of December 2017 was at 69 ng N $m^{-2}$ $s^{-1}$." We rephrased the sentence according to this response.

**Comment R2.54:** *Line 405. "shows the median vd for the corresponding fluxes."*

**Response to R2.54:** Corrected.

**Comment R2.55:** *Line 416. Should this more accurately read "the deposition of total Nr."?*

**Response to R2.55:** Corrected.

**Comment R2.56:** *Figures 8 and 9. I would find these easier to grasp if the plots of the first row were labelled (a), (b), (c) etc.*

**Response to R2.56:** We changed the labelling order.

**Comment R2.57:** *Line 443. "dry conditions (no precipitation) are associated with enhanced deposition"*

**Response to R2.57:** Corrected.

**Comment R2.58:** *Line 507. "In total, we derived a total"*

**Response to R2.58:** Corrected.

**Comment R2.59:** *Line 521. The word "who" cannot refer to authors in brackets.*

**Response to R2.59:** We removed "who" and started a new sentence beginning with "The authors …"

**Comment R2.60:** *Line 524. Change "expectable" to "to be expected" – it is highly unusual and not in all dictionaries.*

**Response to R2.60:** Done.

**Comment R2.61:** *Line 535. Change "phase" to "period".*

**Response to R2.61:** Done.

**Comment R2.62:** *Line 542. "that the flux magnitude"*

**Response to R2.62:** Corrected.

**Comment R2.63:** *Line 559. Correct "SO42-"*

**Response to R2.63:** Please note that the sentence was deleted.

**Comment R2.64:** *Line 560. "is the dominant aerosol form."*

**Response to R2.64:** Please note that the sentence was deleted.

**Comment R2.65:** *Line 599. "that NOx made the largest contribution"*

**Response to R2.65:** Changed.

**Comment R2.66:** *Line 603. "NH3 had a strong presence"*

**Response to R2.66:** Corrected.

**Comment R2.67:** *Line 653. "may also be related"*

**Response to R2.67:** Please note that the sentence was deleted.

**Comment R2.68:** *Line 657. "as a primary controlling"*

**Response to R2.68:** Please note that the sentence was deleted.

**Comment R2.69:** *Line 705. "During periods of"*

**Response to R2.69:** Please note that the sentence was deleted.

**Comment R2.70:** *Line 723. "When the canopy gets drier"*

**Response to R2.70:** Please note that the sentence was deleted.

**Comment R2.71:** Line 726. "than stomatal deposition"

**Response to R2.71:** Please note that the sentence was deleted.

**Comment R2.72:** Line 788. *This acronyms were introduced in line 95 and not used since! Probably worth spelling out in full here.*

**Response to R2.72:** Agreed. We wrote their full form here.

**Comment R2.73**: *Line 834. "to the use of friction"*

**Response to R2.73**: Please note that the sentence was deleted.

**References**

[revised manuscript text omitted]

---

## Author Response (AR3)

**Response to Editor's comments – manuscript BG-2020-364 Forest-atmosphere exchange of reactive nitrogen in a low polluted area – Part I: Measuring temporal dynamics**

We sincerely thank the editor for her comments to the revised version. We rephrased the corresponding lines according to the provided suggestions and clarified the remaining minor points.

Comments range from R1.1 to R1.111. Line numbers in the answers, where new information was added to the manuscript, refer to the last revised version. The text which is enclosed by "..." and highlighted in red is implemented in the manuscript.

**Response to Editor's comments:**

**Comments to the manuscript:**

**Comment R1.1:** *Title: Using "remote region (area)" instead of "low polluted area" would sound much better to my opinion.*
**Response to R1.1:** We replaced *"low polluted area"* by "remote region".

**Comment R1.2:** *Line 5: ...at a mixed forest exposed to air pollution levels.*
**Response to R1.2:** We removed the adjective "low-polluted" and added "exposed to low air pollution levels" to end of the sentence.

**Comment R1.3:** *Line 15: ...high solar radiation....*
**Response to R1.3:** We replaced "incident" by "solar".

**Comment R1.4:** *Line 19: No significant influence of temperature, humidity, friction velocity, or wind speed on ΣNr dry deposition sums were found.*
➔ *This is somewhat in contrast to what was mentioned in other places of the paper. These variables determine the deposition velocity, and, hence, at the end also total deposition. Maybe reformulate or delete.*
**Response to R1.4:** We rephrased the sentence to "No significant influence of temperature, humidity, friction velocity, or wind speed on $\Sigma N_r$ fluxes when using the Mean-Diurnal-Variation (MDV) approach for filling gaps of up to five days was found."

**Comment R1.5:** *Line 21: ...half-hourly value...*
**Response to R1.5:** Added "y".

**Comment R1.6:** *Line 25: ...to a remote forest ecosystem.*
**Response to R1.6:** Replaced "of" by "to" and added "remote"

**Comment R1.7:** Line 37: nitric oxide (NO)
**Response to R1.7:** Corrected.

**Comment R1.8:** *Line 39...aerodynamic gradient method (AGM), please change throughout.*
**Response to R1.8:** We changed it in lines 39, 71, and 72.

**Comment R1.9:** *Lines 79-80: These few long-term micrometeorological measurements of Nr species above forests were made more than 20 years ago and no recent reports on long-term flux measurements of Nr are currently available.*
**Response to R1.9:** We rephrased the lines according to your suggestions.

**Comment R1.10:** *Lines 83-84: As stated above, the outstanding benefit of TRUNC is ….*
**Response to R1.10:** We replaced "true" by "outstanding".

**Comment R1.11:** *Line 85-86: I would combine this sentence with the scientific objectives mentioned in lines 100-104: please reformulate / list points (1), (2), (3) as scientific objectives.*
**Response to R1.11:** We deleted the sentence in line 85 and rephrased lines 100-104 as follows:
"Our study is the first one presenting long-term eddy-covariance flux measurements of $\Sigma N_r$ above a remote forest. Based on the successful implementation of the TRANC methodology, our objectives are:
  1. A discussion of observed concentration and flux patterns of $\Sigma N_r$ in the context of different temporal scales
  2. An investigation of the influence of micrometeorology on deposition velocities
  3. An assessment of annual N deposition using both gap-filling for the dry deposition eddy flux data and complementary wet deposition estimates from local samplers."

**Comment R1.12:** *Lines 105-108: Please delete these lines a they are not required here and can be misleading. I would just mention one sentence that a follow up paper will deal with…*
**Response to R1.12:** We agreed and deleted the lines 105 to 110 since the content of the follow-up paper may be modified during its review process and added the following sentence to the text: "A follow-up paper will investigate the usage of the acquired dataset in a modeling framework to estimate annual N budgets."

**Comment R1.13:** *Line 196: Additionally, fast-response measurements…. (delete…, too).*
**Response to R.1.13:** We added the word "Additionally" at the beginning of the sentence and deleted the word "too".

**Comment R1.14:** *Line 251: … and associated descriptions are based on…*
**Response to R1.14:** Corrected.

**Comment R1.15:** *Line 293: … replace "or nitrogen aerosols" with "or related aerosol compounds"…*
**Response to R1.15:** Done.

**Comment R1.16:** *Lines 305-306: Please delete: "Further details about the implementation of these resistances in surface-atmosphere models can be found in van Zanten et al. (2010)."*
**Response to R1.16:** The sentence was deleted.

**Comment R1.17:** *Line 312: replace "A breakdown…" by "The contribution of individual nitrogen compounds to the total $\Sigma Nr$ concentration pattern is shown in Fig. 2, which…..*
**Response to R1.17:** We rephrased the sentence according to your suggestion.

**Comment R1.18:** *Line 315: NOx also showed a… (delete "too")*
**Response to R1.18:** Added "also" and deleted "too".

**Comment R1.19:** *Line 318: The $\Sigma Nr$ concentration was 3.1….*
**Response to R1.19:** Added "The".

**Comment R1.20:** *Line 321: …in the annual pattern was reasonable…*
**Response to R1.20:** Replaced "good" by "reasonable".

**Comment R1.21:** *Line 322: …with measurement height was observed.*
**Response to R1.21:** Replaced "could be" by "was".

**Comment R1.22:** *Line 323-324: At 50 m the $NH_3$ concentration exceeded that at 30 m by 0.1 µg N $m^{-3}$.*
**Response to R1.22:** We rephrased the sentence.

**Comment R1.23:** *Line 339: I propose to use the expression diurnal cycles instead of daily cycles throughout the MS.*
**Response to R1.23:** We changed the expression throughout the manuscript (lines 373, caption of Fig. 5, 374, 377, 378, 380, 383, 384, 423, caption of Fig. 7, and 693).

**Comment R1.24:** *Line 325: The seasonal variations of the half-hourly ΣNr concentrations are represented by box-and-whisker plots including monthly medians in Fig. S3. (delete: Figure S3 shows monthly box plots of the concentrations.)*
**Response to R1.24:** We rephrased line 325 and deleted the subsequent sentence.

**Comment R1.25:** *Line 327: Medians ranged between…*
**Response to R1.25:** Replaced "were" by "ranged".

**Comment R1.26:** *Figure 2 caption: …Missing NH3 values from the DELTA measurements…. Numbers above the bars indicate the relative coverage of TRANC measurements during each exposure period.*
**Response to R1.26:** We replaced "measurements" by "values" and added the word "indicate".

**Comment R1.27:** *$NH_3$ also featured seasonal variations with….*
**Response to R1.27:** Replaced "showed" by "featured" and "changes" by "variations".

**Comment R1.28:** *Line 353: As shown in Fig. 2, ΣNr…*
**Response to R1.28:** Replaced "seen by" by "shown in".

**Comment R1.29:** *I would split the first results section in two parts:*

*3.1 Measured concentrations of individual reactive nitrogen compounds*
*Including Figures 1-3 and S1-S4*

*3.2 Measured exchange fluxes of total reactive nitrogen*
*Starting on page 14 (break at line 355)*

**Response to R1.29:** We changed the header of 3.1 to "Measured concentrations of $ΣN_r$ and individual $N_r$ compounds" and titled section 3.2 beginning at line 355 "Measured exchange fluxes and deposition velocities of $ΣN_r$".

**Comment R1.30:** *Line 355-356: …on a monthly timescale….*
**Response to R1.30:** Added "a".

**Comment R1.31:** *Line 359: on a half-hourly basis…., On a monthly basis…*
**Response to R1.31:** Added "a".

**Comment R1.31:** *Line 360: According to Langford et al. (2015), the limit of detection (LOD) is calculated by multiplying the random flux error (95% confidence limit) with 1.96.*
**Response to R1.31:** Corrected the position of "1.96" within the sentence.

**Comment R1.31:** *Line 364: This indicates that emission fluxes….*

**Response to R1.31:** Replaced "it shows" by "this indicates".

**Comment R1.32:** *Line 365: In general, median deposition was within the same range for the entire campaign with only small seasonal differences.*
**Response to R1.32:** Replaced "on the same level" by "within the same range" and added "only".

**Comment R1.33:** *Line 367-368: Median deposition was significantly increased from June 2016 till September 2016 than for the same period in 2017 and IQR and whisker also covered a wider range in 2016.*
**Response to R1.33:** Replaced "stronger" by "increased", the dot by "and", "too" by "in 2016", and added "also".

**Comment R1.34:** *Line 374: Fig. 5 shows averaged daily cycles of measured ΣNr fluxes for every month.*
**Response to R1.34:** Added "of measured $\Sigma N_r$ fluxes".

**Comment R1.35:** *Figure 5: Mean diurnal cycle of ΣNr fluxes (ng N m$^{-2}$ s$^{-1}$) based on half-hourly measurements for every month from June 2016 to June 2018. The shaded….*
**Response to R1.35:** We rephrased the caption according to your suggestion.

**Comment R1.36:** *Line 374-375: In general, the ΣNr diurnal cycle exhibited low deposition or fluxes close to zero during nighttime/evening and increasing deposition during daytime. Deposition fluxes were…*
**Response to R1.36:** Replaced "neutral exchange" by "fluxes close to zero" and "rates" by "fluxes".

**Comment R1.37:** *Line 378: …with near-zero or small negative fluxes…*
**Response to R1.37:** Replaced "neutral" by "near-zero".

**Comment R1.38:** *Line 379: … months were comparable.*
**Response to R1.38:** Replaced "uniform" by "comparable"

**Comment R1.39:** *Line 381-382: …was close to zero one year later.*
**Response to R1.39:** Replace "neutral a" by "zero one"

**Comment R1.40:** *Line 386: Again, the average standard error…*
**Response to R1.40:** Added "the".

**Comment R1.41:** *Line 390: The meaning of "From May to September, the curve was approximately bell-shaped." is unclear. Please clarify.*
**Response to R1.41:** Rephrased to: "From May to September, a continuous increase in $v_d$ was observed from 6:00 a.m. until noon. A decrease in $v_d$ followed in the late afternoon (15:00 to 18:00 LT)."

**Comment R1.42:** *3.3 Controlling factors…*
**Response to R1.42:** We changed the numbering of the section title.

**Comment R1.43:** *Line 396 "leading to a constantly low vd during the day (Fig. S10)." From Fig. S10 it is evident that vd even strongly decreases during midday, this should be mentioned (and explained in the discussion).*
**Response to R1.43:** We agree that it should be mentioned. We added the following sentence to line 396: "During that time, a strong decrease in $v_d$ was found with near-zero or even small negative values around 12:00 LT." The following lines were added to the discussion (line 614). "Stomatal uptake of $N_r$ compounds was possible during periods of photosynthetic activity, leading to high values of $v_d$ during the summer month (Fig. S9). Fig. S10 reveals that a certain degree of $\Sigma N_r$ uptake still occurred in winter, but deposition decreased strongly during midday, and even periods of emission were observed. These

emissions may be due to the decomposition of leaves, leading to a release $NH_3$ in late autumn/early winter (Hansen et al., 2013), or from snow-covered soils (see Sec. 4.1)."

**Comment R1.44:** *Line 399: and the concentration of ΣNr, especially changes in the concentration of the individual nitrogen compounds….*
**Response to R1.44:** Added "the" and replaced "sub components" by "individual nitrogen compounds".

**Comment R1.45:** *Line 410: For visualizing the impact of the concentration on $v_d$ (Fig. 6),…*
**Response to R1.45:** Added "the".

**Comment R1.46:** *Line 412: …increments of the ΣNr concentration…*
**Response to R1.46:** Added "the".

**Comment R1.47:** *Line 413: …on the ΣNr concentration…*
**Response to R1.47:** Added "the".

**Comment R1.48:** *Line 414: It demonstrates that the ΣNr concentration…*
**Response to R1.48:** Added "the".

**Comment R1.49:** *Line 419: …* $v_d$ was more influenced by micrometeorological variables than by the ΣNr concentration.
**Response to R1.49:** Replaced "had a higher affinity" and "parameters than to" by "was more influenced by" and "variables than by", respectively.

**Comment R1.50:** *Line 425-426: Combine 2 sentences, they should read as: "During winter (December, January, and February), vd was almost equal and even lower during the day, which resulted in a lower deposition of ΣNr."*
**Response to R1.50:** Done.

**Comment R1.51:** *Line 426-428: The sentence should read as:*

*The different shapes of the diurnal variations of vd could be induced by micrometerological variables, which change the composition of available ΣNr compounds during the day*  *and promote photosynthesis (e.g. stomatal uptake or release of $NO_2$ and $NH_3$).*

*Seinfeld and Pandis, 2006 is for sure not the appropriate literature here, please choose other more specific references (as in the discussion section).*

**Response to R1.51:** We rephrased the sentence according to your suggestion and we cited Munger et al. (1996), Horii et al. (2004,2006), Wyers and Duyzer, (1997), and van Oss et al. (1998) instead of Seinfeld and Pandis" (2006) in this sentence. We further added Thoene et al. (1996) and Wyers and Erisman (1998) to line 428.

**Comment R1.52:** *Figure 7. Mean diurnal cycle of $v_d$ from May to September for low and high temperature (a), relative humidity (b), and concentration (c). Median….*
**Response to R1.52:** We replaced "daily" by "diurnal" and changed the sentence position of $v_d$.

**Comment R1.53:** *Line 435: … lower relative humidity….*
**Response to R1.53:** Replaced "less" by "lower".

**Comment R1.54:** *Line 437: During dawn/nighttime, deposition velocities exhibited no significant difference between the applied thresholds…. I can see a difference for the dry/wet leaf surface. Please double check this statement.*

**Response to R1.54:** "In the presence of dry leaf surfaces, $v_d$ was higher by approximately 0.2 cm s$^{-1}$ compared to wet leaf surfaces during the night". The sentence was added to line 437.

**Comment R1.55**: *Line 439: …compared to the May to September period.*
**Response to R1.55:** Replaced "time frame" by "period".

**Comment R1.56**: *Section 3.3 should be changed to:*

*3.4 Dependence of ΣNr dry deposition sums on micrometeorological variables*
**Response to R1.56:** We changed the section title.

**Comment R1.57**: *Figure 8. … represented by box-and-whisker plots…*
**Response to R1.57:** Replaced "depicted as" by "represented by" and added "-and-whisker".

**Comment R1.58**: *Eq. (3) (Pastorello et al., 2020) please refer to the discussion section here.*
**Response to R1.58:** Replaced the reference by section reference.

**Comment R1.59**: *Line 450: median depositions of the ΣNr fluxes with….*
**Response to R1.59:** Corrected.

**Comment R1.60:** *Line 451: median depositions … -> please correct all instances*
**Response to R1.60:** Done.

**Comment R1.61:** *Figure 9. Annual ΣNr dry deposition shown as bar graphs*
**Response to R1.61:** Done.

**Comment R1.62:** *Line 459: … dry depositions sums…*
**Response to R1.62:** Done.

**Comment R1.63:** *Introduce new section after Line 466:*
*3.5 Wet and total nitrogen deposition*
**Response to R1.63:** Done.

**Comment R1.64:** *Line 473: In the second year, the contribution of dry deposition…*
**Response to R1.64:** Added "the".

**Comment R1.65:** *Line 476-477: Which was probably related to high NH$_3$ concentrations… For sure it was, you measured them, please refer to the corresponding Figure here.*
**Response to R1.65:** Added references to Figs. 2 and S2.

**Comment R1.66:** *Line 509: Thus, their influence on NOx measurements was most likely small.*
**Response to R1.66:** Replaced "appeared to be" by "was most likely".

**Comment R1.67:** *Line 515-516: DELTA measurements further suggested that the ΣNr concentration pattern was mainly influenced by gaseous Nr.*
**Response to R1.67:** Changed the order and wording according to your suggestion.

**Comment R1.68:** *Line 521-522: Due to the reaction of NH$_3$ with HNO$_3$ and sulphuric acid particulate NH$_4^+$ is formed, available as NH$_4$NO$_3$ or (NH$_4$)$_2$SO$_4$.*

➔ I would change the order of compounds here:

*Explanation:*

*In chemical systems composed of $NH_3$, $HNO_3$ and $H_2SO_4$, the formation of non-volatile $(NH_4)_2SO_4$ is preferred. Only when $NH_3$ is available in excess of $H_2SO_4$ and when favourable meteorological conditions (low to moderate T and/or high RH) prevail, neutralization of $HNO_3$ vapor with $NH_3$ occurs (Trebs et al., 2005).*

*Trebs, I., Metzger, S., Meixner, F.X. et al., 2005. The NH4+-NO3--Cl--SO42--H2O aerosol system and its gas phase precursors at a pasture site in the Amazon Basin: How relevant are mineral cations and soluble organic acids? Journal of Geophysical Research-Atmospheres, 110(D07303): doi:10.1029/2004JD005478.*

**Response to R1.68:** We thank the editor for her literature recommendation. We decided to change the order of compounds.

**Comment R1.69:** *Line 523: … fine mode and associated with aerodynamic diameters….*
**Response to R1.69:** Replaced "assigned" by "associated" and added "aerodynamic".

**Comment R1.70:** *Line 537: …, but were probably ….*
**Response to R1.70:** Added "were".

**Comment R1.71:** *Line 553-554: …for instance by bidirectional exchange of $NH_3$ leading to both periods of net emission and deposition of ΣNr.*
**Response to R1.71:** Rephrased the sentence according to your suggestion.

**Comment R1.72:** *Line 567: Also, the $SO_2$ concentration was much larger…*
**Response to R1.72:** Corrected.

**Comment R1.73:** *Line 577: …resulting in a high vd, which is due to efficient turbulent mixing. Hence, even at low concentrations…*
**Response to R1.73:** Replaced "allowed by turbulence" by "due to efficient turbulent mixing" and added "even".

**Comment R1.74:** *Line 578-579: In conclusion, particulate $NH_4^+$ was mainly responsible for the large ΣNr deposition due to its excess over aerosol $NO_3^-$.*
**Response to R1.74:** Rephrased the sentence according to your suggestion.

**Comment R1.75:** *I propose that the section on ΣNr emission and the influence of snow can be shortened. The English writing of this section must be improved (Lines 590-609).*
**Response to R1.75:** Agreed. We shortened the discussion on the decomposition and snow cover, e.g. removed descriptions to referred publications.

**Comment R1.76:** *Line 616: I think it is anyway highly unlikely that the concentration drives the deposition velocity.*
**Response to R1.76:** We deleted the sentence in line 616 since it is written in the previous sentence that $v_d$ was independent of the $ΣN_r$ concentration.

**Comment R1.77:** *However, the impact of increasing concentrations on….*
**Response to R1.77:** Replaced "still" by "however" and added "s" to concentration

**Comment R1.78:** *Line 624: … was nearly zero and emission…*
**Response to R1.78:** Replaced "almost neutral" by "nearly zero".

**Comment R1.79:** *Line 631: … contribution of individual compounds do show a seasonal cycle. Since the ΣNr compounds differ in their vd,…*

**Response to R1.79:** Corrected.

**Comment R1.80:** *Line 635: …than of $NO_2$, but… than of $NO_2$ for woodland.*
**Response to R1.80:** Added "of".

**Comment R1.81:** *…and 2.2 cm s$^{-1}$  (see Schrader…*
**Response to R1.81:** Corrected.

**Comment R1.82:** *Lines 637-643: Rewrite to:*

*However, variations in the composition of ΣNr may correlate with micrometerological parameters. For example, the formation of $HNO_3$ is correlated with Rg. The solar radiation responsible for the stomatal opening also promotes the formation hydroxyl radicals, which react with $NO_2$ to form $HNO_3$ (Seinfeld and Pandis, 2006). Tair influences the diurnal pattern of NH4NO3, which may also volatilize close to the surface due to the depletion of its precursors and in case the temperature gradient is large enough (Wyers and Duyzer, 1997; Van Oss et al., 1998). Thus, part of the $NH_4^+$ and $NO_3^-$ in the aerosol phase may be converted to $NH_3$ and $HNO_3$, which deposits faster to surfaces than aerosols.*

**Response to R1.82:** We implemented your suggestions to these lines and cited additional literature.

**Comment R1.83:** *Line 646: In conclusion, the variability…*
**Response to R1.83:** Added "the".

**Comment R1.84:** *Line 648-649: Delete: *
**Response to R1.84:** Deleted.

**Comment R1.85:** *Line 656: …measured half-hourly values…*
**Response to R1.85:** Added "ly" and "values".

**Comment R1.86:** *Line 657: … low-quality half-hourly values were effectively…*
**Response to R1.86:** Added "ly" and "values" and replaced "could be" by "were".

**Comment R1.87:** *Line 660: Was there any footprint analysis performed or required due to fetch limitations? Could you comment on that? Maybe refer to previous publications.*
**Response to R1.87:** We conducted a footprint analysis using the footprint estimation tool of Kljun et al. (2015) implemented in the software TOVI (LICOR Biosciences, 2020). Fig. 1 shows the footprint of the measurement site of stable and unstable conditions with isolines representing a given percentage of the flux contribution exemplarily for the year 2016.

[Figure]

**Figure 1:** 2D-footprint of the measurement site of stable and unstable conditions exemplarily shown for the year 2016. Isolines represent a given percentage (10% to 80%) of the flux contribution. The heat map illustrates from which direction most of the fluxes originated.

The 2D-footprint analysis showed that the 70% isoline of the flux had an extension of approximately 300 m. In southwest direction of the tower (approx. distance 100 to 300 m), tree density and height were lower than to the Northeast of the tower. Due to the high surface roughness, the flux footprint is limited in its size but the footprint represents the typical forest structure of the Bavarian Forest National Park. Thus, we did not filter half-hourly fluxes from certain wind direction sectors.

We added the text given in this response to line 660.

**Comment R1.88:** *Line 669: of turbulent motions…*
**Response to R1.88:** Added "s".

**Comment R1.89:** *Line 675: As shown in Fig. 8…*
**Response to R1.89:** Replaced "seen" by "shown" and replaced "within the error range of the dry deposition sum" by "small compared to estimated dry deposition after 2 years".

**Comment R1.90:** *Line 679: …a certain half-hourly value was…*
**Response to R1.90:** Added "ly and "value".

**Comment R1.91:** *Line 688:* …estimated dry depositions for…
**Response to R1.91:** Deleted "s" and rephrased the sentence as follows: "The difference in the annual dry deposition estimates was likely related to the large deposition occurring in February 2018".

**Comment R1.92:** *Line 693: …has a distinct diurnal cycle.*
**Response to R1.92:** Corrected.

**Comment R1.93:** *Lines 701-702: Please delete: The comparison of TRANC measurements with nitrogen throughfall measurements will be shown the second part of this study.*
**Response to R1.93:** The sentence was deleted.

**Comment R1.94:** *Line 705: …total N deposition was…*
**Response to R1.94:** Corrected.

**Comment R1.95:** *Lines 708-709: It suggests that the forest is currently not in a critical state in relation atmospheric N input.*

➔ *I think this statement is incorrect. The N input was 10 and 12 kg N ha−1 a−1, which is within the range of the critical load.*

*According to the OECD, the critical load is defined as:*
*Critical Load is the quantitative estimate of the level of exposure of natural systems to pollutants below which significant harmful effects on specified sensitive elements of the environment do not occur.*

*According to my understanding, the forest is just at the limit of receiving too much Nitrogen from the atmosphere. This implies that N inputs should not increase in the next years.*

**Response to R1.95:** We agree and corrected the statement as follows: "is currently close to the limit of receiving too much nitrogen from the atmosphere assuming that the critical load of the forest site is at the upper end of the reported ranges."

**Comment R1.96:** *Line 713: …above a protected temperate mixed forest, that is located in a remote area.*
**Response to R1.96:** Added "temperate" and ", that is located in a remote area".

**Comment R1.97:** *Line 721: …through the year.*
**Response to R1.97:** Corrected.

**Comment R1.98:** *Line 726: …periods of high solar radiation…*
**Response to R1.98:** Replaced "the timeframe" by "periods" and "global" by "solar".

**Comment R1.99:** *Line 727: seasonal changes in the concentrations of the ΣNr compounds,..*
➔ *Before is was written that ΣNr does not influence vd….*

*Please double check.*
**Response to R1.99:** We meant here the contributions of the individually measured $N_r$ compounds. We corrected the sentence.

**Comment R1.100:** *Line 728: From May to September,  vd was….*
**Response to R1.100:** Deleted.

**Comment R1.101:** **
➔ *This sentence does not make sense, please delete.*
**Response to R1.101:** We deleted the sentence.

**Comment R1.102:** *Line 735: No significant influence of micrometeorological parameters on estimated dry depositions sums was found.*
➔ *This sentence does not make sense and is in contrast to what was written before. (micrometeorology influences vd and therefore also the total N deposition)*

**Response to R1.102:** We agree and rephrased the sentence as follows: "No significant influence of micrometeorological parameters on $\Sigma N_r$ fluxes when using the Mean-Diurnal-Variation approach for filling short-term gaps (up to five days) was found."

**Comment R1.103:** *Line 736-737: Using gap-filling approaches based on inferential modeling for long-term gaps, is an option which we investigate in the companion paper.*
   ➔ *Please delete, note relevant here.*
**Response to R1.103:** We deleted the sentence.

**Comment R1.104:** *Please add information to the conclusion that dry deposition contributed 1/3 to the total N deposition.*
**Response to R.104:** We added the following sentence to line 739. "Thus, dry deposition contributed approximately 1/3 to the total N deposition."

**Comments to the supplement**

**Comment R1.105:** *A1 Description of wet deposition measurements*
**Response to R.105:** Replaced "to" by "of".

**Comment R1.106:** *Figure S3: …shown as box-and-whisker plots….*
**Response to R.106:** Replaced "depicted" by "shown" and added "-and-whisker plots"

**Comment R1.107:** *Figure S4. Mean diurnal cycle of $\Sigma N_r$ concentrations ($\mu g\ N\ m_{-3}$) based on half-hourly measurements for every month from June 2016 to June 2018.*
**Response to R.107:** Replaced "daily" by "diurnal", "on half-hourly basis" by "based on half-hourly measurements", and corrected the order accordingly.

**Comment R1.108:** *Figure S5: … presented by box-and-whisker plots…*
**Comment R1.108:** Replaced "depicted as" by "presented by" and added "-and-whisker plots"

**Comment R1.109:** *Figure S6. Mean diurnal cycle of vd($\Sigma N_r$) (cm s−1) based on half-hourly measurements for every month from June 2016 to June 2018.*
**Response to R1.109:** Replaced "daily" by "diurnal", "on half-hourly basis" by "based on half-hourly measurements", and corrected the order accordingly.

**Comment R1.110:** *Figure S11. Diurnal cycles…*
**Response to R1.110:** Replaced "patterns" by "cycles".

**Comment R1.111:** *Figure S13. ….Wind direction corresponds to values measured in three-hourly intervals.*
**Response to R1.111:** Replaced "at 3-h time stamps" by "in three-hourly intervals"

**References**

Hansen, K., Sorensen, L. L., Hertel, O., Geels, C., Skjoth, C. A., Jensen, B., and Boegh, E.: Ammonia emissions from deciduous forest after leaf fall, Biogeosciences, 10, 4577–4589, https://doi.org/10.5194/bg-10-4577-2013, 2013.

Horii, C. V., Munger, J.W.,Wofsy, S. C., Zahniser, M., Nelson, D., and McManus, J. B.: Fluxes of nitrogen oxides over a temperate deciduous forest, Journal of Geophysical Research: Atmospheres, 109, https://doi.org/10.1029/2003JD004326, 2004.

Horii, C. V., Munger, J. W., Wofsy, S. C., Zahniser, M., Nelson, D., and McManus, J. B.: Atmospheric reactive nitrogen concentration and flux budgets at a Northeastern US forest site, Agricultural and Forest Meteorology, 136, 159–174, https://doi.org/10.1016/j.agrformet.2006.03.005, 2006.

Kljun, N., P. Calanca, M.W. Rotach, H.P. Schmid, 2015: A simple two-dimensional parameterisation for Flux Footprint Prediction (FFP). Geosci. Model Dev., 8, 3695-3713. doi:10.5194/gmd-8-3695-2015.

Munger, J. W., Wofsy, S. C., Bakwin, P. S., Fan, S. M., Goulden, M. L., Daube, B. C., Goldstein, A. H., Moore, K. E., and Fitzjarrald, D. R.: Atmospheric deposition of reactive nitrogen oxides and ozone in a temperate deciduous forest and a subarctic woodland: 1. Measurements and mechanisms, Journal of Geophysical Research-Atmospheres, 101, 12639–12657, https://doi.org/10.1029/96JD00230, 1996.

Seinfeld, J. H. and Pandis, S. N.: Atmospheric Chemistry and Physics – From Air Pollution to Climate Change, John Wiley & Sons, New York, USA, 2 edn., 2006.

Thoene, B., Rennenberg, H., and Weber, P.: Absorption of atmospheric $NO_2$ by spruce (Picea abies) trees, New Phytologist, 134, 257–266, https://doi.org/j.1469-8137.1996.tb04630.x, 1996

Van Oss, R., Duyzer, J., and Wyers, P.: The influence of gas-to-particle conversion on measurements of ammonia exchange over forest, Atmospheric Environment, 32, 465 – 471, https://doi.org/10.1016/S1352-2310(97)00280-X, 1998.

Wyers, G. and Duyzer, J.: Micrometeorological measurement of the dry deposition flux of sulphate and nitrate aerosols to coniferous forest, Atmospheric Environment, 31, 333 – 343, https://doi.org/10.1016/S1352-2310(96)00188-4, 1997.

Wyers, G. P. and Erisman, J. W.: Ammonia exchange over coniferous forest, Atmospheric Environment, 32, 441–451, https://doi.org/10.1016/S1352-2310(97)00275-6, 1998.

---

## Author Response (AR4)

**Response to Editor's comments – manuscript BG-2020-364 Forest-atmosphere exchange of reactive nitrogen in a remote region – Part I: Measuring temporal dynamics**

We sincerely thank the editor for thoroughly handling the review process and for her comments to the last remaining points.

Attached to this document, our responses to the technical corrections are given. Comments range from R1.1 to R1.3. Line numbers in the answers, where new information was added to the manuscript, refer to the last revised version. The text which is enclosed by "…" is implemented in the manuscript.

**Response to Editor's comments:**

**Comments to the manuscript:**

**Comment R1.1:** *Section heading 3.4: …dry deposition sums on micrometeorological variables*
**Response to R1.1:** We changed the section title.

**Comment R1.2:** *Line: 639-641: The low correlations of the ΣNr fluxes to micrometeorological variables could be related to time-shifts between exchange processes and micrometeorological changes. ( → please use variations instead of changes)*

*I do not fully agree with this sentence. It could also be related to multiple interactions and feedback mechanisms, which are hard to quantify.*

**Response to R1.2**: We agree that time-shifts are not the main reason for the low correlations. We replaced variations by changes and rephrased the sentence as follows: "The low correlations of the $\Sigma N_r$ fluxes to micrometeorological variables could be related to, for example, time-shifts between exchange processes and micrometeorological variations, multiple (chemical) interactions between the $N_r$ compounds, and feedback mechanisms, which are difficult to quantify."

**Comment R1.3:** Line 785: *In a follow-up paper, a comparison of …*
**Response to R1.3:** We changed the beginning of the sentence.